# Practical Diffusion Planning via Temperature-Guided Reward Conditioning

## Abstract

Diffusion planners address sequential decision-making by framing plan generation as a generative modeling task over trajectories, mitigating compounding errors and myopic predictions typical of autoregressive methods. They sample long-horizon, globally consistent plans in a single pass, enabling parallel refinement and robust handling of multimodal futures. Reward conditioning is typically achieved through classifier guidance or classifier-free guidance (CFG), with CFG favored for its performance and flexibility but requiring extensive, task-specific hyperparameter tuning that limits scalability and generalization. Our analysis reveals that guidance performance hinges on careful adaptation to the data manifold and reward distribution, contributing to CFG's hyperparameter fragility. In this work, we propose the temperature-guided diffusion planner (TGDP), which adapts CFG for reward conditioning by self-calibrating to these task-specific characteristics. TGDP leverages temperature-based sample reweighting during training and adaptive guidance scaling at inference, yielding robust high-reward plan generation without per-task hyperparameter optimization. Across standard reward-driven benchmarks, TGDP matches performance of prior methods while maintaining a single set of default hyperparameters, establishing a practical, scalable, and generalizable approach to diffusion-based planning.

## 1 Introduction

Learning-based decision making approaches offer significant advantages over traditional control methods that rely on manually derived models. Two prominent paradigms in this context are model-free and model-based reinforcement learning. Model-free approaches learn policies that map system states directly to control actions, achieving strong performance but at the cost of high sample complexity. Model-based methods train world models to plan action sequences. While model-based approaches offer better sample efficiency and the ability to generalize across tasks, they are often hindered by inaccuracies in learned models. These can lead to compounding errors during multi-step predictions, resulting in unrealistic plans that perform poorly in the real world (Chua et al., 2018). Furthermore, objective mismatch, where the training-objective of predicting the future is misaligned with the goal of obtaining high returns, can exacerbate these issues, producing inadequate plans (Lambert et al., 2020).

Recent advances in diffusion planners (Janner et al., 2022; Ajay et al., 2022; Liang et al., 2023; Li et al., 2023; Lee et al., 2023; Dong et al., 2024a; Lu et al., 2025a) have demonstrated remarkable success in overcoming the limitations of other model-based frameworks by reimagining plan generation. Unlike autoregressive approaches, diffusion planners leverage diffusion models to generate globally coherent, long-horizon plans in a single forward pass. By modeling the planning task as a generative process over full trajectories, these methods enable parallel refinement of all plan segments simultaneously, which inherently account for multimodal future possibilities while avoiding the fragility of incremental predictions. During inference, the learned diffusion planner can be guided—via classifier guidance (CG) (Dhariwal & Nichol, 2021) or classifier-free guidance (CFG) (Ho & Salimans, 2022)—to sample trajectories that achieve state-of-the-art performance on diverse reward-driven benchmarks. While CG allows for simpler integration with post hoc constraints, CFG has emerged as the predominant approach in generative modeling, outperforming CG in terms of sample quality (Ho & Salimans, 2022). However, reward-conditioning via CFG typically requires extensive per-task hyperparameter tuning to achieve robust results, limiting its practicality. Consequently, recent research proposes to

instead generate a set of unbiased samples and then use a classifier to select the most promising plan (Lu et al., 2025a). While this bypasses per-task hyperparameter tuning, it relies on the sampling distribution to allocate high likelihood to expert behaviors and increases computational cost.

To address these limitations we adapt classifier-free guidance (CFG) for reward-conditioned diffusion planning by introducing a mechanism that adapts guidance to dataset characteristics. During training, rather than directly conditioning the diffusion model on trajectory returns, we weight batch samples according to their discounted returns and a randomly sampled temperature. At inference, conditioning on temperature enables generation of trajectories with targeted performance levels without requiring task-specific calibration of the conditioning input. Following conventional CFG methodology, we linearly combine unconditional and conditional denoising predictions via a scaling parameter, which in CFG necessitates meticulous tuning. We identify "overguidance", wherein non-convex prediction combinations divert generated samples away from the data distribution, and "underguidance", where insufficient scaling fails to steer sampling toward high-reward regions, as dual failure modes. To mitigate these, our framework automatically infers task-specific guidance sensitivity based on characteristics of the data-manifold and return distribution. We term this approach the temperature-guided diffusion planner (TGDP). Empirical results across a suite of standard reward-driven benchmark environments indicate that TGDP consistently matches the performance of other diffusion planners, all while utilizing a single, fixed set of hyperparameters. More broadly, these results suggest that temperature-guided calibration offers a scalable and reliable approach for diffusion planning in complex environments, necessitating minimal human engineering. Our key contributions are:

1. A temperature-based variant of CFG that self-calibrates to dataset characteristics, bypassing manual hyperparameter tuning for reward-conditioned diffusion planning.

2. A theoretic and experimental analysis of the failure modes of "underguidance" and "overguidance" which contribute to the hyperparameter brittleness of CFG.

3. Empirical validation of TGDP's competitive performance across benchmarks under default hyperparameters, evidencing practical robustness to varied reward structures and dynamics.

## 2 BACKGROUND

**Offline reinforcement learning** Sequential decision making under uncertainty is often formalized as a Markov Decision Process (MDP), defined by $(\mathcal{S}, \mathcal{A}, \mathcal{T}, \mathcal{R}, \gamma)$. Here, $\mathcal{S}$ denotes the set of states, $\mathcal{A}$ the set of actions, $\mathcal{T}(s'|s, a)$ the transition dynamics, $\mathcal{R}(s, a)$ the reward function, and $\gamma \in [0, 1)$ the discount factor. The goal is to find a policy that maximizes the expected discounted return $\mathbb{E}\left[\sum_{t=0}^{\infty} \gamma^t \mathcal{R}(s_t, a_t)\right]$. *Reinforcement learning* (RL) methods address this problem by allowing agents to interact with the environment, learning through trial and error to optimize long-term rewards (Sutton & Barto, 2018). However, acquiring sufficient experience through online interaction can be expensive or unsafe in many real-world scenarios. *Imitation learning* (IL) bypasses environment interaction by learning from expert demonstrations $\mathcal{D} = \{(s_i, a_i)\}$, turning decision-making into supervised learning, which tends to be much more stable than RL. On the other hand, IL methods rely heavily on high-quality expert data, which can be expensive and difficult to obtain, and they often struggle to generalize to unseen states. *Offline RL* methods combine ideas from RL and IL, enabling learning from offline datasets of suboptimal demonstrations $\mathcal{D} = (s_i, a_i, r_i)$, while recovering high-return behaviors that can surpass the demonstrated trajectories (Kumar et al., 2020; Fujimoto et al., 2019). This facilitates cost-efficient deployment of high-performing policies in real-world systems without risky online exploration. A widely used offline RL benchmark is D4RL (Fu et al., 2020), providing standardized environments and datasets designed to evaluate algorithms.

**Diffusion generative models** Diffusion models are a class of generative models that achieve impressive performance in complex domains like image generation (Ho & Salimans, 2022; Dhariwal & Nichol, 2021; Karras et al., 2022). They learn to sample from a data distribution through an iterative denoising process that reverses a predefined noise corruption procedure. We adopt the formulation of Karras et al. (2022) for the case of $s(t) = 1$ and $\sigma(t) = t$. The forward process gradually adds Gaussian noise to data samples $\mathbf{x}_0 \sim p_{data}$ according to a noise schedule $\sigma(t)$ to obtain a noised sample $\mathbf{x}_t = \mathbf{x}_0 + \sigma(t) \cdot \mathbf{n}$, $n \sim \mathcal{N}(\mathbf{0}, \boldsymbol{I})$. A neural network $D_\theta(\mathbf{x}_t; \sigma)$ is trained to predict the original sample by minimizing the weighted mean-squared error

$$\mathcal{L} = \mathbb{E}_{\mathbf{x}, \mathbf{n}, \sigma}\left[\lambda(\sigma) \cdot \|D_\theta(\mathbf{x} + \sigma \cdot \mathbf{n}; \sigma) - \mathbf{x}\|_2^2\right], \tag{1}$$

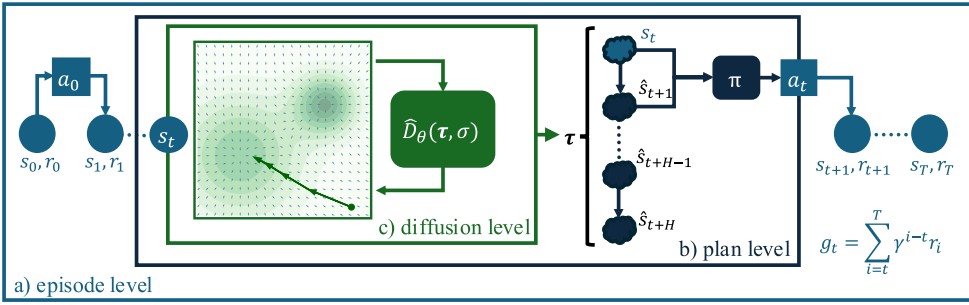

Figure 1: The three operational levels of the diffusion planning framework. a) Episode level: full rollout episodes obtained by repeated planning and environment interaction. b) Plan level: samples drawn from the plan distribution through iterative denoising. c) Diffusion level: score function of the plan distribution/targets of the diffusion model in data-space.

where $\lambda(\sigma)$ weighs different noise levels. At inference, novel samples are generated by initializing $\mathbf{x}_T \sim \mathcal{N}(\mathbf{0}, \sigma^2(T) \cdot \boldsymbol{I})$ and solving the reverse-time stochastic differential equation from $T$ to 0:

$$\frac{d\mathbf{x}}{dt} = -\sigma(t) \cdot \nabla_{\mathbf{x}} \log p(\mathbf{x}; \sigma(t)) \approx \frac{\mathbf{x} - D_\theta(\mathbf{x}; \sigma(t))}{\sigma(t)} \ . \tag{2}$$

Conditioning approaches enable precise control over the generation process: *Classifier Guidance* (Dhariwal & Nichol, 2021) steers generation by training an additional classifier $p_\rho(c|\mathbf{x}, \sigma)$ to predict certain properties $c$ (e.g., class labels) at different noise levels and applying its gradients during the denoising process: $\nabla_{\mathbf{x}} \log p(\mathbf{x}; \sigma(t)|c) = \nabla_{\mathbf{x}} \log p(\mathbf{x}; \sigma(t)) + \nabla_{\mathbf{x}} \log p(c|\mathbf{x}, \sigma(t))$. *Classifier-Free Guidance* (CFG) dispenses with auxiliary classifiers by training a single denoising network $D_\theta(\mathbf{x}; \sigma, c)$ that accepts both the noise level $\sigma$ and an optional condition $c$. During training, the condition is randomly dropped (replaced by a null token $\emptyset$) so that the network learns both conditional and unconditional denoising. At inference, the model is evaluated twice per step, once with the target condition $c$ and once with $\emptyset$, and the outputs are linearly combined as

$$\hat{D}_\theta(\mathbf{x}; \sigma, c) = (1 - s) \cdot D_\theta(\mathbf{x}; \sigma, \emptyset) + s \cdot D_\theta(\mathbf{x}; \sigma, c) \ , \tag{3}$$

where $s$ defines the guidance strength. In practice, optimal performance often occurs for $s > 1$, issuing an interpretation of adding a vector in guidance direction to the unconditional target rather than simple interpolation. CFG is widely adopted for its superior balance of performance and implementation simplicity (Ho & Salimans, 2022). Despite their strengths, diffusion models entail significant computational cost due to the iterative sampling procedure.

**Diffusion models for decision making** Recent advances harness the unique strengths of diffusion models—such as their ability to model complex, multimodal distributions and generate coherent, high-dimensional samples—in a variety of decision-making contexts. In Zhu et al. (2023) approaches were broadly categorized into three subfields: *Diffusion policies* replace simpler policy representations (e.g., Gaussian policies) with expressive generative models that can capture multimodal action distributions (e.g., Chen et al. (2022)). *Diffusion data synthesizers* focus on generating synthetic datasets to support learning processes (e.g., Alonso et al. (2024)). *Diffusion planners* treat plan generation as a trajectory-level denoising problem, sampling full sequences of states (Ajay et al., 2022; Li et al., 2023; Dong et al., 2024a; Lu et al., 2025a), actions (Chi et al., 2024), or state–action pairs (Janner et al., 2022; Liang et al., 2023; Lee et al., 2023) over a fixed horizon. Figure 1 illustrates the architecture of a diffusion planner that plans in state space and generates actions via an inverse dynamics model. The figure delineates three distinct operational levels of the diffusion planner: a) the episode level, which encompasses episodes consisting of repeated interactions between the agent and the environment; b) the plan level, where imagined future state trajectories are generated at each decision step; and c) the diffusion level, which concerns the iterative denoising process used to sample these plans. By treating planning as a generative process over full trajectories, diffusion methods refine all plan segments in parallel, capturing long-range dependencies and complex state–action interactions. This reduces error accumulation common in autoregressive schemes while modeling rich, multimodal future distributions, yielding diverse, high-quality plans that are robust to uncertainty. Despite the high

sample quality, diffusion planners often employ receding horizon control to account for unexpected events or inaccuracies in the prediction. To ground the diffusion model in the agent's current context, many methods employ inpainting conditioning—where a part of the sample is fixed to known values (Janner et al., 2022; Ajay et al., 2022; Li et al., 2023; Dong et al., 2024a; Lee et al., 2023; Liang et al., 2023)—or incorporate auxiliary inputs directly into the model architecture (Chi et al., 2024). Additionally, guided sampling techniques, such as classifier-guidance (Janner et al., 2022; Liang et al., 2023; Lee et al., 2023), classifier-free guidance (Ajay et al., 2022; Li et al., 2023; Dong et al., 2024a), Monte-Carlo sampling with selection (Lu et al., 2025a), or goal conditioning, are frequently used to steer the generative process toward desirable or feasible plans.

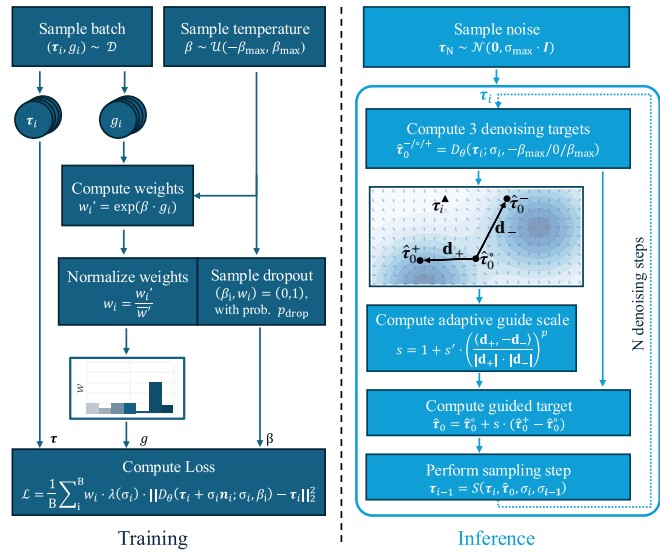

Figure 2: Overview of TGDP. Left: during training, samples are weighted by randomly drawn temperatures. Right: during inference, temperature-conditional and unconditional diffusion targets are combined into an adaptive guidance mechanism.

Many approaches introduce temporal abstraction or hierarchies to improve long-horizon planning.: HDMI (Li et al., 2023) uses hierarchical subgoal generation; DiffuserLite (Dong et al., 2024a) employs coarse-to-fine refinement; and Diffusion Veteran (Lu et al., 2025a) jump-step planning.

## 3 TEMPERATURE GUIDED DIFFUSION PLANNER

We introduce temperature-guided diffusion planning (TGDP), a method designed to bypass classifier-free guidance's (CFG) reliance on tuning its two primary hyperparameters—the conditioning input and the guide scale—by integrating temperature as a central control mechanism. First, we describe a temperature-weighted training scheme and analyze the sampling distribution induced by temperature conditioning. Next, we propose to guide sampling through a temperature-conditioned CFG variant. We conclude by examining how the data manifold and return distribution affect the guidance mechanism and introduce a dynamic adaptation approach tailored to these characteristics. To progressively motivate our design decisions, for each individual component of TGDP, we first explain its respective intuition and realization, and then discuss qualitative evaluations to illustrate its effect. Formal definitions and derivations appear in Appendix A, with additional evaluations in Appendix B. The full algorithm in summarized in Appendix D.1.

**Temperature-weighted training** We propose a temperature-weighted training scheme that decouples conditioning from fixed return targets. For every training batch, a temperature $\beta$ is sampled uniformly from $[-\beta_{\max}, \beta_{\max}]$ and provided to the network as an auxiliary input. During training, for a batch of samples $\tau_i$ and normalized discounted returns $g_i$ (zero mean, unit variance across the full dataset), each sample is assigned an importance weight $w(\beta, g_i)$. In analogy to the dropout probability of CFG, every sample has a probability $p_{\text{drop}}$ of setting the temperature to zero. To prevent high-temperature samples to dominate the loss, we normalize the weights to average to one

$$w(\beta, g_i) = \frac{b \cdot w'(\beta, g_i)}{\sum_{j=0}^{b-1} w'(\beta, g_j)}, \quad w'(\beta, g_i) = \exp(\beta \cdot g_i), \tag{4}$$

where we iterate over the $b$ reweighted samples in the batch. At inference, plan generation solves Equation 2, with denoising targets $D_\theta(\tau; \sigma, \beta_{\text{target}})$. This temperature-modulated reweighting progressively emphasizes trajectories aligned with the return objective for $\beta_{\text{target}} > 0$, deemphasizes them for $\beta_{\text{target}} < 0$, and yields no reweighting at $\beta_{\text{target}} = 0$. As detailed in Appendix A.1 this yields a sampling distribution $\tilde{p}(\tau) \propto p(\tau) \cdot \mathbb{E}_{g \sim p(g|\tau)}(\exp(\beta_{\text{target}} \cdot g))$. Prior methods that condition on a

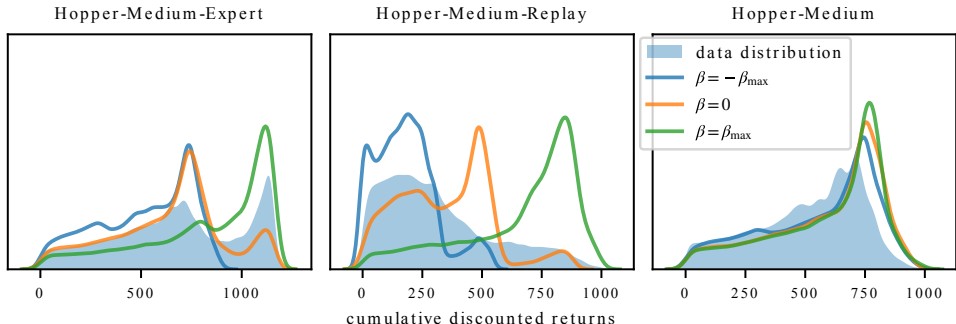

Figure 3: Distribution of cumulative discounted returns for the D4RL hopper locomotion datasets (shaded blue) and for data generated via environment interaction under varying temperature conditions $\beta$. Temperature conditioning reliably reallocates probability mass towards behavioral modes of high/low performance, but fails to consistently push behaviors to the long tails of the data manifold.

specific target return risk out-of-distribution sampling when desired returns conflict with the system's current contextual constraints, potentially inducing unstable or suboptimal behaviors – a limitation typically addressed via task-specific optimization of the conditioning input. In contrast, temperature conditioning reallocates probability mass toward higher-return modes without imposing exact return values, thereby reducing misalignment between return targets and environmental context.

To illustrate the effect of temperature conditioning, Figure 3 shows the distributions of cumulative discounted returns for the three hopper datasets from the D4RL locomotion suite (shaded blue). The *medium* dataset encompasses trajectories of intermediate performance; *medium-expert*, a mixture of intermediate and expert behaviors; and *medium-replay* combines the medium dataset with a long tail of weaker behavior data. Further dataset details are in Appendix E. Discounted return distributions of collected experiences from different temperature conditions $\beta_{\text{target}} \in \{-\beta_{\max}, 0, \beta_{\max}\}$ are also shown. Crucially, temperature conditioning skews the plan distribution (operational level b in Figure 1), not the episode returns directly (level a). Optimal performance therefore necessitates repeated sampling of optimal plans throughout full episodes. The results demonstrate that varying temperature settings systematically bias sampled plans toward low- or high-return data modes with dataset-specific effects: In *medium-expert*, high temperature likely selects the expert mode; low temperature, the medium mode, while zero temperature samples from both. For *medium replay*, conditioning on high temperature approximately recovers the medium mode, while low temperature yields very low rewards as behaviors are sampled from the low-performing tail of the replay mode. In the *medium* dataset, temperature conditioning has limited effect and is unable to reliably bias sampling towards peak performance. Altogether, temperature conditioning seems to be able to select a specific mode of the data distribution but fails to consistently push behaviors towards the high performing long tail of the individual modes. This is further supported by the quantitative evaluation of the influence of temperature conditioning on episode return in Appendix B.2.

**Temperature-guided sampling** Classifier-free guidance combines unbiased and biased predictions via a guide scale, which must be tuned carefully per dataset to achieve optimal planning performance (see Equation 2). We similarly combine predictions to steer samples towards high returns using

$$\hat{D}_\theta(\boldsymbol{\tau}_t; \sigma) = D_\theta(\boldsymbol{\tau}_t; \sigma, 0) + s \cdot \left( D_\theta(\boldsymbol{\tau}_t; \sigma, \beta_{\max}) - D_\theta(\boldsymbol{\tau}_t; \sigma, 0) \right), \tag{5}$$

which can be interpreted as adding a scaled guidance vector to the unbiased target. Previous studies observed that multi-modal datasets (medium-expert and medium-replay) often require smaller guide scales $s$ than uni-modal ones (medium) (Dong et al., 2024b), consistent with our Table 1. We show in Appendix A.2 that this divergence stems from fundamental differences in how unconditional and conditional predictions interact, contingent on these predictions originating from the same behavioral mode (intra-mode guidance) or distinct modes (inter-mode guidance). During intra-mode guidance, the guidance vector aligns with gradients of the discounted return. This permits strong scaling, with the guide scale $s$ acting as a multiplicative factor on the temperature $\beta_{\max}$. Here, overly conservative scales risks "underguidance", failing to sufficiently exploit latent return gradients and yielding suboptimal plans. During inter-mode guidance, guidance vectors may not represent meaningful directions in data space, prohibiting aggressive extrapolation of denoising targets. Here, large guide

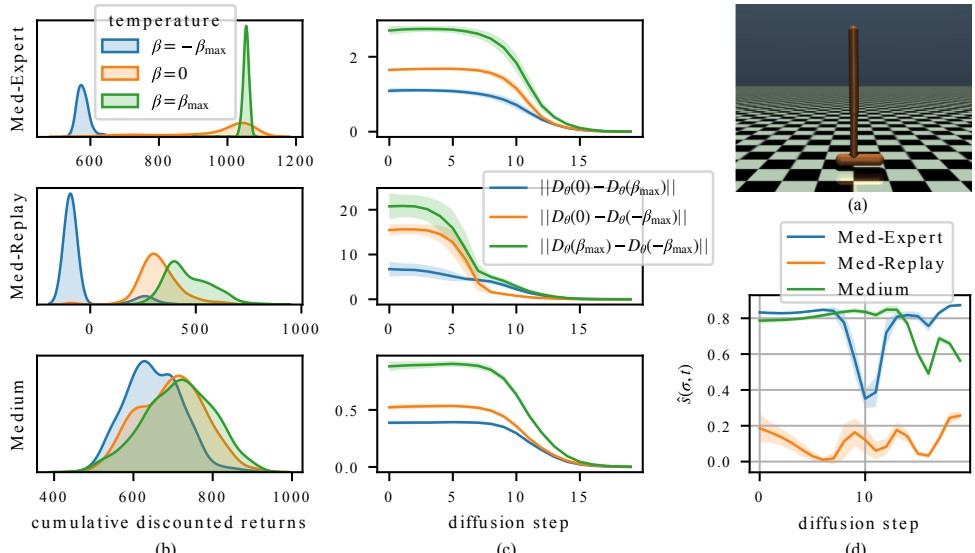

Figure 4: (a) Initial state of the hopper environment. (b) Distributions of cumulative discounted return of potential futures starting from (a) for different datasets and temperature conditions. (c) Pairwise Euclidean distances of temperature-conditioned diffusion targets. (d) Adaptive guide scale based on collinearity of diffusion targets. The strength of guidance crucially depends on whether unbiased and conditioned predictions originate from the same or different probabilistic modes on the data manifold.

scales risk "overguidance", diverting samples away from data marginals and yielding infeasible plans. To illustrate this, Figure 4b shows the distribution over predicted discounted future return of an ensemble of sampled plans (level b in Figure 1), from the initial state in Figure 4a for the three datasets of the hopper environment and for different conditioning temperatures. For the multi-modal datasets, conditioning on high temperature aligns denoising targets with the high performing behavioral mode, such that the guidance vectors connect potentially disjoint sub-manifolds of the data distribution. This is reflected as large magnitudes in Figure 4c, which show the distributions of pairwise Euclidean distances between the denoising targets $D_\theta(\boldsymbol{\tau}; \sigma, \beta)$, $\beta \in \{-\beta_{\max}, 0, \beta_{\max}\}$ throughout a denoising process (level c). Here, the guide scale $s$ should be chosen carefully to avoid "overguidance". Conversely, the single-mode dataset yields diffusion targets from the same mode, resulting in smaller guidance vectors that align with the gradients of the return surface as showcased in Appendix B.4. Here, large guide scales should be used to exploit the latent return gradients and avoid "underguidance". This is further supported by our empirical analysis in Appendix B.3, where we show that some datasets necessitate high guide scales to reach maximal performance, whereas other datasets show steep performance drops for excessive guidance.

**Adaptive guide scales**  We now propose using temperature conditioning across return regimes to dynamically estimate guidance sensitivity, enabling adaptive guidance scales that mitigate under/over-guidance. As detailed in Appendix A.3, under a smooth return surface (illustrated in Appendix B.1), diffusion targets conditioned on different temperatures exhibit approximate collinearity when confined to the same probabilistic mode. Conversely, targets originating from distinct modes violate collinearity. From these geometric considerations we propose an adaptive guide scale based on cosine-similarity

$$s(\boldsymbol{\tau}, \sigma) := 1 + s' \cdot \underbrace{\left( \frac{\langle \mathbf{d}_+(\boldsymbol{\tau}, \sigma), -\mathbf{d}_-(\boldsymbol{\tau}, \sigma) \rangle}{\|\mathbf{d}_+(\boldsymbol{\tau}, \sigma)\| \cdot \|\mathbf{d}_-(\boldsymbol{\tau}, \sigma)\|} \right)^p}_{:= \hat{s}(\boldsymbol{\tau}, \sigma)}, \text{ with}$$

$$\mathbf{d}_\pm(\boldsymbol{\tau}, \sigma) := D(\boldsymbol{\tau}; \sigma, \pm\beta_{\max}) - D(\boldsymbol{\tau}; \sigma, 0), \tag{6}$$

where $s'$ and $p$ are hyperparameters that define the range of the guidance signal and the strength of attenuation from non-collinearity, respectively. Other collinearity measures based on geometric projections yielded comparable results. This adaptive guidance mechanism incurs additional compu-

Table 1: Comparison of performance, sampling complexity per planning step, and optimal hyper-parameters for each guidance method. Here, $D_{\mathrm{fw}}$ and $C_{\mathrm{fw}}$ is the diffusion model's and classifier's forward-pass cost and $C_{\mathrm{bw}}$ the latter's backward cost. Hyperparameters are tuned with Optuna (Akiba et al., 2019) for 100 trials. Performance is averaged over 300 evaluations; higher is better. CG and MCSS underperform; MCSS incurs high compute; CG and CFG need per-task tuning. TGDP achieves robust, high performance with low complexity and default hyperparameters.

| | | CG | | MCSS | | CFG | | TGDP | |
|---|---|---|---|---|---|---|---|---|---|
| Planning Complexity: | | $\mathcal{O}\big(20D_{\mathrm{fw}}+40(C_{\mathrm{fw}}+C_{\mathrm{bw}})\big)$ | | $\mathcal{O}\big(1000D_{\mathrm{fw}}+50C_{\mathrm{fw}}\big)$ | | $\mathcal{O}\big(40D_{\mathrm{fw}}\big)$ | | $\mathcal{O}\big(60D_{\mathrm{fw}}\big)$ | |
| **Environment** | **Dataset** | Value | Params | Value | Params | Value | Params | Value | Params |
| Halfcheetah | Med-Expert | 89.8 | $s$: 0.01 \| $\sigma_s$: 0.40 | 90.3 | $n$: 50 | 93.6 | $s$: 2.5 \| $c$: 1.0 | 92.8 | $s'$: 20 \| $p$: 4 \| $\beta_{\max}$: 3 |
| | Med-Replay | 38.9 | $s$: 0.03 \| $\sigma_s$: 0.10 | 40.3 | $n$: 50 | 41.4 | $s$: 1.0 \| $c$: 0.8 | 43.0 | $s'$: 20 \| $p$: 4 \| $\beta_{\max}$: 3 |
| | Medium | 44.3 | $s$: 0.01 \| $\sigma_s$: 0.10 | 42.8 | $n$: 50 | 48.9 | $s$: 6.0 \| $c$: 1.0 | 45.2 | $s'$: 20 \| $p$: 4 \| $\beta_{\max}$: 3 |
| Hopper | Med-Expert | 53.3 | $s$: 0.04 \| $\sigma_s$: 0.40 | 110.7 | $n$: 50 | 111.2 | $s$: 1.0 \| $c$: 0.8 | 110.4 | $s'$: 20 \| $p$: 4 \| $\beta_{\max}$: 3 |
| | Med-Replay | 35.7 | $s$: 0.08 \| $\sigma_s$: 0.10 | 87.6 | $n$: 50 | 96.6 | $s$: 1.5 \| $c$: 1.0 | 96.2 | $s'$: 20 \| $p$: 4 \| $\beta_{\max}$: 3 |
| | Medium | 62.3 | $s$: 0.04 \| $\sigma_s$: 0.25 | 70.2 | $n$: 50 | 98.5 | $s$: 5.0 \| $c$: 0.8 | 98.3 | $s'$: 20 \| $p$: 4 \| $\beta_{\max}$: 3 |
| Walker2D | Med-Expert | 107.2 | $s$: 0.07 \| $\sigma_s$: 0.25 | 108.3 | $n$: 50 | 108.8 | $s$: 1.0 \| $c$: 1.0 | 108.2 | $s'$: 20 \| $p$: 4 \| $\beta_{\max}$: 3 |
| | Med-Replay | 46.6 | $s$: 0.10 \| $\sigma_s$: 0.40 | 68.9 | $n$: 50 | 72.2 | $s$: 1.5 \| $c$: 0.8 | 80.2 | $s'$: 20 \| $p$: 4 \| $\beta_{\max}$: 3 |
| | Medium | 67.3 | $s$: 0.04 \| $\sigma_s$: 0.25 | 76.7 | $n$: 50 | 77.9 | $s$: 6.0 \| $c$: 0.9 | 78.9 | $s'$: 20 \| $p$: 4 \| $\beta_{\max}$: 3 |
| | **Average** | 60.6 | | 77.3 | | 83.2 | | 83.7 | |

tational cost, requiring an additional function evaluation (low, zero, and high temperature).

Figure 4d shows the distribution of $\hat{s}(\boldsymbol{\tau}, \sigma)$ throughout the denoising process for the potential futures from the state shown in (a) for the three datasets. Notably, there appear to be critical timepoints during the denoising process at which probabilistic modes diverge. This observation supports our earlier conjecture regarding the existence of two qualitatively distinct regimes: intra-modal and inter-modal guidance. When such mode separation occurs, our method adaptively reduces the guide scale to prevent overguidance, while permitting stronger guidance when steering toward the long tail of the return distribution within a single mode.

The full TGDP algorithm, summarized in Figure 2, combines temperature-weighted training in Equation 4 with temperature-conditioned guidance in Equation 5, and the adaptive guide scale in Equation 6. Concise ablations to isolate individual contributions of the components and highlight their synergistic effects appear in Appendix C.1. This approach reduces reliance on fixed guide scales, which inherently struggle to optimally exploit return gradients while preventing manifold deviation.

## 4 EXPERIMENTS AND RESULTS

We here present experimental evaluation of TGDP. Each model was trained for 1M steps on an Nvidia GeForce RTX-4090 or Nvidia H100 GPU. Training of TGDP models takes about two hours for our default configuration; testing for 100 episodes each with 1000 decision steps, two minutes. Additional experiments and ablations appear in Appendix C.

**Comparison of guidance mechanisms** We first compare our temperature-based diffusion guidance approach against other conditioning approaches: classifier-free guidance with explicit return conditioning (CFG), classifier-guidance (CG), and Monte-Carlo sampling with selection (MCSS). For a fair comparison, we alter only the guidance mechanism and keep all other experimental settings identical (see Appendix D.2). For each method we optimize respective sampling hyperparameters: guide scale $s$ and the noise level $\sigma_s$ after which gradients are no longer applied for CG, guide scale $s$ and conditioning input $c$ for CFG, number of candidate samples $n$ for MCSS, and guide scale range $s'$, temperature $\beta_{\max}$, and non-collinearity attenuation $p$ for TGDP. Table 1 compares performance, sampling complexity, and optimal hyperparameters for the four guidance approaches on the D4RL locomotion benchmark. Hyperparameters were tuned with Optuna (Akiba et al., 2019) over 100 trials. Because MCSS and TGDP show stable performance across reasonable settings, all results are reported using a single default configuration. Empirically, TGDP performs on par with CFG and outperforms CG and MCSS, with MCSS struggling in regimes where high-return behavior has weak support under the dataset. For computational analysis, let $D_{\mathrm{fw}}$ and $C_{\mathrm{fw}}$ denote the forward-pass costs of the diffusion model and classifier, respectively, and let $C_{\mathrm{bw}}$ denote the classifier's backward-pass cost. TGDP increases CFG's sampling cost by 50%, due to one additional evaluation for the adaptive guidance. In contrast, CG is more expensive because it requires multiple forward and backward

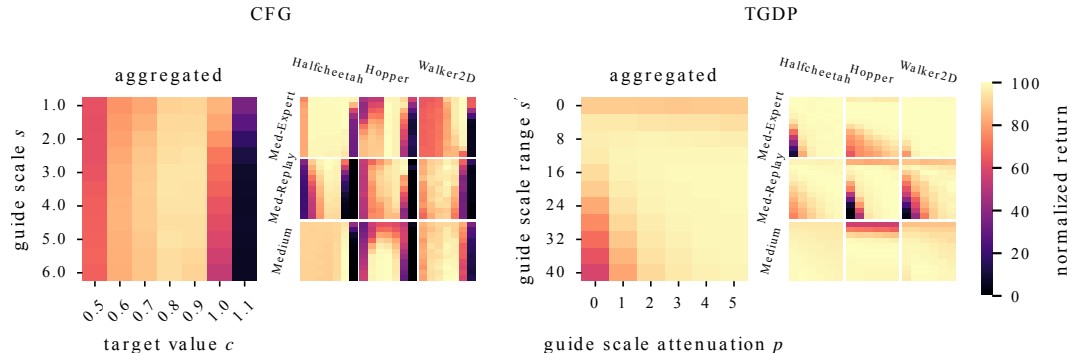

Figure 5: Hyperparameter grids for CFG (guide scale $s$, target value $c$; left) and TGDP (guide scale range $s'$, guide scale attenuation $p$; right) for individual locomotion environments and aggregated across tasks. Colors show mean episode return normalized to the per-task maximum, averaged over three runs × 300 evaluations; higher is better. CFG's brittle per-task hyperparameter optima prevent a single default configuration that consistently reaches high performance. TGDP's adaptive mechanism mitigates brittleness and yields broad high-performing regions, enabling unified defaults that reach ∼ 99% of the observed per-task maxima on average.

passes of the classifier to obtain gradients (here shown for the conventional two gradient steps), and MCSS is the most compute-intensive owing to its linear scaling from parallel planning of $n$ futures. Further evaluations of trading off task performance and inference cost appear in Appendix C.2.

To identify a unifying hyperparameter set for TGDP, we analyze cross-task performance sensitivity. Figure 5 displays hyperparameter grids for CFG (guide scale $s$, target value $c$; left) and TGDP (guide scale range $s'$, guide scale attenuation $p$; right), showing both individual D4RL locomotion task performance and task-aggregated results. Here, TGDP's temperature hyperparameter is not swept over but fixed to $\beta_{\max} = 3$. In practice, this value should be sufficiently large for temperature-conditioned diffusion targets to diverge during inter-modal steering, which is critical for the validity of the adaptive guide scale. At the same time, it should remain small enough to prevent single reweighted samples from dominating the loss, since excessive weighting can destabilize training. As analyzed in Appendix B.3 for values $\beta_{\max} \in \{1, 3, 5\}$, maximal performance across datasets can be obtained for wide temperature ranges. However, because temperature and guide scale interact multiplicatively, adjusting $\beta_{\max}$ also necessitates corresponding retuning of $s'$. Colors represent mean episode returns over three trainings × 300 evaluations, normalized by their respective per-task maxima. High aggregated values indicate configurations that generalize across tasks. CFG exhibits narrow, task-specific hyperparameter optima, underscoring its sensitivity to tuning and hindering robust high performance under a single default configuration. In contrast, TGDP's adaptive guidance produces broader plateaus of high performance. At the borders of reasonable settings, underguidance manifests as depressed returns for very low $s'$, while overguidance appears as sharp performance drops at high $s'$ when $p$ is small. Across environments, TGDP supports a broad spectrum of settings that achieve approximately 99% of each individual task's peak performance. Consequently, TGDP can reproduce CFG's high-reward behaviors using a single, default hyperparameter configuration.

**Comparison of diffusion planners** We now compare the performance of TGDP with other diffusion planners, a diffusion policy and a diffusion data synthesizer on the D4RL locomotion, maze2d, and kitchen benchmarks in Table 2. Additional results on complex manipulation tasks can be found in Appendix C.3. TGDP results show mean and standard error over three trainings × 300 evaluations; higher is better. Performance for other methods follows Dong et al. (2024b), which offers a unified implementation for reliable cross-method comparison. For approaches not included in that framework, we report results of original publications. For TGDP we use the default configuration derived above ($\beta_{\max} = 3, s' = 20, p = 4$) across all benchmarks. The additional environments therefore serve as a test set without additional hyperparameter adaption. TGDP's performance is largely on par with other diffusion planners that typically require per-task tuning. An exception is Diffusion Veteran (DV) (Lu et al., 2025a), which utilizes unbiased sampling with classifier-based selection (MCSS), also bypassing hyperparameter optimization. However, the requirement of substantial probability mass on expert

Table 2: Performance comparison of diffusion planners and other methods on D4RL locomotion, maze2d, and kitchen benchmarks. TGDP results show mean and standard error over three runs × 300 evaluations; higher is better. Where possible, other results are taken from Dong et al. (2024b) or from the respective original papers. TGDP-V combines the TGDP guidance framework with architectural advances from DV (Lu et al., 2025a): jump-step planning and diffusion inverse dynamics models. TGDP largely matches other methods with consistent performance across datasets and architectures using a default configuration. TGDP-V bridges the performance gap between TGDP and DV.

| Dataset | Environment | Diffusion Data Synth./Policy | | Diffusion Planners | | | | | | | |
| | | SynthER | DQL | Diffuser | DD | AdaptDiff | HDMI | DiffuserLite | DV | TGDP (Ours) | TGDP-V (Ours) |
|---|---|---|---|---|---|---|---|---|---|---|---|
| Halfcheetah | Med-Expert | 94.8 ± 0.0 | 95.5 ± 0.1 | 90.3 ± 0.1 | 88.9 ± 1.9 | 90.4 ± 0.1 | 92.1 ± 1.4 | 90.8 ± 0.9 | 92.7 ± 0.3 | 92.8 ± 0.2 | 93.1 ± 0.1 |
| | Med-Replay | 43.4 ± 0.0 | 47.9 ± 0.0 | 36.0 ± 0.7 | 42.9 ± 0.2 | 36.7 ± 0.8 | 44.9 ± 2.0 | 42.9 ± 0.4 | 45.8 ± 0.1 | 43.0 ± 0.1 | 43.0 ± 0.1 |
| | Medium | 48.3 ± 0.0 | 52.3 ± 0.2 | 43.8 ± 0.1 | 45.3 ± 0.3 | 44.3 ± 0.2 | 48.0 ± 0.9 | 48.6 ± 0.7 | 50.4 ± 0.0 | 45.2 ± 0.0 | 45.2 ± 0.0 |
| Hopper | Med-Expert | 76.6 ± 0.4 | 111.1 ± 0.4 | 107.2 ± 0.9 | 110.4 ± 0.6 | 109.3 ± 0.3 | 113.5 ± 0.9 | 110.3 ± 0.3 | 110.0 ± 0.5 | 110.4 ± 0.1 | 110.2 ± 0.1 |
| | Med-Replay | 24.7 ± 0.1 | 101.6 ± 0.0 | 91.8 ± 0.5 | 99.2 ± 0.2 | 91.2 ± 0.1 | 99.6 ± 1.5 | 97.8 ± 1.3 | 91.9 ± 0.0 | 96.2 ± 0.1 | 95.5 ± 0.1 |
| | Medium | 51.9 ± 0.1 | 96.5 ± 1.3 | 89.5 ± 0.7 | 98.2 ± 0.1 | 95.5 ± 1.1 | 76.4 ± 2.6 | 99.5 ± 0.7 | 83.6 ± 1.2 | 98.3 ± 0.2 | 97.0 ± 0.0 |
| Walker2D | Med-Expert | 110.0 ± 0.0 | 111.6 ± 0.0 | 107.4 ± 0.1 | 108.4 ± 0.1 | 107.7 ± 0.1 | 107.9 ± 1.2 | 106.4 ± 0.3 | 109.2 ± 0.0 | 108.2 ± 0.0 | 108.9 ± 0.0 |
| | Med-Replay | 88.6 ± 0.4 | 98.2 ± 0.1 | 58.3 ± 1.8 | 75.6 ± 0.6 | 82.9 ± 1.5 | 80.7 ± 2.1 | 84.6 ± 1.7 | 85.0 ± 0.5 | 80.2 ± 0.5 | 80.5 ± 0.5 |
| | Medium | 86.6 ± 0.0 | 86.8 ± 0.0 | 79.4 ± 1.0 | 79.6 ± 0.9 | 83.8 ± 1.1 | 79.9 ± 1.8 | 85.1 ± 0.5 | 82.8 ± 0.1 | 78.9 ± 0.4 | 78.5 ± 0.4 |
| | **Average** | 69.4 | 89.0 | 78.2 | 83.2 | 82.4 | 82.6 | 85.1 | 83.5 | 83.7 | 83.6 |
| Maze2D | Umaze | – | – | 113.9 ± 3.1 | 116.2 ± 2.7 | 135.1 ± 5.8 | 120.1 ± 2.5 | – | 136.6 ± 1.3 | 137.0 ± 1.4 | 138.7 ± 1.4 |
| | Medium | – | – | 121.5 ± 2.7 | 122.3 ± 2.1 | 129.9 ± 4.6 | 121.8 ± 1.6 | – | 150.7 ± 1.0 | 150.0 ± 1.1 | 153.1 ± 1.0 |
| | Large | – | – | 123.0 ± 6.4 | 125.9 ± 1.6 | 167.9 ± 5.0 | 128.6 ± 2.9 | – | 203.6 ± 1.4 | 186.1 ± 2.1 | 200.9 ± 1.5 |
| | **Average** | – | – | 119.5 | 121.5 | 144.3 | 123.5 | – | 163.6 | 157.7 | 164.3 |
| Kitchen | Mixed | 0.0 ± 0.0 | 62.5 ± 1.5 | 52.5 ± 2.5 | 75.0 ± 0.0 | 51.8 ± 0.8 | 69.2 ± 1.8 | 73.6 ± 0.7 | 73.6 ± 0.1 | 67.6 ± 0.4 | 72.7 ± 0.3 |
| | Partial | 0.0 ± 0.0 | 63.5 ± 1.8 | 55.7 ± 1.3 | 56.5 ± 5.8 | 55.5 ± 0.4 | – | 74.4 ± 0.6 | 94.0 ± 0.3 | 80.8 ± 0.8 | 96.0 ± 0.6 |
| | **Average** | 0.0 | 63.0 | 54.1 | 65.8 | 53.7 | – | 74.0 | 83.8 | 74.2 | 84.3 |

behaviors paired with the increased computational costs limits the applicability of MCSS. Notably, TGDP fails to fully match the performance of DV on the maze2d and kitchen benchmarks. However, as a guidance strategy, TGDP is not tied to any specific planner architecture. To test this, we present TGDP-Veteran (TGDP-V), which adopts two of DV's architectural advances: jump-step planning and diffusion inverse-dynamics models. TGDP-V matches DV on all benchmarks, indicating that advanced architectural choices can yield further gains and supporting TGDP's architectural robustness. Overall, these results demonstrate that TGDP maintains high and consistent performance across diverse domains and architectures without per-task tuning. This streamlines deployment and shows that TGDP can flexibly adapt to diverse settings, making it an effective and practical tool.

## 5 RELATED WORK

Diffusion planning for offline RL relies on the ability of guidance paradigms to bias the generation process towards high-reward subspaces of the data distribution. To this end, previous works have adopted CG (Janner et al., 2022; Liang et al., 2023; Lee et al., 2023), or CFG (Ajay et al., 2022; Li et al., 2023; Dong et al., 2024a) to empower diffusion planners to synthesize trajectories that are not only coherent and feasible, but also aligned with the reward signal. Despite CFG's prevalence in generative domains (e.g., image synthesis), many works in the decision-making context make use of CG (Lu et al., 2025a). However, as classifiers are not explicitly trained to yield meaningful gradients, they run the risk to exploit "shortcuts" in their predictions, leading to suboptimal gradient directions that ultimately degrade sample quality (Dieleman, 2022). To counterbalance this, Janner et al. (2022) employs relatively light CG and doubles the learned classifier as a selector of the most promising plan from an ensemble. CFG on the other hand shows considerable brittleness, requiring extensive hyperparameter tuning tailored to each specific task (Ajay et al., 2022; Dong et al., 2024b;a; Lu et al., 2025a). One explanation, as noted by Lu et al. (2025a), is that the optimal conditioning value for CFG may shift throughout an episode, since the range of achievable future rewards can vary significantly depending on the agent's progress. Yet, in practice, CFG relies on a fixed target value, potentially leading to misaligned state context and return conditions and yielding unrealistic plans.

Infeasible behaviors resulting from manifold deviation are a recognized challenge in guided diffusion planning. Common practice constrains diffusion targets to marginals via simple clipping; however, this often fails to produce meaningful behaviors and may bias samples towards boundary values (Lou & Ermon, 2023). Advanced approaches address manifold deviation in various ways: by predicting plan feasibility—through stability estimation under re- and subsequent denoising steps (Lee et al., 2023) or via exploratory value estimates (Liu et al., 2025)—and using these predictions to guide sampling toward the data manifold; by replacing the classifier's traditional MSE loss with a contrastive energy-based objective that enables exact and stable classifier guidance estimation (Lu et al., 2023); by

leveraging human feedback during generation to steer samples away from implausible or low-quality regions (Wang et al., 2025); by generating plan ensembles and using tree-based state aggregation to identify and prioritize reliable states through consensus-based weighting (Feng et al., 2024); or by locally approximating the data tangent space and projecting diffusion targets onto it during sampling to enforce manifold consistency (Lee & Choi, 2025). Recent work proposes Monte Carlo sample selection (MCSS) to bypass guidance entirely, instead generating diverse, unbiased trajectory ensembles and selecting high-reward candidates via a noise-free classifier (Lu et al., 2025a). Like our approach, MCSS avoids unstable guidance and requires little hyperparameter tuning. However, its success hinges on the generative model's coverage: the ensemble must contain expert-like trajectories to ensure high-reward samples exist. This makes MCSS effectiveness tightly coupled to the data distribution. Furthermore, generating large plan ensembles escalates computational demands.

## 6    LIMITATIONS AND FUTURE WORK

We here discuss limitations regarding computational overhead and potential failure modes of TGDP and identify exciting directions for further refinement of the method. Firstly, even though TGDP shows moderate computational complexity compared to other guidance paradigms, it does add one additional neural evaluation per diffusion step leading to a 50% increase in computational complexity w.r.t. regular CFG. This may prove problematic in systems that work on constrained resource or time budgets or utilize very deep models. Common strategies to decrease computational complexity include the adoption of diffusion variants that reduce the number denoising steps required for high-quality samples (Salimans & Ho, 2022; Liu et al., 2023), warm-starting the diffusion process with partially re-noised plans from a previous decision step (Dong et al., 2024b), or distilling the trained planner into a lightweight policy (Lu et al., 2025b). In Appendix C.2, we provide initial insights into trade-offs between computational complexity and performance for our framework. However, it remains unclear how TGDP's adaptive guide scale interacts with aggressively reduced number of denoising steps, rectified flows, or initial diffusion samples at low noise-regimes around the data manifold, thus leaving ample opportunities for further research.

A second limitation is the unclear generalization to arbitrary task settings. In our derivations in Appendix A, we constrain datasets to consist of well-separated behavioral modes that individually emit smooth value surfaces. While we formalize the intuition behind TGDP's guidance approach and adaptive scaling, we do not provide a formal convergence analysis of the sampling algorithm. Consequently, there may exist datasets that violate the proposed constraints or interact adversely with our proposed framework, leading to diminished performance or even catastrophic failure of TGDP. In particular, as suggested by our experiments in Appendix C.3, mixed-quality teleoperation datasets represent a challenge, as they may violate the proposed constraints. In addition, while our evaluation shows largely strong performance of TGDP, it focuses on a select set of benchmark environments. Exploring generalization to more diverse datasets, including complex robotic environments with sensory state input, as well as a further formal analysis of temperature guidance to better understand limitations of TGDP therefore remain exciting directions for future work.

## 7    CONCLUSION

This work introduces Temperature-Guided Diffusion Planning (TGDP), a novel guidance framework that addresses critical limitations of static guide scales and conditioning inputs of classifier-free guidance in diffusion planning. By integrating temperature-conditional loss reweighting as a central control mechanism and deriving adaptive guidance from geometric relationships between diffusion targets, TGDP dynamically balances the exploitation of latent return gradients with truthfulness to the data manifold. Our analysis reveals that multi-modal datasets inherently induce high guidance vector magnitudes requiring attenuation during inter-modal steering, while single-mode datasets benefit from amplified intra-modal guidance—a distinction overlooked by prior fixed-scale approaches. Empirical evaluations across several benchmarks and for two diffusion planning architectures demonstrate that TGDP achieves performance competitive with other diffusion planners while using a set of default hyperparameters, thereby eliminating the need for laborious per-task tuning. This adaptability stems from its capacity to automatically detect and respond to geometric properties of the learned trajectory distribution, making it both robust to dataset heterogeneity and practical for real-world deployment.

## ACKNOWLEDGMENTS

Language polishing assistance was provided with an LLM; all content was verified by the authors.

## REPRODUCIBILITY STATEMENT

Anonymized code, configuration files, and experiment scripts to reproduce all main results are available at `https://github.com/iclr26-submission24685/tgdp_submission.git`. Algorithm pseudocode appears in Appendix D.1. Hyperparameters are listed in Appendix D.2. Compute details (GPU and runtime) are discussed in Section 4 and Appendix C.2.

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

## A  FORMALIZATION OF TEMPERATURE GUIDED DIFFUSION PLANNING

This appendix develops the theoretical foundations underlying loss reweighting for diffusion models when the data distribution is a mixture of components and the loss is modulated by sample-dependent weights. It shows how reweighting induces a skewed sampling distribution, alters mixture masses and component shapes, and propagates through Gaussian smoothing to the score in Appendix A.1. Building on these results, it derives a classifier-free style guidance rule that combines unskewed and skewed scores, clarifies when guidance acts intra-mode versus inter-mode, analyzes over/underguidance regimes in Appendix A.2, and proposes an adaptive guide scale that mitigates these pathologies in Appendix A.3. We conclude with practical notes on implementation in Appendix A.4. We here discuss the case of an unconditional data distribution $p(\mathbf{x})$. The treatment can be easily extended to conditional distributions $p(\mathbf{x}|c)$.

### A.1  SKEWING SAMPLING DISTRIBUTIONS THROUGH REWEIGHTING

We will first discuss the implications of reweighting the loss in the EDM objective, specifically if the data-distribution is a mixture of components. We will then propose a reweighting scheme and discuss its implications.

**EDM objective and sampling distribution under reweighting**  Recall the EDM objective function from Equation 1. We now introduce a weight $w(g)$ that depends on a conditional random variable $g \sim p(g|\mathbf{x})$. The objective function then becomes

$$\tilde{\mathcal{L}} = \mathbb{E}_{\mathbf{x} \sim p(\mathbf{x})} \mathbb{E}_{g \sim p(g|\mathbf{x})} \mathbb{E}_{\sigma \sim p(\sigma)} \mathbb{E}_{\mathbf{n} \sim \mathcal{N}(\mathbf{0}, \mathbf{I})} \left[ w(g) \cdot \lambda(\sigma) \cdot \| D_\theta(\mathbf{x} + \sigma \cdot \mathbf{n}; \sigma) - \mathbf{x} \|^2 \right], \tag{7}$$

$$= \mathbb{E}_{\mathbf{x} \sim p(\mathbf{x})} \left[ \mathbb{E}_{g \sim p(g|\mathbf{x})} \left[ w(g) \right] \cdot \mathbb{E}_{\sigma \sim p(\sigma)} \mathbb{E}_{\mathbf{n} \sim \mathcal{N}(\mathbf{0}, \mathbf{I})} \left[ \lambda(\sigma) \cdot \| D_\theta(\mathbf{x} + \sigma \cdot \mathbf{n}; \sigma) - \mathbf{x} \|^2 \right] \right], \tag{8}$$

$$= \int_{\mathbf{x}} p(\mathbf{x}) \cdot \mathbb{E}_{g \sim p(g|\mathbf{x})} \left[ w(g) \right] \cdot \mathbb{E}_{\sigma \sim p(\sigma)} \mathbb{E}_{\mathbf{n} \sim \mathcal{N}(\mathbf{0}, \mathbf{I})} \left[ \lambda(\sigma) \cdot \| D_\theta(\mathbf{x} + \sigma \cdot \mathbf{n}; \sigma) - \mathbf{x} \|^2 \right] d\mathbf{x}. \tag{9}$$

We now define the distribution $\tilde{p}(\mathbf{x}) = p(\mathbf{x}) \cdot \mathbb{E}_{g \sim p(g|\mathbf{x})} \left[ w(g) \right]$ to obtain the objective function $\tilde{\mathcal{L}} = \mathbb{E}_{\mathbf{x} \sim \tilde{p}(\mathbf{x})} \mathbb{E}_{\sigma \sim p(\sigma)} \mathbb{E}_{\mathbf{n} \sim \mathcal{N}(\mathbf{0}, \mathbf{I})} \left[ \lambda(\sigma) \cdot \| D_\theta(\mathbf{x} + \sigma \cdot \mathbf{n}, \sigma) - \mathbf{x} \|^2 \right]$. Following the rationale of diffusion models, sampling from this model using Equation 2, will yield samples of $\tilde{p}(\mathbf{x})$. We set the reweighting function $w(g) = \frac{w'(g)}{Z}$ to obtain $\tilde{p}(\mathbf{x}) = p(\mathbf{x}) \cdot \mathbb{E}_{g \sim p(g|\mathbf{x})} \left[ \frac{w'(g)}{Z} \right] = \frac{p(\mathbf{x}) \cdot \mathbb{E}_{g \sim p(g|\mathbf{x})} \left[ w'(g) \right]}{Z}$. From the requirement $\int_{\mathbf{x}} \tilde{p}(\mathbf{x}) d\mathbf{x} = 1$ we obtain $Z = \int_{\mathbf{x}} p(\mathbf{x}) \cdot \mathbb{E}_{g \sim p(g|\mathbf{x})} \left[ w'(g) \right] d\mathbf{x} = \mathbb{E}_{\mathbf{x} \sim p(\mathbf{x})} \mathbb{E}_{g \sim p(g|\mathbf{x})} \left[ w'(g) \right]$, so our sampling distribution becomes

$$\tilde{p}(\mathbf{x}) = p(\mathbf{x}) \cdot \frac{\mathbb{E}_{g \sim p(g|\mathbf{x})} \left[ w'(g) \right]}{\mathbb{E}_{\mathbf{x} \sim p(\mathbf{x})} \mathbb{E}_{g \sim p(g|\mathbf{x})} \left[ w'(g) \right]}. \tag{10}$$

Note that the skewed distribution's support is confined to the original distribution's support. In the further derivation we assume $w'(g) > 0$, $\forall g$; so the support matches exactly. We can now replace the expectations in Equation 7 by the unbiased batch estimate

$$\tilde{\mathcal{L}} \approx \frac{1}{B} \cdot \sum_{i=0}^{B-1} w(g_i) \cdot \lambda(\sigma_i) \cdot \| D_\theta(\mathbf{x}_i + \sigma_i \cdot \mathbf{n}_i; \sigma_i) - \mathbf{x}_i \|^2, \quad w(g_i) = \frac{B \cdot w'(g_i)}{\sum_{j=0}^{B-1} w'(g_j)}. \tag{11}$$

**Reweighting mixture distributions**  Suppose that our original data distribution consists of a weighted mixture of underlying distributions $p(\mathbf{x}) = \sum_{i \in \mathcal{I}} \alpha_i \cdot p_i(\mathbf{x})$, where $\sum_{i \in \mathcal{I}} \alpha_i = 1$, with mixture components $p_i(\mathbf{x})$, mixture weights $\alpha_i$, and each component emitting its own measure distribution $g \sim p_i(g|\mathbf{x})$. We do not assume any specific form of the mixture components. However, we require that the individual mixture components represent well separated manifolds within the data space. In practice the overlap of the support of the components should be limited, which is a relatively mild assumption in high dimensions. We now apply the reweighting scheme from Equation 10 to obtain the reweighted mixture distribution $\tilde{p}(\mathbf{x})$. Using the joint distribution $p(\mathbf{x}, g) =$

$p(\mathbf{x}) \cdot p(g|\mathbf{x}) = \sum_{i \in \mathcal{I}} \alpha_i \cdot p_i(\mathbf{x}) \cdot p_i(g|\mathbf{x})$, we can rewrite the numerator in Equation 10

$$\mathbb{E}_{g \sim p(g|\mathbf{x})}\left[w'(g)\right] = \int p(g|\mathbf{x}) \cdot w'(g)\, dg = \int \frac{p(\mathbf{x}, g)}{p(\mathbf{x})} \cdot w'(g)\, dg \,, \tag{12}$$

$$= \sum_{i \in \mathcal{I}} \frac{\alpha_i \cdot p_i(\mathbf{x})}{p(\mathbf{x})} \cdot \int p_i(g|\mathbf{x}) \cdot w'(g)\, dg \,, \tag{13}$$

$$= \sum_{i \in \mathcal{I}} \frac{\alpha_i \cdot p_i(\mathbf{x})}{p(\mathbf{x})} \cdot \mathbb{E}_{g \sim p_i(g|\mathbf{x})}\left[w'(g)\right] \,, \tag{14}$$

and the denominator

$$\mathbb{E}_{\mathbf{x} \sim p(\mathbf{x})}\mathbb{E}_{g \sim p(g|\mathbf{x})}\left[w'(g)\right] = \int p(\mathbf{x}) \cdot \sum_{j \in \mathcal{I}} \frac{\alpha_j \cdot p_j(\mathbf{x})}{p(\mathbf{x})} \cdot \mathbb{E}_{g \sim p_j(g|\mathbf{x})}\left[w'(g)\right]\, d\mathbf{x} \,, \tag{15}$$

$$= \sum_{j \in \mathcal{I}} \alpha_j \cdot \int p_j(\mathbf{x}) \cdot \mathbb{E}_{g \sim p_j(g|\mathbf{x})}\left[w'(g)\right]\, d\mathbf{x} \,, \tag{16}$$

$$= \sum_{j \in \mathcal{I}} \alpha_j \cdot \underbrace{\mathbb{E}_{\mathbf{x} \sim p_j(\mathbf{x})}\mathbb{E}_{g \sim p_j(g|\mathbf{x})}\left[w'(g)\right]}_{:=Z_j} \,, \tag{17}$$

to obtain the reweighted mixture distribution

$$\tilde{p}(\mathbf{x}) = p(\mathbf{x}) \cdot \frac{\sum_{i \in \mathcal{I}} \frac{\alpha_i\, p_i(\mathbf{x})}{p(\mathbf{x})} \cdot \mathbb{E}_{g \sim p_i(g|\mathbf{x})}\left[w'(g)\right]}{\sum_{j \in \mathcal{I}} \alpha_j \cdot Z_j} \,, \tag{18}$$

$$= \sum_{i \in \mathcal{I}} \alpha_i \cdot p_i(\mathbf{x}) \frac{\mathbb{E}_{g \sim p_i(g|\mathbf{x})}\left[w'(g)\right]}{\sum_{j \in \mathcal{I}} \alpha_j \cdot Z_j} \,, \tag{19}$$

$$= \sum_{i \in \mathcal{I}} \underbrace{\frac{Z_i}{\sum_{j \in \mathcal{I}} \alpha_j \cdot Z_j}}_{:=m_i \cdot \alpha_i := \tilde{\alpha}_i} \cdot \alpha_i \cdot \underbrace{\frac{\mathbb{E}_{g \sim p_i(g|\mathbf{x})}\left[w'(g)\right]}{Z_i} \cdot p_i(\mathbf{x})}_{:=s_i(\mathbf{x}) \cdot p_i(\mathbf{x}) := \tilde{p}_i(\mathbf{x})} \,, \tag{20}$$

where $Z_i$ are the respective normalizing constants of the reweighting of component $i$. As we can see, the reweighting of the samples in the loss function both skewed the original mixture components and reweighted the original mixture weights. We here introduce the mass factors $m_i$ and the skewing function $s_i(\mathbf{x})$. The new mixture weights $\tilde{\alpha}_i$ depend on the relative expected values $Z_i$ of $w'(g)$ of the individual mixture components, i.e., mixture components that achieve high expected $w'(g)$ will be favored in the resulting sampling distribution. The new mixture components $\tilde{p}_i(\mathbf{x})$ are individually skewed versions of the original components.

**Boltzmann value reweighting**   In the decision-making framework, $\mathbf{x}$ may represent a description of system behavior (this may include states, actions, or higher level behavior) and $g$ the discounted future return. We here define the reweighting function as

$$w'(g) = \exp\left(\beta \cdot V(\mathbf{x})\right) \,, \tag{21}$$

where $V(\mathbf{x}) = \mathbb{E}_{g \sim p(g|\mathbf{x})}\left[g\right]$ is the value function given a behavior $\mathbf{x}$, and $\beta$ is a temperature parameter that defines the strength of reweighting. This exponential form – common in softmax action selection in RL – tilts the sampling distribution toward regions of higher expected returns under the base distribution. This reweighting can also be associated with the likelihood of the specific behavior being optimal (Levine, 2018). While the function $V(\mathbf{x})$ is generally unknown, it can be estimated from multiple rollouts or we can learn a parametrized model. For the sake of the argument, we will here assume that we have access to $V(\mathbf{x})$ and discuss the implications of estimating it in Appendix A.4. To optimize for high return we can now sample from the reweighted mixture distribution to obtain samples from

$$\tilde{p}(\mathbf{x}) = \sum_{i \in \mathcal{I}} \frac{\mathbb{E}_{\mathbf{x} \sim p_i(\mathbf{x})}\left[\exp(\beta \cdot V_i(\mathbf{x}))\right] \cdot \alpha_i}{Z} \cdot \tilde{p}_i(\mathbf{x}) \,, \quad \tilde{p}_i(\mathbf{x}) = \frac{\exp(\beta \cdot V_i(\mathbf{x}))}{Z_i} \cdot p_i(\mathbf{x}) \,, \tag{22}$$

where $V_i(\mathbf{x}) = \mathbb{E}_{g \sim p_i(g|\mathbf{x})}[g]$ denotes the value function of mixture component $i$. For large $\beta$, differences in the value functions $V_i(\mathbf{x})$ of components $i$ can result in markedly different weights $V_k(\mathbf{x}) > V_i(\mathbf{x}) \implies \mathbb{E}_{\mathbf{x} \sim p_k(\mathbf{x})}[\exp(\beta \cdot V_k(\mathbf{x}))] >> \mathbb{E}_{\mathbf{x} \sim p_i(\mathbf{x})}[\exp(\beta \cdot V_i(\mathbf{x}))], \ \forall i \neq k$ such that the reweighted mixture weights are dominated by $\tilde{\alpha}_k$ and therefore $\tilde{p}(\mathbf{x}) \approx p_k(\mathbf{x}) \cdot \frac{\exp(\beta \cdot V_k(\mathbf{x}))}{Z_k}$.

In theory, this term should also strongly bias the sampling distribution towards high-performing behaviors within the component. However, in practice, due to limited data and model capacity, $V_k(\mathbf{x})$ may lack sufficient local discriminative resolution to reliably shift the components support towards maximal performance. Furthermore, since the estimator of the normalizing constant for finite batch sizes depends on batch composition, the practical strength of reweighting is inherently limited. Finally, any estimated value function crucially depends on the data-generating policy, as $V(\mathbf{x})$ represents the value of behaving according to $\mathbf{x}$ (e.g., if $\mathbf{x}$ is a plan of action for $h$ steps into the future) and following the data-generating policy after, limiting its discriminative power. This problem could be partly alleviated by learning an optimal value function $V^*(\mathbf{x})$ (Sutton & Barto, 2018). Figure 6, demonstrates for the hopper-medium-expert-v2 dataset, that medium and expert behaviors show distinct value functions, while maintaining low variance within each behavioral mode. This allows reweighting to shift probability mass towards the expert mode but limiting internal mass reallocation within either mode. Collectively, these factors contribute to sub-optimal sampling, particularly when components exhibit relatively constant expected returns, high-variance return distributions, or the dataset contains only few high-performing demonstrations. Consequently in practice, loss reweighting alone cannot always ensure optimal performance across all datasets. This hypothesis is supported empirically, as in our experiments (see Figure 7), strong reweighting led to near-optimal performance in the D4RL locomotion *medium-expert* datasets—which exhibit multi-modal data distributions with distinct return distributions per component—whereas it led to moderate improvement on the *medium-replay* datasets—which feature less defined behavioral modes—and minimal improvement on the *medium* datasets—which consists of a single data mode that emits largely homogeneous returns.

## A.2 GUIDING DIFFUSION SAMPLING THROUGH DIFFERENT REWEIGHTS

As discussed above, simply reweighting the loss does not always lead to optimal performance. We now analyze the implications of reweighting loss functions on the score function. We will then propose and discuss a variant of classifier-free guidance that can be used to further bias the sampling distribution towards high performance.

**Score of smoothed mixture distributions** We create a series of probability distributions from our mixture distribution

$$p^\sigma(\mathbf{x}) := (p * \mathcal{N}(0, \sigma))(\mathbf{x}) = \sum_{i \in \mathcal{I}} \alpha_i \cdot \underbrace{(p_i * \mathcal{N}(0, \sigma))(\mathbf{x})}_{p_i^\sigma(\mathbf{x})}. \tag{23}$$

The score of this probability distribution is

$$\nabla_{\mathbf{x}} \log p^\sigma(\mathbf{x}) = \frac{\nabla_{\mathbf{x}} p^\sigma(\mathbf{x})}{p^\sigma(\mathbf{x})} = \frac{\sum_{i \in \mathcal{I}} \alpha_i \cdot \nabla_{\mathbf{x}} p_i^\sigma(\mathbf{x})}{p^\sigma(\mathbf{x})}, \tag{24}$$

$$= \sum_{i \in \mathcal{I}} \alpha_i \cdot \frac{p_i^\sigma(\mathbf{x})}{p^\sigma(\mathbf{x})} \cdot \nabla_{\mathbf{x}} \log p_i^\sigma(\mathbf{x}). \tag{25}$$

We thus get scores that are weighted combinations of the respective scores of the base distributions. The relative weights—-which we call the prior responsibilities—depend on original mixture weights $\alpha_i$ and the likelihood of the noisy sample $\mathbf{x}$ under the smoothed component $p_i^\sigma(\mathbf{x}) = (p_i * \mathcal{N}(0, \sigma))(\mathbf{x})$. We can define a tolerance $\delta$ such that there exists a maximal threshold $\varepsilon(\mathbf{x}, \sigma)$ that selects a subset the mixture components

$$\mathcal{A}(\mathbf{x}, \sigma) := \left\{ i \ \middle| \ \frac{\alpha_i \cdot p_i^\sigma(\mathbf{x})}{p^\sigma(\mathbf{x})} \geq \varepsilon(\mathbf{x}, \sigma) \right\}, \ \frac{\sum_{i \in \mathcal{A}} \alpha_i \cdot p_i^\sigma(\mathbf{x})}{p^\sigma(\mathbf{x})} \geq 1 - \delta. \tag{26}$$

For a sensibly chosen $\delta$, the set $\mathcal{A}$ thus contains the components that significantly contribute to the score and therefore $\nabla_{\mathbf{x}} \log p^\sigma(\mathbf{x}) \approx \sum_{i \in \mathcal{A}} \alpha_i \cdot \frac{p_i^\sigma(\mathbf{x})}{p^\sigma(\mathbf{x})} \cdot \nabla_{\mathbf{x}} \log(p_i^\sigma)$. We refer to $\mathcal{A}$ as the set of active components. Note that this set typically includes all components for very high noise levels $\sigma$ and decreases in size throughout the diffusion process as $\sigma \to 0$.

**Scores under reweighting**    Substituting the skewed distribution $\tilde{p}(\mathbf{x})$ from Equation 20 into Equation 25, we obtain the score of a reweighted mixture distribution as a function of the original distribution

$$\nabla_{\mathbf{x}} \log \tilde{p}^{\sigma}(\mathbf{x}) = \sum_{i \in \mathcal{I}} \tilde{\alpha}_i \cdot \frac{\tilde{p}_i^{\sigma}(\mathbf{x})}{\tilde{p}^{\sigma}(\mathbf{x})} \cdot \nabla_{\mathbf{x}} \log \tilde{p}_i^{\sigma}(\mathbf{x}) , \tag{27}$$

$$= \sum_{i \in \mathcal{I}} \underbrace{\frac{m_i^{\sigma} \cdot \alpha_i \cdot s_i^{\sigma}(\mathbf{x}) \cdot p_i^{\sigma}(\mathbf{x})}{\sum_{j \in \mathcal{I}} m_j^{\sigma} \cdot \alpha_j \cdot s_j^{\sigma}(\mathbf{x}) \cdot p_j^{\sigma}(\mathbf{x})}}_{:= \gamma_i^{\sigma}(\mathbf{x})} \cdot \nabla_{\mathbf{x}} \log \left( s_i^{\sigma}(\mathbf{x}) \cdot p_i^{\sigma}(\mathbf{x}) \right) , \tag{28}$$

$$= \sum_{i \in \mathcal{I}} \gamma_i^{\sigma}(\mathbf{x}) \cdot \left( \nabla_{\mathbf{x}} \log s_i^{\sigma}(\mathbf{x}) + \nabla_{\mathbf{x}} \log p_i^{\sigma}(\mathbf{x}) \right) . \tag{29}$$

The skewed score function therefore is a weighted combination of the scores of the skewed mixture components, which are a combination of the original scores and log derivative of the skewing functions. As the skewing generally depends on the noise-level, we here introduce the noise depended entities $m_i^{\sigma}$, $s_i^{\sigma}(\mathbf{x})$, and $\gamma_i^{\sigma}(\mathbf{x})$. We discuss this in the next paragraph. We can now substitute $m_i^{\sigma}$ and $s_i^{\sigma}(\mathbf{x})$ to obtain

$$\gamma_i^{\sigma}(\mathbf{x}) = \frac{\alpha_i \cdot p_i^{\sigma}(\mathbf{x}) \cdot \mathbb{E}_{g \sim p_i^{\sigma}(g|\mathbf{x})} [w'(g)]}{\sum_{j \in \mathcal{I}} \alpha_j \cdot p_j^{\sigma}(\mathbf{x}) \cdot \mathbb{E}_{g \sim p_j^{\sigma}(g|\mathbf{x})} [w'(g)]} , \tag{30}$$

and

$$\nabla_{\mathbf{x}} \log s_i^{\sigma}(\mathbf{x}) = \nabla_{\mathbf{x}} \log \frac{\mathbb{E}_{g \sim p_i^{\sigma}(g|\mathbf{x})} [w'(g)]}{Z_i^{\sigma}} \tag{31}$$

$$= \nabla_{\mathbf{x}} \log \mathbb{E}_{g \sim p_i^{\sigma}(g|\mathbf{x})} [w'(g)] , \tag{32}$$

where we denote the noise-dependent normalization constants $Z_i^{\sigma}$ and the noise-depended distribution of the measure $g$ of the $i$th component $p_i^{\sigma}(g|\mathbf{x})$. The weight $\gamma_i^{\sigma}(\mathbf{x})$—which in the further derivation we call the posterior responsibility of component $i$—corresponds to the relative ability of the noised and skewed mixture components to explain the sample. It crucially depends on the expected value of the weight $w'(g)$. The score function of every mixture component is an unweighted combination of the score of the smoothed data distribution and the log-gradient of the expected weight for that component. We can further simplify

$$\nabla_{\mathbf{x}} \log \mathbb{E}_{g \sim p_i^{\sigma}(g|\mathbf{x})} [w'(g)] = \frac{\nabla_{\mathbf{x}} \mathbb{E}_{g \sim p_i^{\sigma}(g|\mathbf{x})} [w'(g)]}{\mathbb{E}_{g \sim p_i^{\sigma}(g|\mathbf{x})} [w'(g)]} , \tag{33}$$

$$= \frac{\nabla_{\mathbf{x}} \int p_i^{\sigma}(g|\mathbf{x}) \cdot w'(g) \, dg}{\mathbb{E}_{g \sim p_i^{\sigma}(g|\mathbf{x})} [w'(g)]} , \tag{34}$$

$$= \int_g \frac{p_i^{\sigma}(g|\mathbf{x}) \cdot w'(g)}{\mathbb{E}_{g \sim p_i^{\sigma}(g|\mathbf{x})} [w'(g)]} \cdot \nabla_{\mathbf{x}} \log p_i^{\sigma}(g|\mathbf{x}) dg , \tag{35}$$

$$= \mathbb{E}_{g \sim \tilde{p}_i^{\sigma}(g|\mathbf{x})} [\nabla_{\mathbf{x}} \log p_i^{\sigma}(g|\mathbf{x})] , \tag{36}$$

$$= \mathbb{E}_{g \sim \tilde{p}_i^{\sigma}(g|\mathbf{x})} [\nabla_{\mathbf{x}} \log p_i^{\sigma}(\mathbf{x}, g)] - \nabla_{\mathbf{x}} \log p_i^{\sigma}(\mathbf{x}) , \tag{37}$$

where the reweighted and noised conditional distribution of $g$ is $\tilde{p}_i^{\sigma}(g|\mathbf{x}) = \frac{p(g|\mathbf{x}) \cdot w'(g)}{\mathbb{E}_{g \sim p_i^{\sigma}(g|\mathbf{x})} [w'(g)]}$. Plugging this into Equation 29, we can see that the reweighted score

$$\nabla_{\mathbf{x}} \log \tilde{p}^{\sigma}(\mathbf{x}) = \sum_{i \in \mathcal{I}} \gamma_i^{\sigma}(\mathbf{x}) \cdot \mathbb{E}_{g \sim \tilde{p}_i^{\sigma}(g|\mathbf{x})} [\nabla_{\mathbf{x}} \log p_i^{\sigma}(\mathbf{x}, g)] . \tag{38}$$

is a weighted combination of the reweighted mixture component scores, where each component score is an expectation over the joint score given the reweighted conditional distribution of $g$, thus pointing into the direction of high data-density of the data-subspace favored by the weights $w'(g)$.

**Weak and strong posterior shift**    Note that in general, the active set after reweighting

$$\tilde{\mathcal{A}}(\mathbf{x}, \sigma) := \{i \mid \gamma_i^{\sigma}(\mathbf{x}) \geq \tilde{\varepsilon}(\mathbf{x}, \sigma)\} , \frac{\sum_{i \in \tilde{\mathcal{A}}} \alpha_i \cdot p_i^{\sigma}(\mathbf{x})}{\sum_{i \in \mathcal{I}} \alpha_i \cdot p_i^{\sigma}(\mathbf{x})} \geq 1 - \delta , \tag{39}$$

does not coincide with the active set before reweighting. However, for bounded expected weights of components $w'_{min}(\mathbf{x}, \sigma) \leq \mathbb{E}_{g \sim p_i^\sigma(g|\mathbf{x})}[w'(g)] \leq w'_{max}(\mathbf{x}, \sigma), \forall i \in \tilde{\mathcal{A}}(\mathbf{x}, \sigma)$, there exists boundaries on the relative difference between the prior and posterior responsibilities

$$\overline{\Delta}_{w'}(\mathbf{x}, \sigma) \leq \frac{w'_{max}(\mathbf{x}, \sigma)}{w'_{min}(\mathbf{x}, \sigma)}, \quad \underline{\Delta}_{w'}(\mathbf{x}, \sigma) \geq \frac{w'_{min}(\mathbf{x}, \sigma)}{w'_{max}(\mathbf{x}, \sigma)}, \tag{40}$$

such that $\underline{\Delta}_{w'}(\mathbf{x}, \sigma) \cdot \frac{\alpha_i p_i^\sigma(\mathbf{x})}{p^\sigma(\mathbf{x})} \leq \gamma_i^\sigma(\mathbf{x}) \leq \overline{\Delta}_{w'}(\mathbf{x}, \sigma) \cdot \frac{\alpha_i p_i^\sigma(\mathbf{x})}{p^\sigma(\mathbf{x})}$. We here want to point out two special cases of the posterior responsibilities:

*a) Weak posterior shift*: If all active noised mixture components locally approximately have the same expected weight, i.e., $\underline{\Delta}_{w'}(\mathbf{x}, \sigma) \approx \overline{\Delta}_{w'}(\mathbf{x}, \sigma) \approx 1$, the posterior responsibility

$$\gamma_i^\sigma(\mathbf{x}) = \frac{\alpha_i \cdot p_i^\sigma(\mathbf{x}) \cdot \mathbb{E}_{g \sim p_i^\sigma(g|\mathbf{x})}[w'(g)]}{\sum_{j \in \mathcal{I}} \alpha_j \cdot p_j^\sigma(\mathbf{x}) \cdot \mathbb{E}_{g \sim p_j^\sigma(g|\mathbf{x})}[w'(g)]} \approx \alpha_i \cdot \frac{p_i^\sigma(\mathbf{x})}{p^\sigma(\mathbf{x})}, \tag{41}$$

differs only mildly from the prior ones. In this case, the active components before and after reweighting are equivalent $\tilde{\mathcal{A}}(\mathbf{x}, \sigma) = \mathcal{A}(\mathbf{x}, \sigma)$.

*b) Strong posterior shift*: If a single component $j$ emits a significantly higher expected weight than others, the posterior responsibility and consequently the score will be dominated by this component: $\gamma_j^\sigma \geq 1 - \delta$, $\gamma_i^\sigma \approx 0$, $\forall i \neq j$ and therefore $\tilde{\mathcal{A}}(\mathbf{x}, \sigma) = \{j\} \neq \mathcal{A}(\mathbf{x}, \sigma)$. This extends to the case of multiple active components falling into a cluster $\tilde{\mathcal{A}}(\mathbf{x}, \sigma) \neq \mathcal{A}(\mathbf{x}, \sigma)$ that jointly exhibit significantly higher expected weights than other components such that

$$\sum_{j \in \tilde{\mathcal{A}}} \gamma_j^\sigma(\mathbf{x}) \geq 1 - \delta, \text{ and } \gamma_i^\sigma(\mathbf{x}) \approx 0, \forall i \notin \tilde{\mathcal{A}}(\mathbf{x}, \sigma). \tag{42}$$

**Smoothing of reweighting measures** Note that above we have introduced the noise-dependent mass factors $m_i^\sigma$ and skewing functions $s_i^\sigma(\mathbf{x})$ as the Gaussian smoothing of $p_i(\mathbf{x})$ also results in a smoothing of the expectations $\mathbb{E}_{g \sim p_i^\sigma(g|\mathbf{x})}[w'(g)]$. Formally, the expectation becomes a mixture over all potential true $\mathbf{x}_{\text{true}}$ that may have produced the noisy sample $\mathbf{x}$

$$p_i^\sigma(g|\mathbf{x}) = \frac{p_i^\sigma(\mathbf{x}, g)}{p_i^\sigma(\mathbf{x})} = \frac{\int p_i(\mathbf{x}_{\text{true}}) \cdot \mathcal{N}(\mathbf{x}|\mathbf{x}_{\text{true}}, \sigma) \cdot p_i(g|\mathbf{x}_{\text{true}}) d\mathbf{x}_{\text{true}}}{p_i^\sigma(\mathbf{x})}. \tag{43}$$

Importantly there is no term of the probability distribution $p_i(g|\mathbf{x})$ given the noisy sample $\mathbf{x}$, which is undefined for $\mathbf{x}$ outside the support of $p(\mathbf{x})$. We therefore get the smoothed expected weight of component $i$

$$\mathbb{E}_{g \sim p_i^\sigma(g|\mathbf{x})}[w'(g)] = \int_g \int_{\mathbf{x}_{\text{true}}} \frac{p_i(\mathbf{x}_{\text{true}}) \cdot \mathcal{N}(\mathbf{x}|\mathbf{x}_{\text{true}}, \sigma) \cdot p_i(g|\mathbf{x}_{\text{true}})}{p_i^\sigma(\mathbf{x})} \cdot w'(g) d\mathbf{x}_{\text{true}} dg, \tag{44}$$

$$= \frac{\int_{\mathbf{x}_{\text{true}}} p_i(\mathbf{x}_{\text{true}}) \cdot \mathcal{N}(\mathbf{x}|\mathbf{x}_{\text{true}}, \sigma) \cdot \mathbb{E}_{g \sim p_i(g|\mathbf{x}_{\text{true}})}[w'(g)] d\mathbf{x}_{\text{true}}}{p_i^\sigma(\mathbf{x})}, \tag{45}$$

$$= \mathbb{E}_{\mathbf{x}_{\text{true}} \sim p_i^\sigma(\mathbf{x}_{\text{true}}|\mathbf{x})} \mathbb{E}_{g \sim p_i(g|\mathbf{x}_{\text{true}})}[w'(g)], \tag{46}$$

where $p_i^\sigma(\mathbf{x}_{\text{true}}|\mathbf{x}) = \frac{p_i(\mathbf{x}_{\text{true}}) \cdot \mathcal{N}(\mathbf{x}|\mathbf{x}_{\text{true}}, \sigma)}{p_i^\sigma(\mathbf{x})}$ is the likelihood of $\mathbf{x}$ being produced by $\mathbf{x}_{\text{true}}$ in component $i$ under the noise-level $\sigma$. Smoothing the reweighting measure thus corresponds to averaging the unnoised conditional expected weight over likely latent true data that could have generated the noisy observation. Importantly this shows that the two component of the score in Equation 29 are smoothed equally strong so we cannot assume any component to vanish for high noise levels. We here want to point out three relevant cases:

*a) Weak smoothing*: For noise levels $\sigma$ that are significantly smaller than the distances between components, the Gaussian kernel concentrates around $\mathbf{x}_{\text{true}}$. As we assume the probabilistic modes to be well-separated, in this case only a single component $i$ is active both before and after reweighting $\tilde{\mathcal{A}}(\mathbf{x}, \sigma) = \mathcal{A}(\mathbf{x}, \sigma) = \{i\}$. *Weak smoothing* therefore results in the case of *weak posterior shift*, discussed above.

*b) Intermediate smoothing*: For larger $\sigma$, the Gaussian kernel in Equation 23 moderately smoothes $p^\sigma(\mathbf{x})$ such that multiple components are active. In this case, we assume that the smoothing over

measures $g$ from Equation 43 is moderate such that in general the expected measures of active modes may vary significantly if the active components emit strongly differing weight statistics such that $\underline{\Delta}_{w'}(\mathbf{x}, \sigma) \cdot \frac{\alpha_i p_i^\sigma(\mathbf{x})}{p^\sigma(\mathbf{x})} \leq \gamma_i^\sigma(\mathbf{x}) \leq \overline{\Delta}_{w'}(\mathbf{x}, \sigma) \cdot \frac{\alpha_i p_i^\sigma(\mathbf{x})}{p^\sigma(\mathbf{x})}$, where $\underline{\Delta}_{w'}(\mathbf{x}, \sigma) << 1$ and $\overline{\Delta}_{w'}(\mathbf{x}, \sigma) >> 1$. It is therefore generally possible that the active sets before and after reweighting are not equivalent $\tilde{\mathcal{A}}(\mathbf{x}, \sigma) \neq \mathcal{A}(\mathbf{x}, \sigma)$, resulting in *strong posterior shift*.

*c) Strong smoothing*: For very large $\sigma$, the Gaussian kernel from Equation 43 significantly blends together the return distributions across different probabilistic modes such that they differ only mildly and therefore $\underline{\Delta}_{w'}(\mathbf{x}, \sigma) \cdot \frac{\alpha_i p_i^\sigma(\mathbf{x})}{p^\sigma(\mathbf{x})} \leq \gamma_i^\sigma(\mathbf{x}) \leq \overline{\Delta}_{w'}(\mathbf{x}, \sigma) \cdot \frac{\alpha_i p_i^\sigma(\mathbf{x})}{p^\sigma(\mathbf{x})}$, where $\underline{\Delta}_{w'}(\mathbf{x}, \sigma) \approx 1$ and $\overline{\Delta}_{w'}(\mathbf{x}, \sigma) \approx 1$. In this case, the active set of components remains the same before and after reweighting $\tilde{\mathcal{A}}(\mathbf{x}, \sigma) = \mathcal{A}(\mathbf{x}, \sigma)$, resulting in *weak posterior shift*

Note that these three cases do not have clear boundaries but mix continuously. The distinction between the different regimes depends on the relation between the noise level $\sigma$, the expected weights induced by the statistics $p_i^\sigma(g|\mathbf{x})$, the internal geometry of active components, and the relative geometry of active components. The three cases therefore describe qualitatively different regimes that arise throughout the denoising process, rather than fixed ranges of the noise level. It is thus possible that throughout a concrete denoising process, we encounter *intermediate smoothing* and *strong posterior shift* but go back to *strong smoothing* and *weak posterior shift*, once a particular component vanishes from the active set. This can be understood as "decision points" where the diffused sample moves closer to a specific data subspace, which is determined by one or more probabilistic modes of the original distribution, making it unlikely for the sample to leave that subspace mode afterward. This is supported by our qualitative evaluations in Figure 4.

**Guiding the diffusion process by combining predictions**    In analogy to classifier guidance – that combines conditional and unconditional scores through $(1 - s) \cdot \nabla_\mathbf{x} \log p(\mathbf{x}|\emptyset) + s \cdot \nabla_\mathbf{x} \log p(\mathbf{x}|c)$ – we now propose to guide the sampling process using a combination of the original and the reweighted scores

$$\nabla_\mathbf{x} \log p^\sigma(\mathbf{x}) + s \cdot \underbrace{(\nabla_\mathbf{x} \log \tilde{p}^\sigma(\mathbf{x}) - \nabla_\mathbf{x} \log p^\sigma(\mathbf{x}))}_{:=\mathbf{d}(\mathbf{x}, \sigma)}, \tag{47}$$

where $\mathbf{d}(\mathbf{x}, \sigma)$ is the score mismatch of the skewed and the unskewed distribution and $s$ is the guide scale that controls the strength of the applied guidance. Empirically we know that in classifier-free guidance best results are often obtained for $s > 1$, issuing a notion of guiding the sample along a vector that points in some direction in data space (e.g., towards higher return in a reinforcement learning setting), rather than merely interpolating between conditional and unconditional predictions. Note that classifier-free guidance can be seen as a special case of our method, in that data samples that exactly match the conditioning value get a weight of 1 and all others a weight of 0. Using Equation 25 and Equation 29, we obtain

$$\mathbf{d}(\mathbf{x}, \sigma) = \nabla_\mathbf{x} \log \tilde{p}^\sigma(\mathbf{x}) - \nabla_\mathbf{x} \log p^\sigma(\mathbf{x}) \tag{48}$$

$$= \sum_{i \in \mathcal{I}} \gamma_i^\sigma(\mathbf{x}) \cdot \nabla_\mathbf{x} \log \left( s_i^\sigma(\mathbf{x}) \cdot p_i^\sigma(\mathbf{x}) \right) - \sum_{i \in \mathcal{I}} \alpha_i \cdot \frac{p_i^\sigma(\mathbf{x})}{p^\sigma(\mathbf{x})} \cdot \nabla_\mathbf{x} \log p_i^\sigma(\mathbf{x}) \tag{49}$$

$$= \sum_{i \in \mathcal{I}} \left( \gamma_i^\sigma(\mathbf{x}) - \alpha_i \cdot \frac{p_i^\sigma(\mathbf{x})}{p^\sigma(\mathbf{x})} \right) \cdot \nabla_\mathbf{x} \log p_i^\sigma(\mathbf{x}) + \gamma_i^\sigma(\mathbf{x}) \cdot \nabla_\mathbf{x} \log s_i^\sigma(\mathbf{x}). \tag{50}$$

The score mismatch thus consists of two components: The scores of the original data distribution, weighted by the difference in prior and posterior responsibilities and the log gradients of the skewing function, weighted by the posterior responsibilities. Note that, as both the prior and the posterior responsibilities independently sum to 1, these combinations of scores and skewing log-gradients are constrained by $0 \leq \sum_{i \in \mathcal{I}} \left| \gamma_i^\sigma(\mathbf{x}) - \alpha_i \cdot \frac{p_i^\sigma(\mathbf{x})}{p^\sigma(\mathbf{x})} \right| \leq 2$, and $\sum_{i \in \mathcal{I}} \gamma_i^\sigma(\mathbf{x}) = 1$.

For the case of *weak posterior shift*, discussed above, the prior and posterior responsibilities approximately cancel out and the score mismatch simplifies to

$$\mathbf{d}^{\text{intra}}(\mathbf{x}, \sigma) \approx \sum_{i \in \mathcal{I}} \alpha_i \cdot \frac{p_i^\sigma(\mathbf{x})}{p^\sigma(\mathbf{x})} \cdot \nabla_\mathbf{x} \log s_i^\sigma(\mathbf{x}). \tag{51}$$

Guiding the diffusion process through different reweights, in this case—i.e., negligible differences of the expected weights of active probabilistic modes of the data distribution, or strong smoothing due to high $\sigma$—therefore performs gradient ascend on the log skewing function. Since in this regime the mismatch is dominated by the internal reweighting within components, we refer to this as *intra-mode guidance*. Vice versa, in the case of *strong posterior shift*—characterized by substantial differences between prior and posterior responsibilities—Equation 50 in general does not simplify. Here we assume that the prior and posterior responsibilities differ significantly and therefore $\sum_{i \in \mathcal{I}} \left| \gamma_i^\sigma(\mathbf{x}) - \alpha_i \cdot \frac{p_i^\sigma(\mathbf{x})}{p^\sigma(\mathbf{x})} \right| >> 0$. As in this regime, the structure of the mixture crucially influences or even dominates the score mismatch (in the case of return statistics that vary only mildly within components), we refer to this as *inter-mode guidance*.

We can now re-introduce the exponential weighting scheme derived in Equation 21 to obtain $\nabla_{\mathbf{x}} \log s_i^\sigma(\mathbf{x}) = \nabla_{\mathbf{x}} \log \mathbb{E}_{g \sim p_i^\sigma(g|\mathbf{x})}[g] = \nabla_{\mathbf{x}} \log \mathbb{E}_{g \sim p_i^\sigma(g|\mathbf{x})}[\exp(\beta \cdot V(\mathbf{x}))] = \beta \cdot \nabla_{\mathbf{x}} V_i^\sigma(\mathbf{x})$ and therefore

$$\mathbf{d}(\mathbf{x}, \sigma) = \sum_{i \in \mathcal{I}} \left( \gamma_i^\sigma - \alpha_i \cdot \frac{p_i^\sigma(\mathbf{x})}{p^\sigma(\mathbf{x})} \right) \cdot \nabla_{\mathbf{x}} \log p_i^\sigma(\mathbf{x}) + \beta \cdot \gamma_i^\sigma(\mathbf{x}) \cdot \nabla_{\mathbf{x}} V_i^\sigma(\mathbf{x}). \tag{52}$$

For the case of *weak posterior shift*, we therefore perform gradient ascend on the value function $\mathbf{d}^{\text{intra}}(\mathbf{x}, \sigma) \approx \beta \cdot \sum_{i \in \mathcal{I}} \alpha_i \cdot \frac{p_i^\sigma(\mathbf{x})}{p^\sigma(\mathbf{x})} \cdot \nabla_{\mathbf{x}} V_i^\sigma(\mathbf{x})$, therefore driving the sampling distribution towards higher return regions within the data submanifold. Rewriting the combined score $\nabla_{\mathbf{x}} \log p^\sigma(\mathbf{x}) + s \cdot \mathbf{d}(\mathbf{x}, \sigma) = \sum_{i \in \mathcal{I}} \alpha_i \cdot \frac{p_i^\sigma(\mathbf{x})}{p^\sigma(\mathbf{x})} \cdot (\nabla_{\mathbf{x}} \log p_i^\sigma(\mathbf{x}) + s \cdot \beta \cdot \nabla_{\mathbf{x}} \log \nabla_{\mathbf{x}} V_i^\sigma(\mathbf{x}))$, we can see that, in the case of *intra-mode guidance* the scaling factor $s$ acts as a multiplicator for the temperature $\beta$, which circumvents the limitations of finite batch sizes and limited data and capacity discussed in Appendix A.1. Our experiments in Figure 8 confirm that, for a reasonably chosen $s$, downstream performance on the medium-replay and medium datasets is maximized, which was not possible through reweighting alone (see Appendix A.1).

### A.3 ADAPTIVE DIFFUSION GUIDANCE

As discussed above, we differentiate between two qualitatively different guidance regimes throughout the diffusion process: *intra-* and *inter-mode guidance*. We will now discuss how these different pathologies give rise to the strong hyperparameter brittleness of classifier-free guidance. Then we propose a methodology to adapt the guide scale to the prevalent pathology.

**Under- and overguidance** As introduced in Appendix A.1, sufficiently strong reweighting typically biases the sampling distribution toward high-performing subspaces when components have distinct return statistics. Our analysis in Appendix A.2 shows that in *strong posterior shift* scenarios, following the score of the reweighted distribution effectively guides the diffusion toward these high-performing components. However, employing the combined prediction in Equation 47 with high guide scales may cause *overguidance*, as the score mismatch does not represent a meaningful direction in data space, thus pushing the samples excessively away from the noised marginal distribution and degrading performance. In contrast, in *weak posterior shift* scenarios (Equation 51), small guide scales risk *underguidance* by insufficiently leveraging the value-gradient information, thus failing to guide the sample towards high-performing long tails of the sampling distribution. In Figure 8 we empirically observe this trade-off on some datasets (e.g., hopper-medium), where an appropriately chosen guide scale prevents *underguidance* and improves downstream performance. Excessive guidance, however, leads to a sharp performance decline in all scenarios, attributed to *overguidance*. Note that return maximization in diffusion planners using classifier-free guidance (e.g., (Ajay et al., 2022)) can be seen as a special case of our reweighting-based methodology. The pathologies of *over-* and *underguidance* therefore also exist in these algorithms, giving rise to their high dependence on task-specific hyperparameter settings.

**Distinguishing guidance modes** During *intra-mode guidance* we want to strongly leverage the gradients of the value function through high guide scales $s > 1$ to avoid *underguidance*. During *inter-mode guidance*, we simply want to follow the score of the skewed distribution (i.e., $s \approx 1$) to avoid *overguidance*. Importantly, these regimes do not occur in a binary fashion; instead, the influence of intra- and inter-mode guidance continuously shifts throughout the diffusion process. We now propose a methodology to adapt the guide scale to the prevalent pathology.

Suppose we employ two different reweighting schemes using weighting functions $w'_+(g)$ and $w'_-(g)$ that satisfy $\nabla_{\mathbf{x}} \log \mathbb{E}_{g \sim p_i^\sigma(g|\mathbf{x})} \left[ w'_+(g) \right] = -\nabla_{\mathbf{x}} \log \mathbb{E}_{g \sim p_i^\sigma(g|\mathbf{x})} \left[ w'_-(g) \right]$ and therefore the skewing functions $\nabla_{\mathbf{x}} \log s_{+,i}^\sigma(\mathbf{x}) = -\nabla_{\mathbf{x}} \log s_{-,i}^\sigma(\mathbf{x})$. We call these two reweightings conjugate as they skew the sampling distribution in exactly opposite directions. Reweighting the original data distribution with the two different weights individually yields two skewed distributions $\tilde{p}_+(\mathbf{x})$ and $\tilde{p}_-(\mathbf{x})$. Given the noised scores of a skewed and the unskewed distribution we obtain the two score mismatches $\mathbf{d}_-(\mathbf{x}, \sigma)$ and $\mathbf{d}_+(\mathbf{x}, \sigma)$. In the case of *intra-mode guidance*, the score functions of reweighted distributions are dominated by the gradients of the skewing functions

$$\mathbf{d}_+^{\text{intra}}(\mathbf{x}, \sigma) \approx -\mathbf{d}_+^{\text{intra}}(\mathbf{x}, \sigma) \approx \sum_{i \in \mathcal{I}} \alpha_i \cdot \frac{p_i^\sigma(\mathbf{x})}{p^\sigma(\mathbf{x})} \cdot \nabla_{\mathbf{x}} \log s_{+,i}^\sigma(\mathbf{x}) \,. \tag{53}$$

As the conjugate reweights skew the sampling distribution in exactly opposite ways, the two score mismatches represent antiparallel vectors in data-space. For the case of *inter-mode guidance* we obtain

$$\mathbf{d}_-^{\text{inter}}(\mathbf{x}, \sigma) \approx \sum_{i \in \mathcal{I}} \left( \gamma_{-,i}(\mathbf{x}) - \alpha_i \cdot \frac{p_i^\sigma(\mathbf{x})}{p_i^\sigma(\mathbf{x})} \right) \cdot \nabla_{\mathbf{x}} \log p_i^\sigma(\mathbf{x}) - \gamma_{-,i}(\mathbf{x}) \cdot \nabla_{\mathbf{x}} \log s_{+,i}^\sigma(\mathbf{x}) \,, \tag{54}$$

$$\mathbf{d}_+^{\text{inter}}(\mathbf{x}, \sigma) \approx \sum_{i \in \mathcal{I}} \left( \gamma_{+,i}(\mathbf{x}) - \alpha_i \cdot \frac{p_i^\sigma(\mathbf{x})}{p_i^\sigma(\mathbf{x})} \right) \cdot \nabla_{\mathbf{x}} \log p_i^\sigma(\mathbf{x}) + \gamma_{+,i}(\mathbf{x}) \cdot \nabla_{\mathbf{x}} \log s_{+,i}^\sigma(\mathbf{x}) \,, \tag{55}$$

where $\sum_{i \in \mathcal{I}} \left| \gamma_{+,i}^\sigma(\mathbf{x}) - \alpha_i \cdot \frac{p_i^\sigma(\mathbf{x})}{p^\sigma(\mathbf{x})} \right| >> 0$ and $\sum_{i \in \mathcal{I}} \left| \gamma_{-,i}^\sigma(\mathbf{x}) - \alpha_i \cdot \frac{p_i^\sigma(\mathbf{x})}{p^\sigma(\mathbf{x})} \right| >> 0$. Since the reweights are conjugate, in the case of *strong posterior shift*, the two posterior active sets $\tilde{\mathcal{A}}_+(\mathbf{x}, \sigma)$ and $\tilde{\mathcal{A}}_-(\mathbf{x}, \sigma)$ are effectively disjoint. This results in the two score mismatches being strictly different combinations of the gradient directions $\nabla_{\mathbf{x}} \log p_i^\sigma(\mathbf{x})$ and $\nabla_{\mathbf{x}} \log s_{+,i}^\sigma(\mathbf{x})$. As $\forall i \neq j : \nabla_{\mathbf{x}} \log p_i^\sigma(\mathbf{x}) \neq \nabla_{\mathbf{x}} \log p_j^\sigma(\mathbf{x})$ and $\nabla_{\mathbf{x}} \log s_{+,i}^\sigma(\mathbf{x}) \neq \nabla_{\mathbf{x}} \log s_{+,j}^\sigma(\mathbf{x})$, for almost all $\mathbf{x}, \sigma$, the resulting score mismatches represent weakly correlated directions in the data-space. Note that in high dimensions, randomly chosen directions tend to be close to orthogonal.

We now propose an adaptive guide scale that exploits the property of the two score mismatches under complementary reweights being close to collinear in the case of *intra-mode guidance* and close to orthogonal in the case of *inter-mode guidance* via their cosine distance

$$s(\mathbf{x}, \sigma) := 1 + s' \cdot \hat{s}(c, \sigma) \,, \text{ with} \tag{56}$$

$$\hat{s}(\mathbf{x}, \sigma) := \big( -\cos\theta(\mathbf{x}, \sigma) \big)^p \,, \text{ and} \tag{57}$$

$$\cos\theta(\mathbf{x}, \sigma) := \frac{\langle \mathbf{d}_+(\mathbf{x}, \sigma), \mathbf{d}_-(\mathbf{x}, \sigma) \rangle}{\|\mathbf{d}_+(\mathbf{x}, \sigma)\| \cdot \|\mathbf{d}_-(\mathbf{x}, \sigma)\|} \,, \tag{58}$$

where $s'$ is a static value and $p$ controls the strength of guidance attenuation. Extending the weighting scheme derived in Equation 21, we can utilize the conjugate pairs of reweights $w'_+(g) = \exp(\beta \cdot V(\mathbf{x}))$ and $w'_-(g) = \exp(-\beta \cdot V(\mathbf{x}))$ to identify the prevalent guidance mode. In Figure 5, we show that this adaptive guidance scheme significantly increases the range of optimal hyperparameter w.r.t. return-conditioned CFG, resulting in the option of default settings that jointly optimize performance across tasks.

## A.4 PRACTICAL CONSIDERATIONS

Above, we have derived the theoretical foundations and behavior of reweighting-induced skewing diffusion models, including their effect on the score function and implications for guidance strategies. We now discuss some practical considerations for our implementation.

**Normalized utility function** We so far have not posed any requirements on the measure $g$. However, as we deal with limited data and model capacity, in practice the scale of the measure has to match the reweighting scheme such that weights are not unbounded. Typically this is achieved by normalizing $g$ such that the weights $w'_{\min} \leq w'(\mathbf{x}) \leq w'_{\max}$, effectively preventing numerical instabilities. However, in order to achieve relevant reweighting even in datasets with few extreme values of the measure $g$, we here normalize to zero mean and unit variance to obtain $g'_i = \frac{g_i - \mu_g}{\sigma_g}$, where $\mu_g$ and $\sigma_g$ are the

empirical mean and standard derivation of $g$ across the dataset. Following our reweighting scheme from Equation 21, we then compute the Boltzmann weights as $w'(g_i) = \exp(\beta \cdot g_i')$. This preserves sensitivity to both high- and low-utility samples without allowing rare outliers to dominate the loss signals.

**Finite batch size**    In Equation 11, we have shown a batch estimator for the reweighted loss function that includes a batch estimate of the weight $w(g)$. However, in further derivations we have not considered the influence of the batch size. In practice, batch sizes are limited, which poses an upper bound on the the amount of reweighting applied as $0 < w(g) < B$, where $B$ is the batch size, effectively dampening the effect on the reweighted sampling distribution. The sampling distribution after reweighting is therefore limited by $0 < \tilde{p}(\mathbf{x}) < p(\mathbf{x}) \cdot B$. As a consequence, reweighting alone may not sufficiently skew the sampling distribution to robustly sample from subspaces that have limited support under the original data distribution but are strongly favored by the unnormalized weights $w'(g)$. This limitation is particularly pronounced for mixture distributions where high-performing components constitute a small fraction of the training data. Recall, that this led us to introduce the classifier-free guidance-style combination of skewed and unskewed score functions in Appendix A.2. In addition, the batch-dependent normalization constant $\sum_{j=0}^{B-1} w'(g_i)$ exhibits high variance when $B$ is small. To avoid these theoretic batch-dependent limitations, we tried out skipping batch normalization, instead setting $Z = \mathbb{E}_{\mathbf{x} \sim p(\mathbf{x})} [\exp(\beta \cdot V(\mathbf{x}))] \approx \exp\left(\frac{\beta^2}{2}\right)$ , which is approximately valid as we normalize our returns to zero mean and unit variance (with mismatches for non-symmetric distributions). However, employing batch normalization shows to significantly stabilize training convergence. We therefore choose to select large batch sizes (256 in our experiments) to minimize these limitations but acknowledge that this may degrade performance on datasets with few high-performing samples.

**Value estimation**    In our derivations in Appendix A.2, we have assumed access to the true value function $V(\mathbf{x}) = \mathbb{E}_{g \sim p(g|\mathbf{x})}[g]$. However, in practice, we cannot directly access this and instead have to estimate it using a parametrized model $\hat{V}_\phi$ by minimizing $\text{MSE}(V(\mathbf{x}), \hat{V}_\phi(\mathbf{x}))$. Experimentally (see Appendix B.5) we obtained comparable performance when skipping the value-estimation, instead reweighting by the discounted cumulative future return $g_i$ of the respective sample $\mathbf{x}_i$ directly $w'(g_i) = \exp(\beta \cdot g_i)$. The resulting sampling distribution becomes

$$\tilde{p}(\mathbf{x}) \propto p(\mathbf{x}) \cdot \mathbb{E}_{g \sim p(g|\mathbf{x})} [\exp(\beta \cdot g)] \tag{59}$$

$$= p(\mathbf{x}) \cdot \exp(\beta \cdot V(\mathbf{x})) \cdot \mathbb{E}_{g \sim p(g|\mathbf{x})} [\exp(\beta \cdot (g - V(\mathbf{x})))] . \tag{60}$$

Beyond biasing towards regions of high expected return $V(\mathbf{x})$, this formulation additionally favors behaviors with high return variance due to the convexity of $\exp(\cdot)$. In stochastic environments, this induces risk-seeking behavior, as behaviors with volatile, occasional high returns may be favored over robust but intermediate returns. On the other hand, accentuating the tails of the return distribution mitigates the dependence of the value function on the data generating policy. As $V(\mathbf{x})$ represents the expected value of the future discounted return when executing behavior $\mathbf{x}$ and following the data generating policy after, the bias towards higher values helps to approximate the optimal value function $V^*(\mathbf{x})$. Jenssen's inequality states that $\log \mathbb{E}_{g \sim p_i^\sigma(g|\mathbf{x})} [w'(g)] \geq \mathbb{E}_{g \sim p_i^\sigma(g|\mathbf{x})} [\log w'(g)]$, where equality holds for $\text{Var}(g) \to 0$. The collinearity assumption of Equation 53 therefore no longer holds if $p_i(g|\mathbf{x})$ exhibit significant variance. In theory, the performance of our algorithm should thus be affected by high variance. We assume that modeling system behaviors $\mathbf{x}$ as temporally extended plans of horizon $h$ partially mitigates the effect of broad return distributions as the variance of the conditional distributions $p_i(g|\mathbf{x})$ is low for long horizons $h$ if the environment itself exhibits weak stochasticity.

**Substituting the score to match EDM formulation**    Up till now, our derivations were in terms of the score functions $\nabla_\mathbf{x} \log p^\sigma(\mathbf{x})$ and $\nabla_\mathbf{x} \log \tilde{p}^\sigma(\mathbf{x})$. Since in this paper we utilize EDM as our diffusion formulation, in practice, we approximate the ideal denoisers $D(\mathbf{x}, \sigma)$ and $\tilde{D}(\mathbf{x}, \sigma)$, where the latter denotes the ideal denoiser for the reweighted distribution $\tilde{p}(\mathbf{x})$ for the noise level $\sigma$. We here want to note that we can define an analogous argument based on the ideal denoisers as

$$\nabla_\mathbf{x} \log p^\sigma(\mathbf{x}) = \frac{D(\mathbf{x}; \sigma) - \mathbf{x}}{\sigma^2} . \tag{61}$$

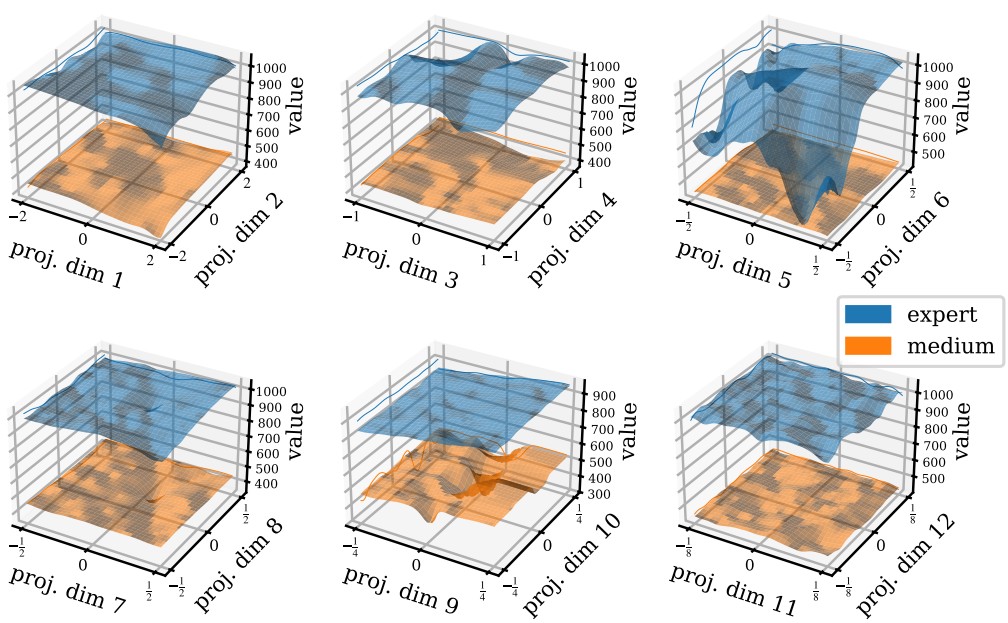

Figure 6: Value surfaces on the medium and expert trajectory manifold in hopper-medium-expert-v2 under six random 2D projections of state-only trajectories (horizon $= 32$, $\gamma = 0.997$), shown at multiple resolutions. Within each mode, the projected value surface is relatively smooth, whereas it differs sharply across modes. This intra-mode smoothness supports aggressive scaling of guidance vectors along on-manifold return gradients. By contrast, inter-mode discontinuities require careful guidance scaling during sampling to avoid leaving the marginal distribution.

We can therefore equivalently define the denoiser mismatch as

$$\mathbf{d}^{\text{denoiser}}(\mathbf{x}, \sigma) := \tilde{D}(\mathbf{x}; \sigma) - D(\mathbf{x}; \sigma) = \sigma^2 \cdot \mathbf{d}^{\text{score}}(\mathbf{x}, \sigma). \tag{62}$$

As cosine distances do not depend on scale, our adaptive guide scale $s(\mathbf{x}, \sigma)$ is unaffected by this.

# B   EXTENSIVE QUALITATIVE AND QUANTITATIVE ANALYSIS ON MIXED QUALITY DATASETS

This appendix extends the qualitative and quantitative evaluations from Section 3 of the TGDP framework in D4RL locomotion environments. It progressively investigates and motivates the design decisions and parameter settings of our final algorithm, alongside our derivations in Appendix A. We illustrate that return manifolds are smooth within behavioral modes and exhibit pronounced discontinuities between modes in Appendix B.1. We then ablate the individual parts of TGDP by (a) training with the temperature-based reweighted loss function and sampling for different temperatures $\beta$ in Appendix B.2 and (b) combining unbiased and reweighted scores in a classifier-free guidance fashion for different guide scales $s$ without adaption in Appendix B.3. We then illustrate how temperature-conditioned score functions relate to gradient ascent on the value function in Appendix B.4. Finally, we empirically show that we can directly reweight samples by sample returns, bypassing value function estimation in Appendix B.5.

## B.1   VALUE SURFACES WITHIN AND BETWEEN PERFORMANCE MODES

In Section 3, we hypothesize that each performance mode (e.g., medium, expert) in the dataset defines a smooth manifold in trajectory space, on which the distribution of discounted future return (and therefore also the value function) varies continuously with trajectory perturbations. Figure 6 illustrates this hypothesis using the 'hopper-medium-expert-v2' dataset. We project the value surfaces of the medium and expert modes onto six distinct 2D subspaces of the trajectory data (state sequences

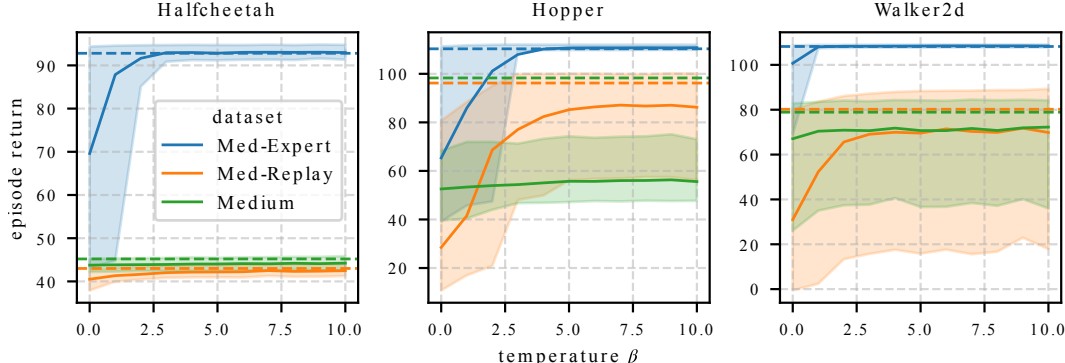

Figure 7: Episode returns in D4RL locomotion tasks under temperature-based loss reweighting averaged over three trainings × 300 evaluations; shaded areas show 90% return quantiles; dotted lines, final performance as reported in Section 4; higher is better. Higher temperatures bias sampling towards better-performing behaviors, with dataset-specific effects: near-optimal returns on *medium–expert* due to sharp inter-mode return gaps; moderate gains for *medium-replay* from upweighting well-performing behaviors; minimal improvement on *medium* due to narrow, unimodal return distributions.

over a horizon of 32 steps, discounted with $\gamma = 0.997$) and show patches of varying resolution. Note that this is a very coarse illustration as we marginalize over most dimensions of the trajectory space. Within each mode, the projected return landscape varies gradually along the projection directions, whereas it differs sharply between behavioral modes. This structure enables continuous optimization paths within a mode towards higher returns, by estimating and following the value gradients of the trajectory manifold. The smoothness within each manifold is critical for classifier-free guidance (and therefore also TGDP), which combines conditional (i.e., return-guided) and unconditional predictions via the guide scale $s$ to interpolate between, or even extrapolate beyond denoising targets that remain on the same manifold. In contrast, inter-mode paths are typically discontinuous because the corresponding trajectory manifolds are often disjoint or separated by low-density regions. When modes emit markedly different return statistics, reweighting the loss function induces strong shifts between unconditional and conditional denoising targets (see Appendix A.2). Extrapolating denoising targets across modes can therefore divert sampling away from the noised marginal since vectors that connect disjoint manifolds do not represent meaningful directions in trajectory space. This suggests distinct strategies: interpolation when steering toward a higher-performing mode manifold; and extrapolation when exploiting the internal return gradients within one mode.

## B.2 TEMPERATURE-BASED REWEIGHTING OF SAMPLING DISTRIBUTIONS

In Section 3 we describe a temperature-based loss reweighting methodology in EDM training to enable sampling from a shifted distribution during inference. In this appendix we empirically explore to which extent temperature-based training alone can yield effective conditioning on high return by simply following the temperature-conditional denoising targets $D_\theta(\mathbf{x}_t, \sigma, \beta)$. Figure 7 shows mean episode return over three trainings × 300 evaluations for the D4RL locomotion environments when sampling from the reweighted distribution for different temperatures $\beta$. Shaded areas represent 90% quantiles of the return distribution; dotted lines show final performance as reported in Section 4; higher is better. Zero-temperature reweighting yields unbiased samples reflecting the original plan distribution, resulting in a broad distribution of episode returns. Increasing the temperature progressively biases the sampling towards high-performing subspaces of the data distribution. Different datasets exhibit qualitatively distinct responses that reflect their manifold structure and return distributions. For the *medium-expert* datasets, conditioning on sufficiently high temperature achieves near-optimal performance. Our analysis in Appendix A.1 indicates that the steep value function differences between performance modes enable reliable sampling of the expert mode through reweighting alone. In contrast, *medium-replay* datasets show poor performance for weak reweighting as occasional sampling of behaviors from the initial experiences of data-collection can lead to catastrophic failures. At higher temperatures, probability mass shifts away from low-return behaviors but does not concentrate tightly enough to reach peak performance consistently. Finally, *medium* datasets show only

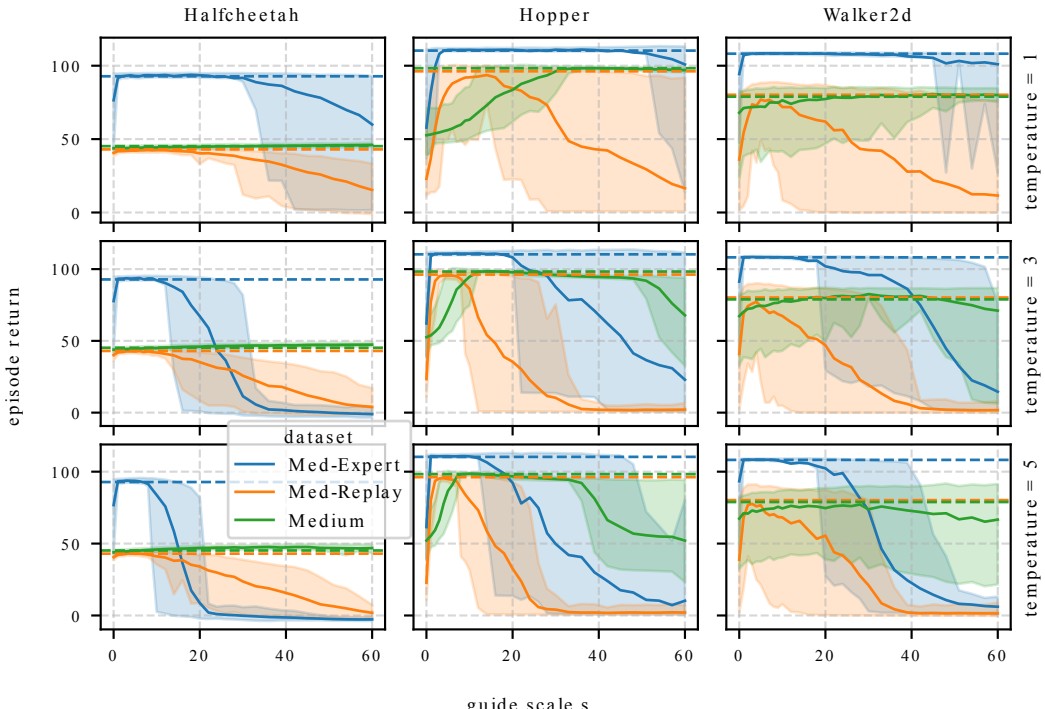

Figure 8: Episode returns in D4RL locomotion tasks as functions of guide scale $s$ and reweighting temperature $\beta$ averaged over three trainings × 300 evaluations; shaded areas show 90% return quantiles; dotted lines, final performance of the full algorithm for the respective dataset; higher is better. Guide scale $s$ blends unconditional and conditional denoising targets. Optimal guide scale depends on dataset and temperature: *medium-expert* and *medium-replay* datasets require moderate guidance, with performance dropping when encountering overguidance; *medium* datasets require high scales to exploit value gradients. Static guidance scaling cannot optimize performance across all tasks, necessitating dataset-specific tuning or adaptive guidance.

modest gains across temperatures. Theoretical analysis in Appendix A.2 indicates that, with a single behavioral mode and a narrow return distribution, reweighting alone cannot shift probability mass sufficiently within the mode to reliably enhance sampling of the highest-return behaviors.

### B.3 GUIDING THE DIFFUSION PROCESS VIA DIFFERENT TEMPERATURES

As empirically shown above, reweighting loss functions alone does not yield a reliable method to sample high-performing behaviors. In Section 3 we propose to blend unbiased and reweighted predictions through a guide scale $s$ to push sampled behaviors towards high performing long tails of the data distribution. In Appendix A.2 we analyze how this exploits value gradients of the trajectory manifold, present in the reweighted denoising targets. Figure 8 shows episode returns, averaged over three trainings × 300 evaluations, for the D4RL locomotion environments and for different guide scales $s$ and reweighting temperatures $\beta$. Shaded areas represent 90% quantiles; dotted lines, final performance as reported in Section 4; higher is better. For every dataset and temperature $\beta$ there exists a range of the guide scales $s$ that yields maximal episode returns. For lower temperatures, this range is generally higher, supporting our analysis in Appendix A.2 that the guide scale acts as an amplifier of the temperature during *intra-mode guidance*. Unfortunately, the ranges of suitable guidance do not overlap between different datasets for any given temperature. In particular, the medium-expert and medium-replay datasets seem to profit from moderate guide scales as performance drops steeply for high values, supporting our analysis of excessive guidance leading to *overguidance* as samples are pushed away from the noised marginals. Medium datasets on the other hand seem to profit from high guide scales to fully exploit value gradient information present in the reweighted score. Static guide scales thus cannot yield optimal performance across datasets, explaining the reliance on per-task optimization of classifier-free guidance based diffusion planners.

Table 3: Comparison of loss reweighting schemes on D4RL locomotion: mean ± standard deviation of episode returns over 3 trainings × 300 evaluations for (i) value-based reweighting and (ii) discounted-return reweighting, using default hyperparameters from Section 4. Results show comparable performance across tasks, supporting the practicality of bypassing explicit value estimation.

| | Halfcheetah | | | Hopper | | | Walker2D | | | |
| Reweighting | Med-Expert | Med-Replay | Medium | Med-Expert | Med-Replay | Medium | Med-Expert | Med-Replay | Medium | **Average** |
|---|---|---|---|---|---|---|---|---|---|---|
| sample return | 92.8 ± 4.6 | 43.0 ± 2.6 | 45.2 ± 0.9 | 110.4 ± 3.2 | 96.2 ± 1.8 | 98.3 ± 0.5 | 108.2 ± 0.7 | 80.2 ± 15.2 | 78.9 ± 11.5 | 83.7 ± 4.6 |
| value | 92.8 ± 5.4 | 42.6 ± 3.0 | 45.4 ± 1.1 | 110.6 ± 1.3 | 95.4 ± 6.2 | 97.8 ± 0.7 | 108.8 ± 0.4 | 76.4 ± 19.0 | 78.0 ± 12.2 | 83.1 ± 5.5 |

### B.4 TEMPERATURE GUIDANCE AS GRADIENT ASCENT ON THE VALUE SURFACE

In Section 3 we hypothesize that conditioning on high temperature incrementally aligns the diffusion target with the gradient of the value surface, while low-temperature conditioning induces movement counter to this gradient. This behavior is illustrated in Figure 9, which plots the mean and standard deviation of the change in predicted return of a single diffusion step from moving in the direction of a high temperature or low temperature-conditioned denoising target relative to unbiased sampling (zero temperature) in the hopper-medium dataset (single behavioral mode). Despite substantial noise, high-temperature conditioning shifts diffusion targets along the return gradient of the data manifold, whereas low-temperature conditioning opposes it. Consequently, displacement vectors between diffusion targets (i.e., score mismatches, see A.2) under different temperature conditions serve as noisy proxies for local value gradients within a behavioral mode. Smooth reward surfaces inside a mode (see Figure 6) ensure that local trajectory perturbations correlate predictably with return changes, enabling stable gradient-based ascent on the value surface during intra-mode guidance.

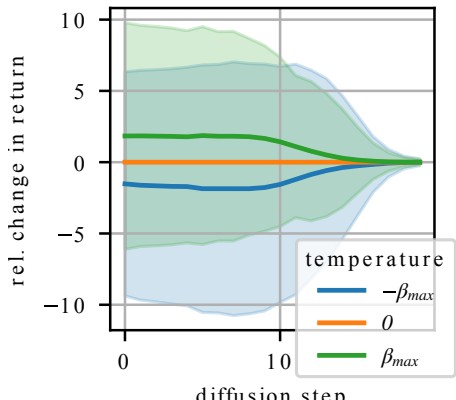

Figure 9: Mean and standard deviation of the change in predicted return of a single diffusion step when moving in the direction of a (i) high temperature or (ii) low temperature conditioned denoising target in the hopper-medium dataset. Conditioning on temperature incrementally aligns diffusion targets with value gradients of the data manifold, illustrating intra-mode guidance as gradient ascent on the value surface.

### B.5 REWEIGHTING WITH SAMPLE RETURNS

In Appendix A.4, we propose reweighting the loss by the discounted future return of data samples rather than by a learned value function. Table 3 compares mean and standard deviation of episode returns from 3 trainings × 300 evaluations for the D4RL locomotion benchmark for reweighting with a learned value function versus reweighting directly with the discounted return (using the default set of hyperparameters derived in Section 4). Performance is comparable across all tasks, supporting the practicality of bypassing explicit value estimation.

## C ADDITIONAL EXPERIMENTS AND ABLATIONS

This appendix extends the quantitative evaluation in Section 4, providing additional evaluations for Temperature-Guided Diffusion Planning (TGDP) across different benchmarks. We ablate the individual components of the final TGDP algorithm to showcase their individual contributions and motivate the full algorithm in Appendix C.1. We show runtime comparisons of advanced diffusion planning algorithms, outlining tradeoffs between decision frequencies and task performance in Appendix C.2. Finally, we provide additional evaluations on complex manipulation benchmarks and identify potential pitfalls of the TGDP framework in Appendix C.3.

### C.1 ABLATION OF TGDP COMPONENTS

In Table 4, we present an ablation study across three benchmark domains: locomotion, maze2d, and kitchen, evaluating the contributions of individual components within our TGDP framework. We

Table 4: Ablation of TGDP components. (tmp-cnd): Temperature conditioning; (guide): Guided sampling via combining conditioned targets; (adpt-sc): Adaptive guide scale based on collinearity of targets. We show episode return mean ± standard deviation of three trainings × 300 evaluations; higher is better; **best results** are in bold. (a) Unbiased sampling yields suboptimal performance, (b) reweighting alone provides inconsistent gains, (c) fixed-scale guidance can degrade results due to overguidance, and (d) the full adaptive TGDP consistently improves performance for default hyperparameter values, validating its design choices.

| incl. components
excl. components | (a)
—
(tmp-cnd), (guide), (adpt-sc) | (b)
(tmp-cnd)
(guide), (adpt-sc) | (c)
(tmp-cnd), (guide)
(adpt-sc) | (d)
(tmp-cnd), (guide), (adpt-sc)
— |
|---|---|---|---|---|
| **Dataset** / **Environment** | | | | |
| **Halfcheetah** Med-Expert | $79.6 \pm 21.2$ | $\mathbf{92.9 \pm 4.6}$ | $83.5 \pm 20.1$ | $92.8 \pm 4.6$ |
| Med-Replay | $37.6 \pm 8.7$ | $41.8 \pm 4.8$ | $38.9 \pm 7.4$ | $\mathbf{43.0 \pm 2.6}$ |
| Medium | $44.1 \pm 1.3$ | $44.3 \pm 1.1$ | $\mathbf{46.2 \pm 1.0}$ | $45.2 \pm 0.9$ |
| **Hopper** Med-Expert | $51.7 \pm 19.7$ | $104.9 \pm 18.3$ | $102.9 \pm 21.6$ | $\mathbf{110.4 \pm 3.2}$ |
| Med-Replay | $25.8 \pm 15.5$ | $81.3 \pm 16.5$ | $58.0 \pm 36.8$ | $\mathbf{96.2 \pm 1.8}$ |
| Medium | $53.6 \pm 9.8$ | $56.4 \pm 9.2$ | $95.7 \pm 1.2$ | $\mathbf{98.3 \pm 0.5}$ |
| **Walker2D** Med-Expert | $99.2 \pm 17.7$ | $108.1 \pm 2.9$ | $106.2 \pm 14.8$ | $\mathbf{108.2 \pm 0.7}$ |
| Med-Replay | $29.9 \pm 33.2$ | $64.5 \pm 23.3$ | $51.5 \pm 30.9$ | $\mathbf{80.2 \pm 15.2}$ |
| Medium | $67.5 \pm 16.9$ | $69.8 \pm 15.8$ | $73.8 \pm 16.3$ | $\mathbf{78.9 \pm 11.5}$ |
| **Average** | $54.3 \pm 29.0$ | $73.8 \pm 26.7$ | $73.0 \pm 31.5$ | $\mathbf{83.7 \pm 24.4}$ |
| **Maze2D** Umaze | $93.6 \pm 62.0$ | $129.5 \pm 45.4$ | $\mathbf{140.4 \pm 40.5}$ | $137.0 \pm 42.7$ |
| Medium | $108.3 \pm 60.2$ | $146.2 \pm 33.1$ | $144.7 \pm 45.6$ | $\mathbf{150.0 \pm 34.3}$ |
| Large | $130.2 \pm 93.3$ | $\mathbf{191.2 \pm 51.8}$ | $178.6 \pm 74.2$ | $186.1 \pm 61.5$ |
| **Average** | $110.7 \pm 74.9$ | $155.6 \pm 51.2$ | $154.6 \pm 58.0$ | $\mathbf{157.7 \pm 51.9}$ |
| **Kitchen** Mixed | $34.8 \pm 13.8$ | $42.0 \pm 18.0$ | $12.6 \pm 18.4$ | $\mathbf{67.6 \pm 11.5}$ |
| Partial | $36.7 \pm 21.7$ | $58.4 \pm 25.7$ | $3.7 \pm 9.9$ | $\mathbf{80.8 \pm 24.4}$ |
| **Average** | $35.7 \pm 18.2$ | $50.2 \pm 23.7$ | $8.2 \pm 15.4$ | $\mathbf{74.2 \pm 20.2}$ |

report episode return mean ± standard deviation of three trainings × 300 evaluations per dataset; higher is better. The four experimental conditions are: (a) unbiased sampling without reweighting, guidance, or adaptive guide scale; (b) reweighting with $\beta_{\max} = 3$ without guidance or adaptive scaling; (c) reweighting with $\beta_{\max} = 3$ combined with guidance at a fixed guide scale $s' = 20$, but no adaptive scaling; and (d) the full TGDP algorithm with reweighting, guidance, and adaptive guide scale enabled and full default parameters. (a) Our results indicate that baseline unbiased sampling consistently yields suboptimal performance across all domains, confirming the necessity of reweighting. This is unsurprising as datasets contain mixed quality behavior data. (b) While reweighting alone improves performance in several tasks, it fails to reliably achieve maximal results across datasets. This confirms our analysis in Appendix A.1 and evaluations in Appendix B.2 that in practice the effect of reweighting on the sampling distribution is limited. (c) Guided sampling with a fixed scale is unable to reliably maximize performance across task and is even adversely affecting performance in many settings as overguidance can lead to manifold deviation. (d) The full TGDP algorithm, which adaptively adjusts the guide scale during inference, demonstrates superior and consistent performance gains, underscoring the critical role of adaptive scaling in balancing under- and overguidance across datasets. These findings clarify the individual contributions and synergistic effects of TGDP components, validate the design choices underpinning the TGDP framework, and highlight the importance of adaptive guidance for achieving state-of-the-art results across diverse planning tasks under default parameter settings.

## C.2 PERFORMANCE AND RUNTIME TRADE-OFFS OF DIFFUSION PLANNERS

Practical deployment of diffusion planners face the challenge of computational overhead due to the iterative sampling procedure. Many approaches accelerate general diffusion sampling through optimized noise schedules and deterministic samplers (e.g., DDIM (Song et al., 2021)), progressive distillation (Salimans & Ho, 2022), or flow matching (Lipman et al., 2023; Liu et al., 2023). Specifically in diffusion planning, inference times are often minimized through warm-starting the diffusion process with partially re-noised plans from a previous decision step (Dong et al., 2024b) or by distilling the computationally expensive planner into a lightweight policy (Lu et al., 2025b). DiffuserLite (Dong et al., 2024a) significantly speeds up diffusion planning through a combination of a fast DIT architecture (Peebles & Xie, 2023), a hierarchical decomposition of planning horizons with

Table 5: Performance and runtime trade-offs of advanced diffusion planners. Values show episode return mean ± standard error of three trainings × 300 evaluations (higher is better). Runtime is compared in terms of neural evaluations (lower is better) and decision frequency when diffusing a single sample or 100 samples in parallel (higher is better). Frequency values are obtained on AMD Ryzen9 7950X CPU @ 4.5 GHz and NVIDIA GeForce RTX 4090. Diffusion planners show similar performance at slowest settings, with faster variants trading off accuracy for speed. DiffuserLite achieves superior computational efficiency but requires per-task tuning. While DV requires many neural evaluations, due to batched inference, DV and TGDP exhibit comparable decision rates for single environments. However, DV's speed drops with large parallel batches due to hardware limits. TGDP-V suffers steep declines at low step counts, likely due to its adaptive guide scale requiring smaller step sizes to avoid misadjusted guidance.

| Dataset | Environment | DiffuserLite | | | Diffusion Veteran | | | TGDP-Veteran (Ours) | | |
|---|---|---|---|---|---|---|---|---|---|---|
| | | Diffusion | RecFlow | RecFlow+Reflow | Steps=20 | Steps=8 | Steps=4 | Steps=20 | Steps=8 | Steps=4 |
| Halfcheetah | Med-Expert | 88.5 ± 0.4 | 90.8 ± 0.9 | 84.0 ± 2.9 | 92.7 ± 0.3 | 92.9 ± 0.2 | 93.3 ± 0.2 | 93.1 ± 0.1 | 92.8 ± 0.1 | 92.5 ± 0.0 |
| | Med-Replay | 41.6 ± 0.4 | 42.9 ± 0.4 | 39.6 ± 0.4 | 45.8 ± 0.1 | 44.8 ± 0.0 | 44.2 ± 0.0 | 43.0 ± 0.1 | 42.6 ± 0.1 | 41.3 ± 0.1 |
| | Medium | 48.9 ± 1.1 | 48.6 ± 0.7 | 45.3 ± 0.5 | 50.4 ± 0.0 | 48.1 ± 0.1 | 47.1 ± 0.0 | 45.2 ± 0.0 | 45.0 ± 0.0 | 44.6 ± 0.0 |
| Hopper | Med-Expert | 111.6 ± 0.2 | 110.3 ± 0.3 | 110.1 ± 0.5 | 110.0 ± 0.5 | 109.9 ± 0.2 | 110.4 ± 0.1 | 110.2 ± 0.1 | 110.0 ± 0.2 | 82.7 ± 1.4 |
| | Med-Replay | 96.6 ± 0.3 | 97.8 ± 1.3 | 93.2 ± 0.7 | 91.9 ± 0.0 | 91.6 ± 0.0 | 91.0 ± 0.0 | 95.5 ± 0.1 | 94.5 ± 0.1 | 70.3 ± 0.9 |
| | Medium | 100.9 ± 1.1 | 99.5 ± 0.7 | 96.8 ± 0.3 | 83.6 ± 1.2 | 71.4 ± 0.4 | 69.7 ± 0.4 | 97.0 ± 0.0 | 97.9 ± 0.0 | 97.8 ± 0.1 |
| Walker2D | Med-Expert | 107.1 ± 0.6 | 106.4 ± 0.3 | 106.1 ± 0.7 | 109.2 ± 0.0 | 109.4 ± 0.0 | 109.3 ± 0.0 | 108.9 ± 0.0 | 108.3 ± 0.1 | 98.6 ± 0.7 |
| | Med-Replay | 90.2 ± 0.5 | 84.6 ± 1.7 | 78.2 ± 1.7 | 85.0 ± 0.5 | 86.0 ± 0.1 | 85.1 ± 0.3 | 80.5 ± 0.5 | 74.5 ± 0.6 | 57.3 ± 0.8 |
| | Medium | 88.8 ± 0.6 | 85.1 ± 0.5 | 83.7 ± 1.0 | 82.8 ± 0.1 | 81.1 ± 0.1 | 81.9 ± 0.1 | 78.5 ± 0.4 | 78.7 ± 0.4 | 79.1 ± 0.36 |
| | **Average** | 86.0 | 85.1 | 81.9 | 83.5 | 81.7 | 81.3 | 83.6 | 82.7 | 73.8 |
| Maze2D | Umaze | – | – | – | 136.6 ± 1.3 | 137.3 ± 1.3 | 135.4 ± 1.4 | 138.7 ± 1.4 | 137.4 ± 1.4 | 134.6 ± 1.7 |
| | Medium | – | – | – | 150.7 ± 1.0 | 152.2 ± 1.0 | 146.7 ± 1.1 | 153.1 ± 1.0 | 151.4 ± 1.0 | 107.2 ± 2.5 |
| | Large | – | – | – | 203.6 ± 1.4 | 199.1 ± 1.7 | 200.1 ± 1.6 | 200.9 ± 1.5 | 204.4 ± 1.4 | 164.3 ± 3.0 |
| | **Average** | – | – | – | 163.6 | 162.9 | 160.8 | 164.3 | 162.9 | 160.8 |
| Kitchen | Mixed | 73.6 ± 0.7 | 71.9 ± 1.4 | 64.8 ± 1.8 | 73.6 ± 0.1 | 70.3 ± 0.3 | 73.1 ± 0.3 | 72.7 ± 0.3 | 71.9 ± 0.5 | 52.8 ± 0.9 |
| | Partial | 74.4 ± 0.6 | 69.9 ± 0.7 | 71.4 ± 1.2 | 94.0 ± 0.3 | 71.6 ± 0.4 | 74.3 ± 0.2 | 96.0 ± 0.6 | 81.1 ± 0.9 | 7.3 ± 0.4 |
| | **Average** | 74.0 | 70.9 | 68.1 | 83.8 | 71.5 | 73.8 | 84.3 | 76.5 | 30.1 |
| Neural evaluations per planning step | | 12 | 12 | 4 | 1050 | 450 | 250 | 60 | 24 | 12 |
| Decision Freq. [Hz] | single env | 168 | 170 | 414 | 45.0 | 84.1 | 117.2 | 38.1 | 78.0 | 120.5 |
| | parallel envs | 151 | 156 | 360 | 1.8 | 3.9 | 6.5 | 26.2 | 56.9 | 92.1 |

few planning steps and CFG applied only to the lowest layer, and advanced samplers with very few denoising steps. In particular they compare three sampling methods: DDIM (Song et al., 2021) with three denoising steps per hierarchical level (Diffusion), rectified-flow (Liu et al., 2023) with three denoising steps (RecFlow) and rectified-flow with subsequent reflow training and one denoising step (RecFlow+Reflow). While neither Diffusion Veteran (DV) (Lu et al., 2025a) nor our work primarily focus on inference speed, both implement fast DiT architectures and hierarchical jump-step planning for longer effective horizons (DV and TGDP-V). We here compare trade-offs of performance and computational overhead by reducing the number of denoising steps of the respective sampler (DV uses DDIM, TGDP uses EDM (Karras et al., 2022)) from the default 20 down to 8 or 4. Table 5 compares computational overhead in terms of neural evaluations (lower is better) and decision frequency (higher is better) as well as performance (higher is better) for DiffuserLite, DV, and our TGDP guidance algorithm paired with the Diffusion Veteran architecture (TGDP-Veteran/TGDP-V). We report decision frequency for the case of controlling a single environment and for 100 environments in parallel. Values shows episode return mean ± standard error of three trainings × 300 evaluations for three different versions per algorithm. Decision frequency crucially depends on hardware and implementation, with shown values obtained on a machine with an AMD Ryzen9 7950X CPU @ 4.5 GHz and an NVIDIA GeForce RTX 4090. We here re-ran the DiffuserLite and DV experiments using the CleanDiffuser implementations Dong et al. (2024b) on our hardware for an unbiased speed comparison. As planning horizon strongly affects performance, we here compare decision frequency for an effective horizon of 128 across methods, achieved through three decision levels with strides 32, 8, and 1 in DiffuserLite; and through jump-step planning with stride 4 in DV and TGDP-V. Even though DV denoises 150 parallel samples in the kitchen environment, we here report computational overhead for the default value of 50. All diffusion planners show approximately comparable average performance when using their respective best but slowest version. Faster variants show gradually increasing sampling speed but decreasing performance across methods. The accelerated architecture of DiffuserLite consistently outperforms both DV and TGDP-V in terms of computational overhead, but requires per-task hyperparameter optimization for optimal performance. DV requires significantly more neural evaluations per planning step than both DiffuserLite or TGDP-V. For a single environment, DV and TGDP-V exhibit comparable decision frequencies since both parallelize computation

Table 6: Performance on adroit hand (Rajeswaran et al., 2018). TGDP results show episode return mean ± standard error of three runs × 300 evaluations; higher is better. Marked results* are from (Lu et al., 2025a). All diffusion planners excel on expert but struggle on cloned datasets, highlighting challenges in steering toward high-performing behaviors. TGDP is comparable but slightly outperformed by IDQL, DD, and DV; TGDP-V shows minimal architectural gains over TGDP.

| Dataset | Environment | IDQL* | DQL* | Diffuser* | DD* | DV* | TGDP (Ours) | TGDP-V (Ours) |
|---|---|---|---|---|---|---|---|---|
| Pen | Expert | 137.8 ± 2.4 | 133.5 ± 3.9 | 99.7 ± 4.8 | 139.8 ± 3.5 | 122.2 ± 1.8 | 136.2 ± 1.2 | 135.4 ± 1.3 |
| | Cloned | 82.3 ± 5.0 | 23.3 ± 4.0 | 61.7 ± 5.0 | 72.0 ± 4.2 | 80.2 ± 2.0 | 48.3 ± 2.0 | 69.1 ± 2.1 |
| Door | Expert | 105.0 ± 0.3 | 104.8 ± 0.3 | 103.0 ± 0.5 | 105.5 ± 0.3 | 104.7 ± 0.5 | 104.4 ± 0.1 | 104.1 ± 0.1 |
| | Cloned | 4.4 ± 0.6 | -0.3 ± 0.0 | 0.1 ± 0.1 | 15.4 ± 0.5 | 1.5 ± 0.0 | 0.4 ± 0.1 | 0.6 ± 0.1 |
| Hammer | Expert | 127.6 ± 0.1 | 128.3 ± 0.1 | 103.1 ± 3.8 | 124.8 ± 2.1 | 125.8 ± 1.1 | 126.9 ± 0.1 | 125.3 ± 0.2 |
| | Cloned | 3.5 ± 0.5 | 0.2 ± 0.0 | 1.2 ± 0.1 | 1.6 ± 0.1 | 11.9 ± 0.7 | 0.4 ± 0.0 | 0.5 ± 0.0 |
| Relocate | Expert | 107.0 ± 0.8 | 108.5 ± 0.6 | 102.2 ± 1.5 | 110.3 ± 1.1 | 108.9 ± 0.2 | 106.7 ± 0.4 | 106.3 ± 0.3 |
| | Cloned | 0.0 ± 0.1 | 0.1 ± 0.0 | -0.0 ± 0.0 | 0.3 ± 0.0 | 0.6 ± 0.0 | -0.2 ± 0.0 | -0.2 ± 0.0 |
| | **Average** | 71.0 | 62.3 | 58.9 | 71.2 | 69.5 | 65.4 | 67.6 |

Table 7: Performance on robomimic low-dimensional (Mandlekar et al., 2021). Values show episode return mean ± standard error over three runs × 300 evaluations; higher is better. Marked results* are obtained from our own re-implementation. DP and DV outperform other approaches. Human data's non-Markovian characteristics and blurred behavioral modes likely challenge TGDP's mode-separation assumptions, yielding mediocre results.

| Dataset | Environment | DP | Diffuser* | DD* | DiffuserLite | DV* | TGDP (Ours) | TGDP-V (Ours) |
|---|---|---|---|---|---|---|---|---|
| Lift | Prof-Human | 100.0 | 93.3 ± 0.8 | 99.0 ± 0.3 | 100.0 | 98.4 ± 0.4 | 93.4 ± 0.8 | 98.3 ± 0.4 |
| | Mixed-Human | 100.0 | 99.2 ± 0.3 | 71.8 ± 1.5 | – | 90.8 ± 0.5 | 80.1 ± 1.3 | 85.8 ± 1.0 |
| Can | Prof-Human | 100.0 | 56.3 ± 1.7 | 95.3 ± 0.7 | 100.0 | 97.2 ± 0.5 | 72.4 ± 1.5 | 95.2 ± 0.7 |
| | Mixed-Human | 94.0 | 59.4 ± 1.6 | 95.4 ± 0.7 | – | 90.7 ± 0.9 | 59.8 ± 1.6 | 82.7 ± 1.2 |
| Square | Prof-Human | 89.0 | 57.8 ± 1.6 | 81.4 ± 1.3 | 81.8 | 94.3 ± 0.7 | 43.1 ± 1.7 | 75.7 ± 1.4 |
| | Mixed-Human | 81.0 | 33.0 ± 1.6 | 32.9 ± 1.6 | – | 84.1 ± 1.2 | 38.0 ± 1.6 | 61.7 ± 1.6 |
| | **Average** | 94.0 | 66.5 | 79.3 | – | 92.6 | 64.5 | 83.2 |

through batching neural evaluations. However, with 100 parallel environments, DV's sampling speed drops sharply as batch sizes approach GPU limits. DV's computational overhead of large parallel ensemble sampling can thus be mitigated with powerful hardware – however, this may not be feasible in resource-constrained robotic systems. While DV's performance decreases slowly when reducing the number of denoising steps down to 4, TGDP-V shows a steep decline when reducing from 8 to 4 (we observed a similar decrease when reducing DV's sampling steps to 2, not shown). We hypothesize that this is a consequence of the adaptive guide scale, which needs smaller stepsizes to successfully adapt during inter-modal steering, thus risking poorly adjusted guidance for very few denoising steps. This is supported by the observation that datasets with a single behavioral mode—which do not have to adapt guide scale during inter-modal steering—reach peak performance for 4 denoising steps, while multi-modal ones show strongly diminished results.

## C.3 Additional Experimental Results on Manipulation Tasks

We evaluate several diffusion planners on advanced manipulation benchmarks: adroit hand (Rajeswaran et al., 2018) in Table 6 and robomimic low-dimensional (Mandlekar et al., 2021) in Table 7. Dataset details are provided in Appendix E. For adroit hand, we base our comparison on results from Lu et al. (2025a); for robomimic, we re-implement several diffusion planners in our framework. DV and TGDP(-V) use default configurations; for Diffuser (Janner et al., 2022) and Decision Diffuser (DD; Ajay et al. 2022) we optimized sampling parameters using Optuna (Akiba et al., 2019) over 100 trials. On adroit hand, all methods achieve strong performance on expert datasets but exhibit poor performance on cloned datasets. This pattern suggests that diffusion planners effectively capture system dynamics in high-DoF dexterous manipulation, while steering sampling toward high-performing behavioral subspaces presents a substantial challenge across all approaches. TGDP performs competitively with other methods, though IDQL, DD, and DV show slight advantages. TGDP-V yields only marginal improvements over TGDP, indicating that the simpler architecture already adequately models system dynamics. On robomimic low-dimensional, Diffusion Policy (DP; Chi et al. 2024) and DV show strong results, outperforming other approaches. We hypothesize that the human generated

data presents a significant challenge to our TGDP framework. Specifically, human data often exhibits non-Markovian characteristics due to external factors like teleoperation devices, episode history, or operator-dependent strategies, violating i.i.d. assumptions (Mandlekar et al., 2021). TGDP relies on well-separated behavioral modes with distinct, smooth performance surfaces as formalized in Appendix A, whereas human data rarely presents such clean multimodal structure. Instead, it features continuously varying proficiency levels, inconsistent strategies across demonstrators, and temporal dependencies that blur mode boundaries and increase return variance. These properties likely violate our proposed constraints, leading to misadjusted adaptive guide scales and poorly aligned guidance directions, ultimately hindering guidance toward high-performing subspaces during diffusion sampling.

## D IMPLEMENTATION DETAILS

This appendix provides additional information about the implementation of our approach, including pseudocode of the full algorithm, hyperparameter choices, and training procedures used in our experiments. The goal is to facilitate reproducibility and enable readers to better understand the technical aspects of our method.

### D.1 ALGORITHM

Below, we present the full training and inference algorithm for our temperature-guided diffusion planner. This pseudocode details all key steps, including sampling, weighting, loss computation, and the guided diffusion sampling procedure used for inference. It also references the relevant equations from the main body of the paper. We here describe the case of planning in state-space and deriving actions from an inverse dynamics policy. However, the methodology extends to other realizations of diffusion planning.

### D.2 PARAMETER SETTINGS AND MODEL ARCHITECTURE

This appendix provides details on the model architectures, training procedures, and sampling strategies used in our experiments. All hyperparameter values are summarized in Table 8 for reproducibility.

**Model architecture**  As diffusion backbone we use a transformer-based DiT model (Peebles & Xie, 2023) that has been shown to outperform other architectures such as UNets for diffusion planning (Lu et al., 2025a). To encode the noise scale, we use sinusoidal embeddings, while global conditioning is achieved via an MLP embedding. For classifier-guided and Monte-Carlo selection variants, we employ a Half-DiT to estimate value functions. The (diffusion) inverse dynamics model is implemented as a multi-layer feedforward network. All models utilize an exponential moving average (EMA) variant during inference.

**Diffusion sampling**  For trajectory generation, we use an Euler-Maruyama sampler. We tried out different samplers such as the second order Heun sampler proposed in Karras et al. (2022), but found the difference in performance negligible. As is common in diffusion planners, we use low temperature sampling, which encourages the generation of higher-quality samples by reducing stochasticity (Ajay et al., 2022). The exception is the planner in MCSS, which relies on broad sampling distributions. Noise levels are scheduled according to the Karras schedule (Karras et al., 2022), which geometrically interpolates between minimum and maximum noise levels to prioritize fine detail in later diffusion steps. We apply Karras or TrajectoryKarras (epsilon) scaling to adaptively normalize model inputs and outputs at each noise level, improving numerical stability and preventing gradient explosion.

**Model training**  Model training is performed using the AdamW optimizer with initial learning rate of $2 \times 10^{-4}$ and a cosine annealing schedule. To ensure stable training with batch-wise reweighting, we use a batch size of 256, which is larger than the typical 128. During training, sigma values are sampled from a random log-logistic distribution, which provides a diverse range of noise scales and encourages better generalization. We use a simple (weighted) L2 loss to train all models, respectively. In accordance with previous methods, we weight the first observations at time $t + 1$ with a weight of 10 to decrease errors for the first transition (Dong et al., 2024b) which increases performance in a receding horizon control setting.

---

**Algorithm 1** Temperature-Guided Diffusion Planning

---

**Input:** Diffusion model $D_\theta(\boldsymbol{\tau}; \sigma, \beta)$, policy $\pi_\phi(\boldsymbol{\tau})$, horizon $H$, max. temperature $\beta_{\max}$

**Training:**

**Input:** Dataset $\mathcal{D}$, batch size $B$, learning rate $\eta$, dropout prob. $p_{\mathrm{drop}}$, noise distribution $p(\sigma)$, sample weights $\lambda(\sigma)$

**while** *not converged* **do**

$\quad$ Sample batch $\{(\boldsymbol{s}_{0:H}^{(i)}, \boldsymbol{a}_{0:H}^{(i)}, g_0^{(i)})\}_{i=0}^{B-1} \sim \mathcal{D}$

$\quad$ Sample temperature $\beta \sim \mathcal{U}(-\beta_{\max}, \beta_{\max})$

$\quad$ Sample number of samples for re-weighting $b \sim \mathrm{Bin}(B, 1 - p_{\mathrm{drop}})$

$\quad$ **foreach** *sample* $i \in 0..B-1$ **do**

$\qquad$ Compute sample weight: $w_i \leftarrow \begin{cases} \frac{b \cdot \exp(\beta \cdot g_i)}{\sum_{j=0}^{b-1} \exp(\beta \cdot g_j)} & \text{if } i < b \\ 1 & \text{otherwise} \end{cases}$ $\qquad$ // Eq.4

$\qquad$ Sample sigma $\sigma_i \sim p(\sigma)$

$\qquad$ Sample noise vector $\boldsymbol{\epsilon}_i \sim \mathcal{N}(\mathbf{0}, \boldsymbol{I})$

$\qquad$ Extract (state-)trajectory $\boldsymbol{\tau}_i \leftarrow \boldsymbol{s}_{0:H}^{(i)}$

$\qquad$ Compute sample loss: $\mathcal{L}_D^{(i)} \leftarrow w_i \cdot \lambda(\sigma_i) \cdot ||D_\theta(\boldsymbol{\tau}_i + \sigma_i \cdot \boldsymbol{\epsilon}_i; \sigma_i, \beta) - \boldsymbol{\tau}_i||_2^2$ $\quad$ // Eq. 1

$\quad$ **end**

$\quad$ Compute average loss: $\mathcal{L}_D \leftarrow \frac{1}{B} \sum_{i=1}^{B} \mathcal{L}_D^{(i)}$

$\quad$ Update diffusion model: $\theta \leftarrow \theta - \eta \nabla_\theta \mathcal{L}_D$

$\quad$ (Update policy $\pi_\phi$ )

**end**

**Inference:**

**Input:** Current state $\boldsymbol{s}_0$, noise schedule $\rho(i)$, sampler $S(\boldsymbol{\tau}, \boldsymbol{\tau}', \sigma, \sigma')$, diffusion steps $N$, guidance scaling parameters $s'$, $p$

Initial noise sample: $\boldsymbol{\tau}_N \sim \rho(N) \cdot \mathcal{N}(\mathbf{0}, \boldsymbol{I})$

State conditioning: $\boldsymbol{\tau}_{N,0} \leftarrow \boldsymbol{s}_0$

**foreach** $i \in N..1$ **do**

$\quad$ Get noise levels $\sigma_i, \sigma_{i-1} \leftarrow \rho(i), \rho(i-1)$

$\quad$ Compute high/zero/low temp. targets: $\hat{\boldsymbol{\tau}}_0^{-/0/+} \leftarrow D_\theta(\boldsymbol{\tau}_i; \sigma_i, -\beta_{\max}/0/\beta_{\max})$

$\quad$ Compute target mismatches $\mathbf{d}_{-/+} = \hat{\boldsymbol{\tau}}_0^{-/+} - \hat{\boldsymbol{\tau}}_0^\circ$

$\quad$ Compute adaptive guide scale: $s \leftarrow 1 + s' \cdot \left( \frac{\langle \mathbf{d}_+, -\mathbf{d}_- \rangle}{\|\mathbf{d}_+\| \cdot \|\mathbf{d}_-\|} \right)^p$ $\qquad$ // Eq.6

$\quad$ Diffusion target: $\hat{\boldsymbol{\tau}}_0 \leftarrow \hat{\boldsymbol{\tau}}_0^\circ + s \cdot (\hat{\boldsymbol{\tau}}_0^+ - \hat{\boldsymbol{\tau}}_0^\circ)$ $\qquad$ // Eq.5

$\quad$ Sampling step: $\boldsymbol{\tau}_{i-1} \leftarrow S(\boldsymbol{\tau}_i, \hat{\boldsymbol{\tau}}_0, \sigma_i, \sigma_{i-1})$ $\qquad$ // Eq.2

$\quad$ State conditioning: $\boldsymbol{\tau}_{i-1,0} \leftarrow \boldsymbol{s}_0$

**end**

(Predict action: $\mathbf{a}_0 = \pi_\phi(\boldsymbol{\tau}_0)$)

---

# E   EVALUATION ENVIRONMENT DETAILS

**D4RL locomotion: behavioral mode separation**   The D4RL MuJoCo locomotion datasets (Fu et al., 2020) for hopper, halfcheetah, and walker2d provide standardized offline reinforcement learning benchmarks with three main dataset types, each reflecting a distinct data collection process:

- **Medium**: Collected by running a policy trained with Soft Actor-Critic (SAC) (Haarnoja et al., 2018) that has been stopped halfway through training. The resulting episode distribution features several episodes of intermediate performance with relatively long tails towards higher or lower performance. The challenge here is to consistently sample from the high-performing tail of the data.

- **Medium-Replay**: Consists of all transitions stored in the replay buffer during the training of the medium policy. This dataset contains a mix of data from early (less skilled) and later (more skilled) stages of training, resulting in a broader distribution of behaviors. The

Table 8: Summary of all used hyperparameters.

| Component | Parameter Setting |
|---|---|
| **Architecture** | |
| Diffusion Planner network | DiT (hidden size: 256, attention heads: 8, depth: 2, condition embedding size: 128, non-linearity: Mish, preconditioning: Karras [locomotion, maze2d] / TrajectoryKarras-Epsilon [kitchen, adroit hand, robomimic]) |
| Classifier network (CG/MCSS only) | Half-DiT (hidden size: 256, attention heads: 8, depth: 2, non-linearity: Mish, preconditioning: Karras) |
| Policy network | 2-layer FFNN (units: 512, non-linearity: Mish, norm: batch norm) |
| Diff. policy network (TGDP-V only) | 3-layer FFNN (units: 256, condition embedding size: 64, non-linearity: Mish, preconditioning: Karras) |
| Horizon $H$ | 32 |
| Jumpstep-stride (TGDP-V only) | 1 [locomotion] / 2 [adroit hand, robomimic] / 4 [kitchen] / 15 [maze2d] |
| Plan features | States |
| **Sampling** | |
| Diffusion steps $N$ | 20 |
| Noise scheduler | Karras ($\sigma_{\min}$: 0.002, $\sigma_{\max}$: 80, $\rho$: 7.0) |
| $\sigma$ distribution | RandLogLogistic (loc: 0.5, scale: 0.5, $\sigma_{\min}$: 0.002, $\sigma_{\max}$: 80) |
| Sampler | Euler ($s_{\text{churn}}$: 80, $s_{\text{noise}}$: 0.5 [default] / 1.0 [MCSS]) |
| Guidance | TGDP ($s'$: 20.0, $\beta_{\max}$: 3.0, $p$: 4.0) |
| Diff. policy sampl. (TGDP-V only) | Diffusion steps: 10, sampler: DDPM (temperature: 0.5), guidance: None |
| **Training** | |
| Training steps | 1.000.000 |
| Batch size | 256 |
| Dropout probability | 0.25 |
| Optimizer | AdamW (lr: $2 \times 10^{-4}$, cosine annealing schedule) |
| Discount factor $\gamma$ | 0.997 |
| EMA alpha $\alpha$ | 0.9999 (updated every step) |
| Losses | Weighted L2 (next obs. weight 10.0) |

challenge here is to avoid sampling from poorly performing behaviors while exploiting the knowledge of the episodes from the replay buffer.

- **Medium-Expert**: Formed by mixing equal amounts of data from the medium policy and an expert policy (SAC trained to near-optimal performance). This dataset provides a heterogeneous set of trajectories, combining both suboptimal and expert behaviors. The challenge here is to avoid the medium mode completely and consistently sample from the expert mode as it delivers consistent near-optimal performance.

These differences allow for systematic evaluation of offline RL algorithms under varying data quality and distributional scenarios. We therefore use these datasets to develop our methodology and heavily leverage the idea of latent distributional modes with distinct performance statistics in our formalization in Appendix A as well as in our extensive experimental analysis in Appendix B.

**Maze2D: Long-horizon planning**  The D4RL maze2d (Fu et al., 2020) environments require an agent to navigate a ball with mass through two-dimensional mazes of varying complexity. The environments feature three maze layouts of increasing difficulty: umaze, medium, and large. Each layout features a single goal location which is considered reached inside a radius around the goal position and random initial locations. Rewards are given for reaching the goal. This tests the agent ability to derive long-horizon plans with delayed rewards.

**Franka kitchen: Multi-task composition**  The D4RL kitchen environment (Fu et al., 2020) is a challenging multitask benchmark in which a 9-DoF Franka robotic arm must interact with various household objects—such as a microwave, kettle, light switch, and cabinet doors—to achieve specific goal configurations within a simulated kitchen. The environment is designed to test an agent's ability to generalize and compose relevant behaviors from complex, human-demonstrated trajectories. D4RL provides several types of datasets for this environment, including the *mixed* and *partial* datasets. The *partial* dataset contains demonstrations where the four main subtasks are sometimes completed in sequence, offering sub-trajectories that can be leveraged by selective imitation learning. In contrast, the *mixed* dataset features a variety of subtasks but never includes a trajectory where all four target subtasks are completed together, requiring agents to compose successful behaviors from disparate segments of demonstration data.

**Adroit Hand: Dexterous manipulation with multi-finger hand**  The adroit hand environments (Rajeswaran et al., 2018) feature a 24-DoF Shadow Dexterous Hand performing precise manipulation tasks and requiring fine motor control. It includes four tasks—hammer, door, pen, and relocate—each with two dataset qualities:

- **Expert**: Demonstrations collected by a near-optimally RL trained policy, providing high-quality reference trajectories that achieve task success consistently. The challenge is to accurately imitate expert dexterity without overfitting to narrow behavioral modes.
- **Cloned**: Behavioral cloning demonstrations from a supervised imitation policy trained on (1) expert data and (2) human demonstrations and mixing behaviors of the two policies in equal amounts, resulting in mixed optimal and suboptimal behaviors. The challenge lies in distilling robust dexterous skills from imperfect, error-prone demonstrations.

These datasets test the ability to learn precise, high-dimensional manipulation from both optimal and noisy imitation data, making adroit hand a key benchmark for dexterous offline RL.

**RoboMimic: Object manipulation with gripper**  The robomimic datasets (Mandlekar et al., 2021) provide high-fidelity data from human demonstration for a Franka Emika Panda robot arm performing tabletop manipulation tasks. While the dataset provides both low-dimensional state representations and high-dimensional image-based states, we here only utilize the former. We focus on three tasks—lift, can, and square—each with two human performance levels:

- **Proficient-Human (ph)**: Demonstrations from highly skilled human teleoperators achieving near-perfect task execution with efficient, optimal motions. The challenge is precise imitation of expert manipulation strategies.
- **Mixed-Human (mh)**: Demonstrations from human operators of varying skill levels, containing both successful and failed attempts with suboptimal trajectories. The challenge is robust policy learning from heterogeneous human data exhibiting diverse strategies and failure modes.

These datasets emphasize realistic object manipulation with human-collected data of varying quality, testing selective imitation capabilities.

