# OpenReview forum: "Practical Diffusion Planning via Temperature-Guided Reward Conditioning"
_ICLR.cc/2026/Conference — Submitted to ICLR 2026_

### Official Review · Reviewer_1nMT · 2025-10-15

[review text omitted: it was posted to a different submission]

---

> ### Author Response · Authors · 2025-11-12
>
> This review seems to have been added to the wrong submission.

---

> > ### Comment · Reviewer_1nMT · 2025-11-13
> > **Correction of comments**
> >
> > Sorry for the misuploaded materials. I have updated the comments here.

---

> ### Author Response · Authors · 2025-11-20
> **Response 1/2 to Reviewer 1nMT**
>
> Thank you for this incredibly detailed and insightful feedback. We welcome the positive assessment of the writing and motivation, the soundness and elegance of the proposed method, and the comprehensiveness of the theoretical intuition and empirical evaluation. We would like to address your raised issues and questions individually and hope to further improve your view on the quality of the paper. In the interest of space, as some questions and weaknesses are closely related, we answer these jointly.
>
> **Weakness 1: Limited novelty as TGDP is a version of CFG rather than a planning paradigm**
>
> You are correct that TGDP is an extension of CFG that allows for generating samples which align with some numerical value and thus is independent of the underlying architecture. Offline RL is the most obvious application but there may be others that could benefit from the methodology. We consider this a strength as TGDP can be paired with any diffusion planning architecture (as we show with TGDP-V). We note in the Abstract: *[...] TGDP, which adapts CFG for reward-conditioning by self-calibrating to these task specific characteristics*, and similarly in the Introduction and Methodology.
>
> We now also explicitly state in the Conclusion: *”This work introduces Temperature-Guided Diffusion Planning (TGDP), a novel guidance framework that addresses critical limitations of static guide scales of classifier-free guidance in diffusion planning.”*
>
> **Weakness 2 and Question 1: The $\beta_{max}$ parameter**
>
> Thank you for pointing out that we do not give proper guidance on selecting $\beta_{max}$.
> We found that TGDP works robustly for broad ranges of this parameter. Importantly, it is not a task-specific parameter but can be set to a default value across tasks.
>
> To provide guidance on selecting $\beta_{max}$ :
> - We added the discussion and guidance on selecting $\beta_{max}$ to the experiments section:
> *”[...]Here, TGDP's temperature hyperparameter is not swept over but fixed to $\beta_\text{max}=3$.
> In practice, this value should be sufficiently large for temperature-conditioned diffusion targets to diverge during inter-modal steering, which is critical for the validity of the adaptive guide scale.
> At the same time, it should remain small enough to prevent single reweighted samples from dominating the loss, since excessive weighting can destabilize training.
> As analyzed in Appendix B.3 for values $\beta_\text{max}\in\{1,3,5\}$, maximal performance across datasets can be obtained for wide temperature ranges.
> However, because temperature and guide scale interact multiplicatively, adjusting $\beta_\text{max}$ also necessitates corresponding retuning of $s'$.”*
> - We added the parameter $\beta_{max}$ to Table 1 to increase its visibility.
>
> **Weakness 3+4+6, Question 4+5+7: Limitations in mathematical formalization, experimental evaluation, and computational overhead**
>
> You are pointing out many very relevant limitations of the current paper, which we consider very interesting paths for future research. In the interest of your time, we subsume these issues and questions as "Limitations and Future Work" and answer them briefly in the paper but are happy to discuss them further.
>
> We added a Section “Limitations and Future Work”:
> *”[...] TGDP shows moderate computational complexity compared to other guidance paradigms, it does add one additional neural evaluation per diffusion step leading to a 50% increase in computational complexity w.r.t. regular CFG. This may prove problematic in systems that work on constrained resource or time budgets or utilize very deep models. Common strategies to decrease computational complexity include the adoption of diffusion variants that reduce the number denoising steps required for high-quality samples [1,2] or warm-starting the diffusion process with partially re-noised plans from a previous decision step [3]. It remains unclear how TGDP's adaptive guide scale interacts with a strongly reduced number of diffusion steps or initial diffusion samples at low noise-regimes around the data manifold, thus leaving ample opportunities for further research. A second limitation is the unclear generalization to arbitrary task settings. In our derivations in Appendix 1, we constrain datasets to consist of well-separated behavioral modes that individually emit smooth value surfaces. While we formalize the intuition behind TGDP's guidance approach and adaptive scaling, we do not provide a formal convergence analysis of the sampling algorithm. Consequently, there may exist datasets that violate the proposed constraints or interact adversely with our proposed framework, leading to diminished performance or even catastrophic failure of TGDP. In addition, while our evaluation shows strong performance of TGDP, it focuses on a select set of benchmark environments.*

---

> > ### Author Response · Authors · 2025-11-20
> > **Response 2/2 to Reviewer 1nMT**
> >
> > *Exploring generalization to more diverse datasets, including complex robotic environments with sensory state input, as well as a further formal analysis of temperature guidance therefore remain exciting directions for future work.*
> >
> > *[1] Tim Salimans and Jonathan Ho. Progressive Distillation for Fast Sampling of Diffusion Models, 2022.\
> > [2] Xingchao Liu, Xiwen Zhang, Jianzhu Ma, Jian Peng, and Qiang Liu. InstaFlow: One Step is Enough for High-Quality Diffusion-Based Text-to-Image Generation, 2024.\
> > [3] Zibin Dong, Yifu Yuan, Jianye Hao, Fei Ni, Yi Ma, Pengyi Li, and Yan Zheng. CleanDiffuser: An Easy-to-use Modularized Library for Diffusion Models in Decision Making, 2024.“*
> >
> > **Weakness 5 and Question 1+6: Adaptive or optimized temperature $\beta$**
> >
> > You raise a very interesting question which we have not yet considered. We agree that the method could probably be extended through a learned or theoretically motivated temperature value or potentially even a temperature schedule. It might be possible to formalize the notion of “large enough to significantly bias the diffusion targets but small enough to not destabilize training” and directly integrate this into the training process. We are currently considering how to best implement this idea and, in case it yields interesting results, discuss it in a later revision.
> >
> > **Question 2: Generalization to other other frameworks**
> >
> > Thank you for this very relevant question. Our adaptive guide scale fundamentally relies on temperature-conditioned targets representing weighted combinations of score and return gradients, which is a premise specific to our formulation. While classifier-guidance shares issues of manifold deviation due to overguidance and insufficient exploitation of return gradients due to underguidance, we do not see a similar argument based on a separately trained classifier. While we could compute our adaptive guide scale and apply it to classifier gradients, this seems wasteful and we don’t see a benefit. We extended our state of the art to discuss approaches to mitigating manifold deviation in guided diffusion sampling (-> see Reviewer McDw - Weakness 3).
> >
> > Regarding more general decision-making problems, we do not see our results extending beyond decision-making for return maximization with diffusion models, i.e., diffusion planning and diffusion policies. While we do not see any reason why TGDP (as a guidance mechanism) should generally fail to optimize actions in a diffusion policy, one issue that might arise is the low dimensionality of the action space compared to extended state plans. Our adaptive guide scale operates on the assumption that diffusion targets during inter-modal steering are uncorrelated in the data space and thus tend to be orthogonal in high dimensions, which might break in lower dimensional action spaces. We would likely need to retune the hyperparameters of the adaptive guide scale or directly adapt it to data dimensionality.
> >
> > We hope this answers your question.
> >
> > **Question 3: Further justification of the adaptive guidance via collinearity heuristics**
> >
> > We agree that stronger links between formalization and results would strengthen the paper. We acknowledge the handwavy notion of “disconnected modes and smooth surfaces” could be formalized, but resulting constraints would likely be tight and difficult to verify in practice. While we agree that there is more to explore in the mathematical formalization of reweighting-based guidance, we think that this paper would benefit more from additional visual evidence of how the adaptive guide scale interacts with uni-/multi-modal data distributions. We are currently considering how to best implement this idea and, in case it yields interesting results, discuss it in a later revision.
> >
> > **Conclusion**
> >
> > Thank you again for your detailed engagement with the deeper technical questions of TGDP. We believe our revisions—including explicit positioning of TGDP as a guidance framework that extends CFG, comprehensive guidance on temperature selection, a detailed discussion of computational overhead and generalization limitations, and an expanded state of the art addressing manifold deviation in guided diffusion—directly address your thoughtful critiques and substantially improve the paper’s clarity, rigor, and positioning. Given the short time, we will try our best in exploring your ideas of optimizing or scheduling the temperature value and adding better visualization of inter-/intra-mode guidance and may add the results in a later revision. Please let us know if you are interested in an extended answer regarding the "Limitations and Future Work” issues and questions. We look forward to a fruitful discussion, and hope these changes have further strengthened your confidence in the quality and contribution of our work.

---

> > > ### Comment · Reviewer_1nMT · 2025-11-24
> > > **Acknowledging practical value but maintaining borderline rating due to limited novelty and scope**
> > >
> > > I would like to thank the authors for their detailed and comprehensive response. I have carefully read the rebuttal and reconsidered the paper.
> > >
> > > I acknowledge that the paper makes a valid practical contribution. The proposed TGDP addresses a meaningful and prevalent issue in diffusion planning: the brittleness of hyperparameters in Classifier-Free Guidance (CFG). The solution of using temperature-guided self-calibration is empirically effective in avoiding the need for per-task tuning, which is a significant practical pain point.
> > >
> > > However, after considering the rebuttal, I maintain that this paper is on the borderline for the following reasons:
> > >
> > > 1. Scope and Novelty: As noted in the discussion, the core contribution focuses essentially on the technical refinement of the CFG mechanism—specifically on how to adjust/tune the guidance scale and temperature (beta). While the problem is meaningful, the solution represents an incremental improvement on an existing guidance technique rather than a distinct planning paradigm.
> > >
> > > 2. Relevance to Long-Horizon Tasks: The authors emphasize the benefits for long-horizon tasks. While I agree the method improves performance here, I view this as a side effect of the method's ability to handle multi-modal distributions better than static CFG. The adaptive mechanism is based on geometric constraints of the data distribution (intra- vs. inter-mode), not fundamentally designed specifically for the temporal structure of long-horizon problems. It is a general distributional fix that happens to benefit long-horizon tasks due to their inherent multi-modality.
> > >
> > > 3. Persisting Limitations: The weaknesses identified in the initial review and acknowledged by the authors remain relevant factors in my assessment. Specifically, the method incurs a non-negligible computational overhead (approximately 50% increase per step), and the reliance on the heuristic of geometric collinearity lacks a formal convergence analysis.
> > >
> > > In summary, while the problem solved is meaningful and the method is practically useful, the incremental nature of the novelty and the remaining limitations prevent me from giving a stronger rating. I view this work as Borderline.

---

> > > > ### Author Response · Authors · 2025-12-03
> > > > **Response to Follow-Up of Reviewer 1nMT 1/1**
> > > >
> > > > Thank you again for taking time to deeply interact with our work and give detailed feedback. While we acknowledge your decision, we still would like to point out that we have by now added extensive additional experimental evaluations on manipulation benchmarks (relating to your original Weakness 3 and Question 4) and analyze trade-offs of performance and sampling speed (relating to your original Question 5).

---

### Official Review · Reviewer_McDw · 2025-10-18

**Soundness:** 3
**Presentation:** 2
**Contribution:** 2
**Rating:** 4
**Confidence:** 4

**Summary:**

The author introduces the Temperature-Guided Diffusion Planner (TGDP), an approach that addresses the hyperparameter fragility of Classifier-Free Guidance (CFG). Conventional CFG in diffusion planning often necessitates extensive, task-specific hyperparameter tuning because guidance performance critically depends on adapting to the data manifold and reward distribution. TGDP mitigates this by self-calibrating to these task-specific characteristics through a proposed training and inference scheme. During training, it reweights the diffusion loss for each sample based on its return and a randomly sampled temperature. At inference, TGDP computes an adaptive guidance scale by measuring the geometric collinearity between denoising targets conditioned on high, zero, and low temperatures. The authors provide emprical evidence on D4RL locomotion, Maze2D, and Kitchen benchmarks demonstrating that TGDP consistently matches or surpasses the performance of prior diffusion planners (e.g., CFG, CG, MCSS) while utilizing a single, fixed set of default hyperparameters.

**Strengths:**

1. The paper tackles a highly practical and impactful problem, "hyperparameter brittleness of CFG".
2. The motivation is highly intuitive and reasonbale.
3. The suggested method is simple and novel to my understanding.

**Weaknesses:**

1. The method replace the tuning of CFG's guidance scale and target with its own hyperparameters. However, the maximum temperature is also a critical hyperparameter that defines the training objective and guidance targets at inference. The paper uses a fixed value but does not provide an anylsis of how this value was chosen or how sensitive the model's performance is to it.
2. The adaptive scaling relies on the assumption that the collinearity of diffusion targets is a reliable proxy for distinguishing between intra-mode and inter-mode guidance. While this appears to hold true for the tested D4RL benchmarks, these environments, while standard, may not cover all possible return landscape complexities. The paper would be strengthened by a brief discussion on potential failure modes or types of data distributions where this geometric heuristic might be less effective.
3. Although this pepr focused on improving CFG, I suggest the author include more related works based on CG that tackles inaccurate guidance [1-3].

[1] Contrastive Energy Prediction for Exact Energy-Guided Diffusion Sampling in Offline Reinforcement Learning, 2023

[2] Inference-Time Policy Steering through Human Interactions, 2025

[3] Local Manifold Approximation and Projection for Manifold-Aware Diffusion Planning, 2025

Comment: I believe this work has promise. While I currently recommend borderline reject, I am willing to raise my score if the authors adequately address my concerns through revisions or clarifications.

**Questions:**

see weakness

---

> ### Author Response · Authors · 2025-11-20
> **Response 1/2 to Reviewer McDw**
>
> Thank you very much for your insightful and actionable review of our paper. We are pleased that you share our view on the high relevance of the tackled problem and novelty of the proposed solution. We also thank you for your assessment of our work having promise and hope to be able to convince you to raise your score through clarifying unclear points below and integrating given advice in our revision.
>
> **Weakness 1: The maximum temperature parameter**
>
> Thank you for pointing out that we do not give proper guidance on the selection of $\beta_{max}$. In our experiments we found that TGDP works robustly for broad ranges of this parameter. Importantly, it is not a task-specific parameter but can be set to a default value across tasks. This is in contrast to the selection of target return of traditional return-conditioning, which requires task-specific tuning of the condition to the task-specific return distribution. It should be noted though that the temperature $\beta_{max}$ and the guide scale range $s’$ have to be set jointly as they interact multiplicatively (see Appendix A2).
>
> To provide guidance on selecting $\beta_{max}$ :
> - We added the discussion and guidance on selecting $\beta_{max}$ to the experiments section:
> *[...]Here, TGDP's temperature hyperparameter is not swept over but fixed to $\beta_\text{max}=3$.
> In practice, this value should be sufficiently large for temperature-conditioned diffusion targets to diverge during inter-modal steering, which is critical for the validity of the adaptive guide scale.
> At the same time, it should remain small enough to prevent single reweighted samples from dominating the loss, since excessive weighting can destabilize training.
> As analyzed in Appendix B.3 for values $\beta_\text{max}\in\{1,3,5\}$, maximal performance across datasets can be obtained for wide temperature ranges.
> However, because temperature and guide scale interact multiplicatively, adjusting $\beta_\text{max}$ also necessitates corresponding retuning of $s'$.*
> - We added the parameter $\beta_{max}$ to Table 1 to increase its visibility.
>
> **Weakness 2: Limitations of the geometric considerations**
>
> Yes. This is an excellent point. We pose some constraints on the geometry of the data distribution and the return surface in Appendix A. We fully agree that limitation of TGDP should be discussed in further detail at a more prominent point in the paper.
>
> We added a Section “Limitations and Future Work”, discussing limitations of the method and potential issues from specific dataset geometries: *”[...] unclear generalization to arbitrary task settings. In our derivations in Appendix A, we constrain datasets to consist of well-separated behavioral modes that individually emit smooth value surfaces. While we formalize the intuition behind TGDP's guidance approach and adaptive scaling, we do not provide a formal convergence analysis of the sampling algorithm. Consequently, there may exist datasets that violate the proposed constraints or interact adversely with our proposed framework, leading to diminished performance or even catastrophic failure of TGDP. In addition, while our evaluation shows strong performance of TGDP, it focuses on a select set of benchmark environments. Exploring generalization to more diverse datasets, including complex robotic environments with sensory state input, as well as a further formal analysis of temperature guidance therefore remain exciting directions for future work.”*

---

> > ### Author Response · Authors · 2025-11-20
> > **Response 2/2 to Reviewer McDw**
> >
> > **Weakness 3: Limited discussion of stabilizing Classifier-Guidance**
> >
> > We fully agree that the paper would benefit from a more detailed analysis of approaches to stabilize CG and avoid manifold deviation during inference.
> >
> > Based on your advice, we have significantly extended and improved our Related Work section, adding a paragraph on the prevention of manifold deviation in diffusion planning, including but not limited to the proposed papers:
> > *”Infeasible behaviors resulting from manifold deviation are a recognized challenge in guided diffusion planning. Common practice constrains diffusion targets to marginals via simple clipping; however, this often fails to produce meaningful behaviors and may bias samples towards boundary values [1]. Advanced approaches address manifold deviation in various ways: by predicting plan feasibility through stability estimation under re- and subsequent denoising steps [2] or via exploratory value estimates [3], and using these predictions to guide sampling toward the data manifold; by replacing the classifier's traditional MSE loss with a contrastive energy-based objective that enables exact and stable classifier guidance estimation [4]; by leveraging human feedback during generation to steer samples away from implausible or low-quality regions[5]; by generating plan ensembles and using tree-based state aggregation to identify and prioritize reliable states through consensus-based weighting [6]; or by locally approximating the data tangent space and projecting diffusion targets onto it during sampling to enforce manifold consistency [7].*
> >
> > *[1] Aaron Lou and Stefano Ermon. Reflected Diffusion Models. In International Conference on Machine Learning, 2023.\
> > [2] Kyowoon Lee, Seongun Kim, and Jaesik Choi. Refining Diffusion Planner for Reliable Behavior Synthesis by Automatic Detection of Infeasible Plans, 2023.\
> > [3] Zihao Liu, Xing Liu, Yizhai Zhang, Zhengxiong Liu, and Panfeng Huang. Curiosity-Diffuser: Curiosity Guide Diffusion Models for Reliability, arXiv:2503.14833, 2025.\
> > [4] Cheng Lu, Huayu Chen, Jianfei Chen, Hang Su, Chongxuan Li, and Jun Zhu. Contrastive energy prediction for exact energy-guided diffusion sampling in offline reinforcement learning, 2023.\
> > [5] Yanwei Wang, Lirui Wang, Yilun Du, Balakumar Sundaralingam, Xuning Yang, Yu-Wei Chao, Claudia Pérez-D’Arpino, Dieter Fox, and Julie Shah. Inference-time policy steering through human interactions, 2025.\
> > [6] Lang Feng, Pengjie Gu, Bo An, and Gang Pan. Resisting stochastic risks in diffusion planners with the trajectory aggregation tree, 2024.\
> > [7] Kyowoon Lee and Jaesik Choi. Local Manifold Approximation and Projection for Manifold-Aware Diffusion Planning, 2025.”*
> >
> > **Summary**
> >
> > Thank you again for the thoughtful and actionable feedback. We hope that our revisions improve your assessment of the work. The main changes include a detailed discussion of the maximum temperature parameter and guidance on its selection, a new “Limitations and Future Work” section addressing geometric assumptions and potential failure modes, and an expanded SotA, discussing recent approaches to avoid manifold deviation in diffusion planning. We look forward to further discussion and appreciate your time and consideration.

---

> > > ### Comment · Reviewer_McDw · 2025-11-21
> > >
> > > Thank you for the detailed and comprehensive rebuttal. I appreciate the authors’ thorough responses to Weaknesses 2 and 3, which help clarify the contribution of the proposed research and further improve the paper.
> > >
> > > Regarding Weakness 1, the authors have provided convincing clarification and experimental evidence demonstrating how effectively the maximum temperature parameter can be selected.
> > >
> > > One minor suggestion: it would be helpful if the revised sections of the manuscript were highlighted in a different color (for example, blue), as this would make it easier to identify the updated parts.
> > >
> > > Since all of my concerns have been well addressed, I have raised my score to 8.
> > >
> > > Note: Although the empirical performance of TGDP is not dramatically superior to prior work, I still find the paper valuable because it addresses a very practical challenge: reducing the complexity of hyperparameter selection in diffusion based planning. I believe this direction is promising, and I encourage the authors to continue exploring it, potentially extending the approach to more complex robotic environments in future work.

---

> ### Comment · Reviewer_McDw · 2025-11-22
>
> After revisiting the authors’ rebuttal and reading Reviewer Payn’s detailed comment, I realized that my previous evaluation might have been overly optimistic. I would like to reconsider my assessment based on the following remaining concerns.
>
> 1. After carefully revisiting both the main paper and the rebuttal, I agree with Reviewer Payn that a substantial gain might (possibly) come from the weighted-regression trick rather than the proposed adaptive guide scale. To properly isolate the contribution of the adaptive guidance mechanism, the ablation study requested by Reviewer Payn appears necessary.
> 2. Although the authors acknowledge limitations regarding the applicability of the collinearity assumption, further empirical evidence is needed. Since rebuttal time still remains, I would like to ask the authors to provide additional analysis on:
> (1) whether the collinearity assumption holds in more complex or realistic environments, and
> (2) if it does not hold, what specific failure modes arise and how severe they are.
>
> Given these unresolved concerns, I believe my earlier score should be reconsidered. I look forward to further clarification from the authors within the remaining rebuttal period.

---

> > ### Author Response · Authors · 2025-12-03
> > **Response to Follow-up of Reviewer McDw 1/1**
> >
> > We thank you for your time and advice as well as the reconsideration of the score. In the following we would like to address your follow-up questions and concerns in the hope to further convince you of the quality of our work, potentially raising the score back up to its former value of 8.
> >
> > **1. Ablations**
> > We agree that the ablations suggested by Reviewer Payn are highly relevant towards dissecting the individual contributions of the components of TGDP. We now provide the requested ablations in the updated manuscript in a new Appendix C1. The results clearly show that weighted regression alone does not reliably yield maximal performance and that the adaptive guide scale is crucial towards maximizing return across tasks while using a default set of hyperparameters.
> >
> > **2. Additional experimental evaluations**
> > We now provide additional experiments on complex manipulation environments in Appendix C3. For this we implement both the advanced tasks from the DiffuserLite paper: RoboMimic, and the advanced tasks from the Diffusion Veteran paper: Adroit Hand. This raises the number of tasks we compare our approach on from 14 to 28. The mixed-quality human operator data of RoboMimic seems to pose a challenge to TGDP as it breaks the underlying assumptions of well-separated submanifolds with distinct, smooth return surfaces that we propose in Appendix A. We discuss this in the new appendix and the Limitations section.
> >
> > We are confident to have fully addressed all your concerns and questions. We thank you again for the productive discussion phase.

---

### Official Review · Reviewer_Payn · 2025-10-25

**Soundness:** 3
**Presentation:** 1
**Contribution:** 2
**Rating:** 4
**Confidence:** 2

**Summary:**

This paper introduces Temperature-Guided Diffusion Planning (TGDP), a practical and scalable framework for reward-conditioned diffusion planning. TGDP improves upon Classifier-Free Guidance (CFG), which is widely used in diffusion models but suffers from fragile, task-specific hyper-parameter tuning.

**Strengths:**

This work presents a Practical Diffusion Planning pipeline after DV [1] in the field of diffusion planning. I think this paper will be another attempt on improving diffusion planning, providing inspiration for robotics (especially for the manipulation field), and also offline RL community. The motivation behind this paper is solid: CFG/CG is fast but require heavy hyper-parameter tuning for target return and CFG scale. MCSS (DV), requiring no parameters tuning, provides high-quality inference, but is slow for generating since it requires more unbiased data for selection. This paper provide experiments on the classic D4RL dataset, which should be good for other researcher to have a try.

[1] What Makes a Good Diffusion Planner for Decision Making? Lu et.al ICLR 2025

**Weaknesses:**

I think the main weakness is the paper writing. Till now I still not very sure what is actually temperature conditioned diffusion planning. So if the author can provide a detailed answer to my question, I am considering raising my score. I will be staying online during the rebuttal period. I hope the authors can provide active response during rebuttal.

1) What is $\beta_{max}$ for? Is that a task-specific parameter? Why do we need $\beta_{max}$? Is $\beta_{max}$ also a condition that is required to feed into the network of diffusion?
2) What is the intuition behind designing eq (5)? Could the author elaborating more on the introduction of **cosine-similarity** between the 0/high/low-temperature diffusion output? Any math / intuition supported is quite encouraged. My understanding: "if predictions come from different modes → reduce scale (avoid over-guidance) If within same mode → strengthen guidance (avoid underguidance)" but this makes no sense to me, I can also say "if predictions come from different modes → you should improve scale (to quickly passing the confusion state)." I am hoping to get more insights here.
3) For Algorithm 1 line 1322, there's $\tau_{low}$, but it is never used below? This is quite confusing. Is that a typo?
4) Can the authors provide more experiments on other datasets, except for the 9 classic dataset in D4RL?
5) I do not find any codes in OpenReview or from the paper.

**Questions:**

If CFG can be represented as:

 `sample = D(zero) + s * (D(target_return) - D(zero))`,

where target_return requires to be tuned.

Can TGDP be summarized in one sentence, as:

`sample = D(eps, zero) + s_adaptive * (D(eps, $\beta_{max}$) - D(eps, zero))`,

where s_adaptive is large when D(eps, $\beta_{max}$) is similar to D(eps, -$\beta_{max}$), otherwise it is small?

Is my understanding is correct?

---

> ### Author Response · Authors · 2025-11-20
> **Response 1/2 to Reviewer Payn**
>
> We thank the reviewer for the valuable and insightful feedback and for the acknowledgement of the relevance of the work. We improved the manuscript according to your feedback by making the following clarifications and adaptations, and hope to convince you to consider raising your score.
>
> **Weakness 0: What is TGDP?**
>
> We apologize that the overall presentation of our methodology is not sufficiently concise. TGDP is an extension of return-conditioned diffusion planning with CFG, based on the combination of three components we describe in the methodology section: (1) temperature-conditional reweighting and sampling, (2) temperature-guidance through combining conditional and unconditional denoising targets, and (3) adaptive guidance scaling based on our geometric considerations.
>
> To increase conciseness and flow of the presentation:
> - We adapted the methodology section to increase readability through a clearer separation of the individual components of the algorithm (reweighting, guidance, and adaptive scaling) as individual paragraphs in the methodology section.
> - We added a figure explaining TGDP’s training and inference.
> - We add a concluding sentence to the methodology section: *”The full TGDP algorithm combines temperature-weighted training with weights from Equation 4 with temperature-conditioned guidance in Equation 5, using the adaptive guide scale from Equation 6. This approach reduces reliance on fixed guide scales, which
> inherently struggle to optimally exploit return gradients while preventing manifold deviation.”*
>
> **Weakness 1: Unclear role of $\beta_{max}$**
>
> Thank you for pointing out that we should clarify the role of $\beta_{max}$. It is used both during training and during inference: During training, we randomly draw a temperature $\beta$ from a uniform distribution between $-\beta_{max}$ and $+\beta_{max}$ which is then used in our exponential reweighting scheme to define the strength of the reweighting. The temperature is also passed as an auxiliary input to the diffusion model. During inference, we condition the diffusion model on different temperatures: {$-\beta_{max}$, 0, $+\beta_{max}$} to obtain different diffusion targets which we then combine to compute the adaptive guide scale and the guidance vector (so you are right that during inference $-\beta_{max}$ is fed to the network as our counterpart to the target return in traditional return-conditioned CFG).
>
> You are also correct that at this point we do not provide proper guidance on selecting $\beta_{max}$. In our experiments we found that TGDP works robustly for broad ranges of this parameter. Importantly, it is not a task-specific parameter but can be set to a default value across tasks. This is in contrast to the target return of traditional return-conditioning, which requires task-specific tuning of the condition to the task-specific return distribution. It should be noted though that the temperature $\beta_{max}$ and the guide scale range $s’$ have to be set jointly as they interact multiplicatively (see Appendix A2).
>
> To improve the visibility of the explanation of $\beta_{max}$.
> - We added the discussion and guidance on selecting $\beta_{max}$ to the experiments section: *”[...]Here, TGDP's temperature hyperparameter is not swept over but fixed to $\beta_\text{max}=3$. In practice, this value should be sufficiently large for temperature-conditioned diffusion targets to diverge during inter-modal steering, which is critical for the validity of the adaptive guide scale.At the same time, it should remain small enough to prevent single reweighted samples from dominating the loss, since excessive weighting can destabilize training. As analyzed in Appendix B.3 for values $\beta_\text{max}\in\{1,3,5\}$, maximal performance across datasets can be obtained for wide temperature ranges. However, because temperature and guide scale interact multiplicatively, adjusting $\beta_\text{max}$ also necessitates corresponding retuning of $s'$.”*
> - We added the parameter $\beta_{max}$ to Table 1 to increase its visibility.
> - We added crossreferences between the main body of the paper and the relevant parts of Appendices A (mathematical formalization) and B (additional evaluations) to facilitate the search for extended information of the individual parts of the algorithm.
>
> **Weakness 2: Unclear reasoning behind collinearity assumption**
>
> Thank you for raising this issue. We realize that the justification of the collinearity assumption falls short in the main body of the paper. The idea of collinearity in the case of intra-mode guidance and non-collinearity during intra-mode guidance is a result of our mathematical formalization in Appendix A2 and A3.

---

> > ### Author Response · Authors · 2025-11-20
> > **Response 2/2 to Reviewer Payn**
> >
> > Simply put, if the diffusion targets for low, zero, and high temperature conditioning point to the same probabilistic mode of the data distribution, the differences of targets (high temp. target - zero temp. target, and zero temp. target - low temp. target) are dominated by the local gradients of the return surface, and are therefore collinear. By contrast, if the targets point to different modes of the data distribution, the differences between the targets represent different weighted combinations of uncorrelated vectors and thus most likely are not collinear. We use the cosine-similarity as a measure of collinearity. We have also tried out different measures of collinearity but there was no significant difference in the results. The idea now is that if the high temperature target already reliably points to a high-performing mode of the data distribution, then we should simply follow this target to achieve high performance. If we were to extrapolate (as is common in CFG) in this case, we would encounter “overguidance” which results in the sample being pushed away from the data distribution. On the other hand, if all targets point to the same mode, we should use the gradient information of the return surface by extrapolating beyond the high temperature target and thus performing gradient ascent on the return surface.
> >
> > We have added a clearer description of the results of Appendix A in the Methodology section and added crossreferences to point to the more extensive mathematical formalization in Appendix A and the additional quantitative evaluations in Appendix B.
> > We hope that this clarifies your concern.
> >
> >
> > **Weakness 3: Typo in algorithm**
> >
> > Yes this is a typo. Thank you very much for catching it. We corrected it and carefully went through our presentation of notation to root out further inconsistencies.
> >
> > **Weakness 4: Insufficient experimental evaluation**
> >
> > We provide additional experiments on the three maze2d and the two kitchen environments in Section 4 (see Table 2 and Paragraph “Comparison of diffusion planners”).
> > We extensively leverage the 9 locomotion environments for the development of our method in Section 3 and Appendix A + B and for the comparison of different guidance methods in Section 4, Paragraph “Comparison of guidance mechanisms”.
> > Importantly, the additional experiments on maze2d and kitchen are conducted without adapting the guidance hyperparameters and can thus be seen as a test set of the generalization of TGDP.
> > The results on additional environments (maze2d and kitchen) are therefore in the paper but are presented less prominently than the ones for the 9 locomotion tasks.
> >
> > We hope this clarifies your concern but let us know if you think we should present the additional experiments more prominently.
> >
> > **Weakness 5: Missing code**
> >
> > We have not provided the code repo at the time of submission but have now made it available here: https://github.com/iclr26-submission24685/tgdp_submission.git.
> >
> > **Question 1: Summarization of TGDP**
> >
> > Your summary is almost correct. In TGDP we indeed compute:
> >
> > > sample = $D(0) + s_{adaptive} \cdot (D(\beta_{max}) - D(0))$ (we are not sure what you mean by eps; could you elaborate?)
> >
> > however, $s_{adaptive}$ is large if the direction of the vectors $D(\beta_{max})-D(0))$ and $(D(0)-D(\beta_{max}))$ are similar and small if the direction is different. This realizes our derivations in Appendix A that guide scales should be large if the diffusion targets $-D(\beta_{max})$,  $D(0))$, and $D(\beta_{max})$ are collinear, and small if they are not. As a result, if diffusion targets point to different data modes (inter-mode guidance), the guide scale is small and we approximately follow the diffusion targets $D(\beta_{max}$. If diffusion targets point to the same data mode (intra-mode guidance), the guide scale is large and we follow extrapolated diffusion targets beyond $D(\beta_{max}$ (as is commonly done in CFG). We provide the derivations of this in Appendix A3 but we agree with your point that the presentation of the details of the algorithm in the methodology section should be more concise.
> >
> > We have updated Equation 5 to better represent the intuition of guidance as extrapolating along a dimension in dataspace and Equation 6 to more concisely represent the concept of the adaptive guide scale.
> >
> > **Conclusion**
> >
> > We thank you once again for your constructive and insightful feedback, which helped us clarify key aspects of TGDP and improve the overall readability of the paper. The revisions include a new figure outlining training and inference in TGDP, a clearer description of the role of $\beta_{max}$ and guidance how to choose it, overall improvements of the presentation of the methodology (including a tighter coupling between Section 3 and the Appendices A and B), and a thorough check for inconsistencies in notation. We hope the revisions address your concerns and enhance your impression of the work. We look forward to a fruitful discussion.

---

> ### Comment · Reviewer_Payn · 2025-11-21
> **Thank you for your response**
>
> Thank you for the detailed response from the authors.
>
> 1) What the hyper-parameter of the two added experiments for beta and base guidance scale s?
>
> 2) Given the improvement on the D4RL experiments over previous work is marginal, and the rebuttal period is three weeks, could the authors provide more none-toy environment results like robomimic.
>
> 3) Thank you for your answering. The verbal explanation to question 1 is not convincing to me, without theoretical justification.
>
> 4) If my understanding is correct, the only advantage of your work against previous works is for the low-computation resource (assuming you use the same set of hyper-parameter for all the experiments, w/o fine-tuning them). Also, I find the overall performance is still lower than DffuserLite. How's the performance comparison between yours and DiffuserLite[2]. You can also adopt their neural network.
>
> 5) What's your advantage over distillation/deterministic methods [3,4,5,6,7,8,9,10]. It seems that they do not need hyper-parameter tuning also. If your advantage is the decision frequency, I think you should compare your work with these lines of work. Could you provide more results on the decision frequency against other baselines?
>
> Thank you for your acknowledgement. **The overall contribution can be summarized as** (correct me if not, again)
>
> 1) $s_{adaptive}$
>
> 2) D( $\beta_{max}$ ) - D(0)
>
> where $s_{adaptive}$ is chosen without theoretical support; where D( $\beta_{max}$ ) - D(0) is the same as Decision Diffuser [1] with a constant exp mapping for the target-return, if you use the fixed beta, $\beta_{max}$ is the same as 1.0 for modern neural network, right? So that the main performance gained only from the weighted-regression, which is, actually, a heavily adopted trick for offline-RL. **It is hard to attribute this as your core contribution**, right?
>
> Could the author make an ablation on
>
> (1) w/ 1. w/o 2.
>
> (2) w/o 1. w/ 2.
>
> (3) w/o 1. w/o 2.
>
> On all the main experiments in your paper?
>
> **Reference**
>
> [1] Ajay, Anurag, et al. "Is conditional generative modeling all you need for decision-making?." arXiv preprint arXiv:2211.15657 (2022).
>
> [2] Dong, Zibin, et al. "Diffuserlite: Towards real-time diffusion planning." Advances in Neural Information Processing Systems 37 (2024): 122556-122583.
>
> [3] Nguyen, Thanh, and Chang D. Yoo. "Revisiting Diffusion Q-Learning: From Iterative Denoising to One-Step Action Generation." arXiv preprint arXiv:2508.13904 (2025).
>
> [4] Lu, Haofei, et al. "Habitizing Diffusion Planning for Efficient and Effective Decision Making." arXiv preprint arXiv:2502.06401 (2025).
>
> [5] Wang, Zhendong, et al. "One-step diffusion policy: Fast visuomotor policies via diffusion distillation." arXiv preprint arXiv:2410.21257 (2024).
>
> [6] Zhang, Zhilong, et al. "Flow to better: Offline preference-based reinforcement learning via preferred trajectory generation." The Twelfth International Conference on Learning Representations. 2023.
>
> [7] Zhang, Qinglun, et al. "FlowPolicy: Enabling Fast and Robust 3D Flow-based Policy via Consistency Flow Matching for Robot Manipulation." arXiv preprint arXiv:2412.04987 (2024).
>
> [8] Chen, Yuhui, Haoran Li, and Dongbin Zhao. "Boosting continuous control with consistency policy." arXiv preprint arXiv:2310.06343 (2023).
>
> [9] Prasad, Aaditya, et al. "Consistency policy: Accelerated visuomotor policies via consistency distillation." arXiv preprint arXiv:2405.07503 (2024).
>
> [10] Chen, Huayu, et al. "Score regularized policy optimization through diffusion behavior." arXiv preprint arXiv:2310.07297 (2023).

---

> ### Comment · Reviewer_Payn · 2025-11-21
> **2/2 Please make an explanation for the sparse return-to-go situation**
>
> For this paragraph: "batch of samples τi and normalized discounted returns gi (zero mean, unit variance across the full dataset), each sample is assigned an importance weight w(β, gi)."
>
> I am very curious: **if the trajectory is sampled randomly into a batch, how could it achieve good results for the sparse return-to-go environment?**
>
> States that close to the starting point are penalised all the time, where states that close to the goal point are encouraged all the time, since all these states are equally fed into a batch.
>
> **This seems that it does not jump outside of the weakness of CFG, at all?**
>
> **I hope the authors provide a justification on this point and also the validation of the results on Maze2D and also the Antmaze (results on sparse-reward environment are encouraged).**
>
> Thank you for your response.

---

> > ### Author Response · Authors · 2025-11-25
> > **Response to Follow-up of Reviewer Payn 1/2**
> >
> > We appreciate the detailed feedback. Below, we clarify follow-up questions and discuss requested additions to further demonstrate the quality of our work.
> >
> > **1: Parameter settings**\
> > A key advantage of TGDP is its insensitivity to guidance hyperparameter tuning, allowing us to use default parameter settings ($\beta_\text{max}=3$, $s’=20$) across all experiments.
> >
> > **2: Additional experiments**\
> > We do not aspire to significantly increase per-task performance of the SotA. Instead, our key contribution lies in a novel guidance approach that does not need per-task tuning of guidance hyperparameters. The D4RL benchmark is the standard benchmark used in diffusion planning, thus allowing us to compare against most other methods. However, we do acknowledge that additional experiments would be interesting and currently work on their implementation, to be added in a future revision.
> >
> > **3. Temperature parameter $\beta_\text{max}$**\
> > We empirically show in the ablations in Appendix B that TGDP is robust to wide ranges of $\beta_\text{max}$. In Figure 7 we show that a value between approx. 1 and 5 improves the downstream performance of the planner with performance stagnating for higher values. In Figure 8 we show that for values [1,3,5] we can reach maximum performance on all environments through temperature guidance. Please note that we also provide extensive derivations on the influence of the temperature on the sampling distribution and the score function in Appendix A. While $\beta_\text{max}=3$ is empirically chosen, such parameter selection is standard practice in ML. Could you elaborate on the necessity of a theoretically justified setting of $\beta_\text{max}$?
> >
> > **4. Performance comparison with DiffuserLite**\
> > No, this is not correct. In this paper we aim to bypass hyperparameter optimization in guided sampling, not reduce computational cost. While it is true that we are computationally more efficient than brute-force approaches (DV/MCSS), this is not our key contribution.
> >
> > We acknowledge that DiffuserLite shows increased performance in the 9 locomotion environments over other diffusion planners, including ours. The authors hypothesize that this stems from the short horizons of the individual layers of their hierarchical architecture that allow for the stitching of shorter trajectory segments. DiffuserLite uses CFG for maximizing return, thus requiring per-task hyperparameter tuning. Analogous to our experiments of TGDP-based guidance paired with DV’s architecture, we could introduce “TGDP-DiffuserLite” as a combination of the DiffuserLite architecture and TGDP guidance. We hypothesize that this would result in an algorithm that replicates DiffuserLite’s performance on the locomotion environment without the need of per-task hyperparameter tuning. However, due to substantial architectural differences and time constraints, this remains future work.
> >
> > **5. Distillation and deterministic sampling**\
> > We appreciate the questions about the possibility of speeding up diffusion sampling in combination with TGDP.
> >
> > Regarding deterministic sampling approaches:\
> > Reducing sampling steps is a well-studied avenue to speed diffusion inference. Our core contribution focuses on bypassing hyperparameter search in diffusion guidance, not on improving sampling speed. However, we do acknowledge that it remains unclear how TGDP interacts with few denoising steps, as our guide scale relies on the idea of adapting to the local curvature of the return surface. In response to your comment, we added additional experiments on using a reduced number of diffusion step and a deterministic DDIM sampler in a new Appendix B.6. The results show that maximal performance can be obtained for significantly fewer diffusion steps than the 20 we use in main experiments. Interestingly, the minimum required steps depend on the dataset geometry: single behavioral mode datasets (medium) reach peak performance at around 4 diffusion steps, multi-modal ones (medium-expert, medium-replay) around 8 diffusion steps. Replacing the stochastic sampler with a deterministic one did not significantly affect performance.
> >
> > Regarding plan distillation approaches:\
> > Distilling a trained TGDP model into a lightweight policy is a promising idea but beyond this paper’s scope.

---

> > > ### Author Response · Authors · 2025-11-25
> > > **Response to Follow-up of Reviewer Payn 2/2**
> > >
> > > **6. Contribution and ablations**\
> > > We believe there remains a misunderstanding about our algorithm and how it relates to traditional return-conditional CFG (which is used in Decision Diffuser).
> > > - Traditional CFG:\
> > > (Training) minimize $\sum_i||x_i - D(x_i +\sigma \cdot n, \sigma, g_i)||$\
> > > (Inference) combine predictions $D(x_i, \sigma_i, 0) + s (D(x_i, \sigma_i, g_{target})-D(x_i, \sigma_i, 0))$
> > > - TGDP (temperature-conditional):\
> > > (Training) minimize $\sum_i w(g_i, \beta)\cdot||x_i - D(x_i +\sigma \cdot n, \sigma, \beta)||$ where $\beta$ is randomly sampled between $-\beta_{max}, \beta_{max}$\
> > > (Inference) combine predictions $D(x_i, \sigma_i, 0) + s_{adaptive}\cdot (D(x_i, \sigma_i, \beta_{max})-D(x_i, \sigma_i, 0))$, where $s_{adaptive}$ is based on the collinearity of $D(x_i,\sigma_i, -\beta_{max}), D(x_i,\sigma_i, 0), D(x_i,\sigma_i, +\beta_{max})$.
> > >
> > > Please note that there is no direct mapping between $\beta_\text{max}$ and $g_{target}$.
> > >
> > > Regarding your questions/comments:
> > > 1. “[...] where $s_{adaptive}$ is chosen without theoretical support[...]”: Could you elaborate why it is without theoretical support? While we acknowledge that it is an approximation, we would argue that our derivations in Appendix A support the collinearity assumption and that the cosine similarity is a valid way of testing for it.
> > > 2. “[...] where $D(\beta_{max}) - D(0)$ is the same as Decision Diffuser [1] with a constant exp mapping for the target-return, if you use the fixed beta, $\beta_{max}$ is the same as 1.0 for modern neural network, right?”: No. There seems to be a misunderstanding. We do not do any exp mapping during inference but we compute exp weights during training and use a temperature condition during sampling. This is fundamentally different from the guidance applied in Decision Diffuser.
> > > 3. “So that the main performance gained only from the weighted-regression, which is, actually, a heavily adopted trick for offline-RL. It is hard to attribute this as your core contribution, right?”: Note that we condition on the temperature that is used to reweight samples during training (not a target return). This allows us to condition on a target temperature during inference to align sampling with high returns. To the best of our knowledge this is a novel idea and a core contribution of our work. Also, our ablations show that the performance gain does not come from weighted regression alone (see below).
> > >
> > > Regarding the requested ablations:
> > > - Ablation (3): We provide this ablation in Figure 7 in the Appendix of the revised manuscript. We show performance for the locomotion environments by sweeping $\beta$, following conditioned diffusion targets $D(\beta)$ without guidance. Note that we here clearly show that weighted regression alone does not yield peak performance in most environments.
> > > - Ablation (1): Please note that $D(0)+s_{adaptive}\cdot 0 = D(0)$. The requested ablation therefore corresponds to unbiased sampling and is shown in Figure 7 for the value $\beta=0$.
> > > - Ablation (2): We provide this ablation in Figure 8 in the Appendix of the revised manuscript. We show performance when replacing the adaptive guide scale with a fixed one that we then sweep over. The ablation shows that different environments require different guide scales for maximum performance. From this observation we propose the adaptive guide scale as a means to skip per-task hyperparameter optimization (not to improve per-task performance).
> > >
> > > **7. Sparse return-to-go**\
> > > This is a relevant concern. We acknowledge that in sparse reward environments for $\gamma<1$ samples that start close to the goal are structurally upweighted. Given limited model capacity, this could result in a model that samples less accurate plans if we are far away from the goal. This can easily be countered though if we set $\gamma$ very close to 1 or through a reasonably sized neural net. Please note that during diffusion sampling, we implement contextual state-constraints through inpainting conditioning. Assuming a properly trained model, we thus sample from the set of feasible trajectories, given the current state. Also note that our experiments on Maze2D, shown in Table 2, exhibit strong performance. Could you elaborate on your concern that “it does not jump outside of the weakness of CFG”?
> > >
> > > **Conclusion**\
> > > We thank you for your insightful feedback. In response, we provide all requested ablations to analyze the individual contributions of TGDP’s components and investigate accelerated diffusion sampling in a new Appendix B.6. Additional experiments on complex environments are ongoing. We hope these clarifications and results demonstrate the robustness and significance of our contributions and may convince you to raise your score. We look forward to further discussions.

---

> ### Comment · Reviewer_Payn · 2025-11-26
> **The current version does not meet the standard for acceptance.**
>
> I hope our discussion can focus clearly on the core issues and touch the root. After reviewing your rebuttal and your latest clarifications. **I am disappointed that you are refusing to provide the majority of experiments (which is important for evaluating your work) for my previous response.**
>
> This paper states that the core contribution is bypassing hyper-parameter search for diffusion guidance, rather than improving sampling speed. However, this has already been addressed inMCSS, which—if my understanding is correct—also removes the need for task-specific guidance hyper-parameters. MCSS is indeed very slow, and that was precisely your motivation stated in the introduction, right?
>
> You begin by improving CFG, yet the following concerns remain unresolved:
>
> (1) **Quality**.
> As McDw pointed out, the performance gain over prior SOTA is marginal. Moreover, much of the improvement appears to come from the weighted-regression trick and from an adaptive guidance scheme without theoretical support.
>
> (2) **Speed**.
> According to the analysis above, speed is actually your main contribution, but you decline to provide comparisons with other existing fast baselines. The reported performance is even lower than DiffuserLite, which is also based on CFG.
>
> (3) **Limited results**.
> Most experiments—especially the ablations—are conducted solely on nine toy D4RL tasks, which are no longer considered standard benchmarks. (**how do 9 locomotion tasks become the benchmark for decision making and planning?** You do not need much planning for just moving forward. I think the AntMaze Environment is at least required?) It is unclear why you are unwilling to provide more simulation results, given the long period of rebuttal time?
>
> (4) **Method Weakness**.
> Compared with MCSS, CFG fundamentally struggles on sparse-reward tasks because it cannot adaptively adjust the target return, which MCSS achieves by selecting the relatively best candidates at the cost of heavier computation. **This method does not address this core-limitation.** (And the mention of state-constraints via inpainting conditioning is irrelevant here and already standard practice in prior work.) As you acknowledged, the model may making worse decisions when it is far from the goal. It has difficulty in dealing with long-term decision making and sparse-reward environments.
>
> I think this paper is a pipeline based on CFG, and its conclusion provides little guidance on CFG, MCSS and other methods. Let alone the main conclusion and ablation is from 9 toy locomotion tasks in D4RL. Consequently, I find it is possible an over-claim by saying a general guidance for diffusion planning.
>
> I have decided to reject this paper, as the current revision does not meet the standard for acceptance. I hope you can refine the work more thoroughly for future submissions. And I think it still has great potential in the future.
>
> [1] Ajay, Anurag, et al. "Is conditional generative modeling all you need for decision-making?." arXiv preprint arXiv:2211.15657 (2022).

---

> > ### Author Response · Authors · 2025-12-03
> > **Response to Follow-Up of Reviewer Payn 1/1**
> >
> > We thank you for the active participation in this discussion phase. Please note that we have in no way refused to implement your requested experiments. Instead we have clearly stated that we aim to add them in a later revision, as extensive experimental evaluation takes time under constrained resource budgets. Consequently, we have now implemented all of your requested additional experiments and ablations, significantly extending the paper. Below we present individual changes and clarify open questions.
> >
> > **Changes**
> > - We provide the requested ablation of the individual components of TGDP for all main experiments in Appendix C1. Among other things, the experiments clearly show that weighted regression alone does not reliably yield maximal performance, solving your expressed concern.
> > - We provide the requested comparative analysis of sampling speed of diffusion planners for all main experiments in Appendix C2. We compare performance and computational overhead (both in neural evaluations and decision frequency) of DiffuserLite, Diffusion Veteran, and TGDP-Veteran. Similarly to the ablations on accelerated sampling in DiffuserLite, we investigate the trade-offs between performance and sampling speed under reduced denoising steps.
> > - We provide additional experiments on complex manipulation environments in Appendix C3. For this we implement both the advanced tasks from the DiffuserLite paper: RoboMimic, and the advanced tasks from the Diffusion Veteran paper: Adroit Hand. This raises the number of tasks we compare our approach on from 14 to 28.
> > - We have added crossreferences to the new Appendices in the main body and added descriptions of the new datasets to Appendix E.
> >
> > **Clarifications**\
> > - *1. Quality*: The new ablation disproves the concern that improvement results purely from weighted regression.
> > - *4. Method weakness*: We seem to have misinterpreted your original question. This is actually a good example of where conditioning on temperature mitigates issues of return-conditioning. For return-guided CFG in sparse reward environments, conditioning on a fixed target return can be misaligned with the current environment state. E.g., if we are in the initial state (and inpaint this state during sampling) and condition on target return 1.0 it is impossible to simultaneously fulfill the state constraint and return condition, thus risking out of distribution samples. By contrast, in TGDP we condition on a temperature that aligns diffusion targets with the latent return gradients (outlined in Appendix A). In the example from above, we therefore do not explicitly condition on a specific target return but instead implicitly condition on “relatively high” return. Through this we bypass CFG’s problem of unachievable (state, condition) combinations. Note that we also compare with other methods on the sparse reward Maze2D environments.
> >
> > We hope to have fully addressed all your concerns and questions and thank you again for the invested time and effort that has helped us to significantly improve our paper.

---

### Official Review · Reviewer_grLj · 2025-11-01

**Soundness:** 3
**Presentation:** 2
**Contribution:** 3
**Rating:** 6
**Confidence:** 3

**Summary:**

The paper introduces a Temperature-Guided Diffusion Planner (TGDP) to address the limitations of traditional reward guidance based on classifier-free guidance (CFG), which is often inflexible and not scalable due to its high sensitivity to hyperparameters. In this work, the authors incorporate a temperature parameter as an auxiliary input to the diffusion model. During training, the diffusion model is optimized with temperature-weighted sampling to capture reward-aware distributions, and during inference, the temperature is adaptively utilized to control the guidance strength in the sampling process. Experimental results demonstrate that TGDP achieves significant improvements over conventional diffusion-based planners.

**Strengths:**

1. The paper introduces a novel idea of adaptively adjusting the level of reward guidance, which is both meaningful and practical for diffusion-based planning.

2. The authors provide comprehensive theoretical analysis on the application of temperature in both training and guidance, presented in the appendix.

3. Extensive experimental results are included to validate the effectiveness of the proposed method, along with detailed implementation information.

**Weaknesses:**

1. The method section is somewhat disorganized, as parts of it include content that would be more appropriate for the experimental section (e.g. detailed descriptions of the D4RL implementation). This mixing of methodological explanation and implementation details makes it difficult to follow the core ideas of the proposed approach.

2. In addition, there are numerous formatting issues throughout the paper, including unusually large spacing between text, figures, and tables. Several figure legends overlap with the plots themselves, obscuring key details and making it hard to interpret the results clearly.

**Questions:**

1. How can the authors ensure that trajectory returns are effectively encoded through temperature-weighted training? Would this implicit formulation reduce scalability or generalizability compared to traditional reward-guided inference methods, especially when the reward function changes across tasks?

2. The paper states that traditional reward guidance is sensitive to the hyperparameter controlling guidance strength. However, could the maximum temperature $\beta_\text{max}$ itself become an important hyperparameter that significantly affects the performance of TGDP?

---

> ### Author Response · Authors · 2025-11-20
> **Response 1/2 to Reviewer grLj**
>
> Thank you very much for the thoughtful evaluation of our work and the actionable advice on increasing its quality. We appreciate the recognition of the novelty and relevance of TGDP. In the following we want to address raised issues and questions individually to clarify unclear points and improve the quality of our paper.
>
> **Weakness 1: Unclear separation of methodology and experimental sections**
>
> Thank you for pointing out that the organization of the methodology section is not ideal for readability and understandability. We decided to interleave methodology with quantitative demonstrations to progressively motivate TGDP's components: (1) temperature-weighting for skewed sampling, (2) diffusion guidance through different temperature conditions, and (3) collinearity-based automatic adaptation. We implement this narrative in Section 3, Appendix A (formal derivations), and Appendix B (additional evaluations). We consider this structure beneficial, as it directly shows the respective role of the individual components, rather than extensively introducing back and forth crossreferences. However, we agree that this organization deviates from common practices and might hinder readability.
>
> To improve organization and flow:
> - We adapted the methodology section to increase readability through a clearer separation of the individual components of the algorithm (reweighting, guidance, and adaptive scaling) as individual paragraphs. For each component, we separate a) its motivation and realization and b) the qualitative demonstration of the component’s effect as individually set textblocks.
> - To introduce this structure, we added a sentence in the first paragraph of Section 3: *”To progressively motivate our design decisions, for each individual component of TGDP, we first explain its respective intuition and realization, and then discuss qualitative evaluations to illustrate its effect.”*
> - We moved the detailed introduction of the D4RL dataset to Appendix D.
> - We added a figure to visualize training and inference of TGDP for a concise overview of the method.
>
> We think that these changes significantly improve readability and hope you share this view.
>
> **Weakness 2: Formatting errors in figures**
>
> Thank you for pointing this out. We have adapted all figures to reduce unnecessary whitespace and prevent legends from blocking relevant information.
>
> **Question 1: Alignment of temperature and trajectory returns**
>
> In Appendix A we formalize the temperature-based reweighting and show that higher temperatures bias the sampling distribution towards high performing subspaces. In contrast to return-conditioned CFG—which conditions on explicit return values—the implicit temperature-conditioned diffusion targets progressively align with the latent gradients of the return distribution. As a consequence TGDP is less prone to misaligned (state-)conditioning and (return-)guidance, increasing generalization and scalability. While we do not see any reason why any component of TGDP should reduce generalization or scalability across tasks, we do acknowledge that there might be dataset geometries and return surfaces that are better captured by other conditioning approaches.
>
> To present results and limitations from our formalization more clearly:
> - We added crossreferences between the main body of the paper and the relevant parts of Appendices A and B to facilitate the search for extended information and formalization of the individual parts of the algorithm.
> - We added a Section “Limitations and Future Work”, discussing limitations of the method and potential issues from specific dataset geometries: *”[...] unclear generalization to arbitrary task settings. In our derivations in Appendix A, we constrain datasets to consist of well-separated behavioral modes that individually emit smooth value surfaces. [...] there may exist datasets that violate the proposed constraints or interact adversely with our proposed framework, leading to diminished performance or even catastrophic failure of TGDP. [...] Exploring generalization to more diverse datasets, including complex robotic environments with sensory state input, as well as a further formal analysis of temperature guidance therefore remain exciting directions for future work.”*
>
> We hope to have understood your question correctly and have answered it fully.

---

> > ### Author Response · Authors · 2025-11-20
> > **Response 2/2 to Reviewer grLj**
> >
> > **Question 2: The $\beta_{max}$ hyperparameter**
> >
> > Thank you for pointing out that we do not give proper guidance on the selection of $\beta_{max}$. In our experiments we found that TGDP works robustly for broad ranges of this parameter. Importantly, it is not a task-specific parameter but can be set to a default value across tasks. It should be noted though that the temperature $\beta_{max}$ and the guide scale range $s’$ have to be set jointly as they interact multiplicatively (see Appendix A2).
> >
> > To provide guidance on selecting $\beta_{max}$ :
> > - We added its discussion to the experiments section: *”[...] Here, TGDP's temperature hyperparameter is not swept over but fixed to $\beta_\text{max}=3$. In practice, this value should be sufficiently large for temperature-conditioned diffusion targets to diverge during inter-modal steering, which is critical for the validity of the adaptive guide scale. At the same time, it should remain small enough to prevent single reweighted samples from dominating the loss, since excessive weighting can destabilize training. As analyzed in Appendix B.3 for values $\beta_\text{max}\in\{1,3,5\}$, maximal performance across datasets can be obtained for wide temperature ranges. However, because temperature and guide scale interact multiplicatively, adjusting $\beta_\text{max}$ also necessitates corresponding retuning of $s'$.”*
> > - We added the parameter $\beta_{max}$ to Table 1 to increase its visibility.
> >
> > **Conclusion**
> >
> > We sincerely appreciate your thorough and constructive feedback, which is invaluable towards refining and clarifying our work. In response, we have made significant improvements, hoping to increase the clarity of the presentation and quality of the paper.
> > The changes include a refinement of the structure of the methodology section and a tighter coupling with the related Appendices A and B, a figure illustrating training and inference of TGDP, improvements in the formatting of figures, the addition of a new section “Limitations and Future Work”, and a clearer presentation of the role of $\beta_{max}$ as well as guidance on how to select it. We hope to have fully addressed your questions and concerns and were able to further improve your view on the quality of our paper. We look forward to a productive discussion.

---

### Author Response · Authors · 2025-11-20
**Global Response**

We sincerely thank all reviewers for their valuable comments and suggestions that helped us to improve our paper. We particularly appreciate the expressed recognition of the novelty and relevance of our method. In response to reviewers’ questions and concerns we have updated the manuscript, significantly extending experimental evaluations and improving content and readability. Below we outline individual improvements and list [reviewers that requested it]. Changes in the manuscript are marked in blue.

**Improvements:**
- We provide ablations of the individual components of TGDP for all main experiments in Appendix C1. [Reviewers Payn, McDw]
- We compare trade-offs performance and computational overhead of diffusion planners for all main experiments in Appendix C2. [Reviewers Payn, 1nMT]
- We provide additional experiments on RoboMimic and Adroit Hand in Appendix C3, raising the number of evaluation tasks from 14 to 28. [Reviewers Payn, McDw, 1nMT]
- We revised the methodology section, including an explanatory figure, a clearer structure, and added cross-references to corresponding parts in the Appendix. [Reviewers grLj, Payn]
- We added a section "Limitations and Future Work", discussing computational overhead and generalization of the method. [Reviewers grLj, McDw, 1nMT]
- We added a discussion of the parameter $\beta_{max}$ and guidance on selecting it. [Reviewers grLj, Payn, McDw, 1nMT]
- We improved the formatting of several figures for better visibility. [Reviewer grLj]
- We extended the SotA, discussing recent approaches to avoid manifold deviation in diffusion planning. [Reviewer McDw]

We appreciate the time and dedication invested by reviewers, ACs and SACs and thank everyone for the highly productive reviews and discussions.

---

### Meta-Review · Area_Chair_Pdfw · 2026-01-07

**Summary:**

This paper introduces a Temperature-Guided Diffusion Planner (TGDP) to address the limitations of traditional reward guidance based on classifier-free guidance (CFG), and provided a set of experimental results to validate the proposed approach. It received review scores of (6, 4, 4, 6).
The reviewers raised several significant concerns.
First, the novelty is limited, as the core contribution focuses on a technical refinement of the CFG mechanism while incurring non-negligible computational overhead.
Second, the experimental results are relatively weak, with only marginal improvements over prior work.
Third, notable limitations remain: the adaptive guidance mechanism is largely heuristic, and its convergence under diverse diffusion architectures is not thoroughly analyzed.
In addition, the paper exhibits issues in organization, writing, and formatting.
Although the authors’ rebuttal provided additional results and clarifications, the reviewers remain unconvinced. Overall, the paper is considered borderline.

**Reviewer Concerns:**

The authors’ rebuttal provided the requested ablation studies and added results on additional benchmarks. However, some specifically requested benchmarks (e.g., AntMaze) were not included, and the comparative results remain weak. As a result, the reviewers remain unconvinced regarding both the novelty of the approach and the depth of the investigation into the adaptive guidance mechanism.

In addition, it is worth noting that Reviewer 1nMT initially posted an incorrect review and score intended for a different paper, which were subsequently corrected. However, the system currently displays only the initial version, and thus the incorrect review and score remain visible.

**Reviewer Scores:**

I feel the review score will remain unchanged on this paper.

---

### Decision · Program_Chairs · 2026-01-26

Reject